# A global model-measurement evaluation of particle light scattering coefficients at elevated relative humidity

María A. Burgos[1,2], Elisabeth Andrews[3], Gloria Titos[4], Angela Benedetti[5], Huisheng Bian[6,7],
Virginie Buchard[6,8], Gabriele Curci[9,10], Zak Kipling[5], Alf Kirkevåg[11], Harri Kokkola[12], Anton Laakso[12],
Julie Letertre-Danczak[5], Marianne T. Lund[13], Hitoshi Matsui[14], Gunnar Myhre[13], Cynthia Randles[6],
Michael Schulz[11], Twan van Noije[15], Kai Zhang[16], Lucas Alados-Arboledas[4], Urs Baltensperger[17],
Anne Jefferson[3], James Sherman[18], Junying Sun[19], Ernest Weingartner[17,20], and Paul Zieger[1,2]

[1]Department of Environmental Science and Analytical Chemistry, Stockholm University, Stockholm, Sweden
[2]Bolin Centre for Climate Research, Stockholm, Sweden
[3]Cooperative Institute for Research in Environmental Studies, University of Colorado, Boulder, USA
[4]Andalusian Institute for Earth System Research, University of Granada, Granada, Spain
[5]European Centre for Medium-Range Weather Forecasts, Reading, UK
[6]NASA/Goddard Space Flight Center, USA
[7]University of Maryland Baltimore County, Maryland, USA
[8]GESTAR/Universities Space Research Association, Columbia, USA
[9]Dipartimento di Scienze Fisiche e Chimiche, Universita' degli Studi dell'Aquila, L'Aquila, Italy
[10]Centre of Excellence CETEMPS, Università degli Studi dell'Aquila, L'Aquila, Italy
[11]Norwegian Meteorological Institute, Oslo, Norway
[12]Finnish Meteorological Institute, Kuopio, Finland
[13]Center for International Climate Research, Oslo, Norway
[14]Graduate School of Environmental Studies, Nagoya University, Nagoya, Japan
[15]Royal Netherlands Meteorological Institute, De Bilt, Netherlands
[16]Earth Systems Analysis and Modeling, Pacific Northwest National Laboratory, Richland, WA, USA
[17]Laboratory of Atmospheric Chemistry, Paul Scherrer Institute, Villigen, Switzerland
[18]Department of Physics and Astronomy, Appalachian State University, Boone, USA
[19]Key Laboratory of Atmospheric Chemistry of CMA, Chinese Academy of Meteorological Sciences, Beijing 100081, China
[20]Now at: Institute for Sensing and Electronics, University of Applied Sciences, Windisch, Switzerland

**Correspondence:** M.A. Burgos (maria.burgos@aces.su.se) and P. Zieger (paul.zieger@aces.su.se)

**Abstract.** The uptake of water by atmospheric aerosols has a pronounced effect on particle light scattering properties which in turn are strongly dependent on the ambient relative humidity (RH). Earth system models need to account for the aerosol water uptake and its influence on light scattering in order to properly capture the overall radiative effects of aerosols. Here we present a comprehensive model-measurement evaluation of the particle light scattering enhancement factor $f$(RH), defined as the particle light scattering coefficient at elevated RH (here set to 85 %) divided by its dry value. The comparison uses simulations from 10 Earth system models and a global dataset of surface-based in situ measurements. In general, we find a large diversity in the magnitude of predicted $f$(RH) amongst the different models which can not be explained by the site types. Based on our evaluation of sea salt scattering enhancement and simulated organic mass fraction, there is strong indication that differences in the model parameterizations of hygroscopicity and model chemistry are driving at least some of the observed diversity in simulated $f$(RH). Additionally, a key point is that defining dry conditions is difficult from an observational point

of view and, depending on the aerosol, may influence the measured $f(\mathrm{RH})$. The definition of dry also impacts our model evaluation because several models exhibit significant water uptake between RH=0 % and 40 %. The multi-site average ratio between model outputs and measurements is 1.64 when RH=0 % is assumed as the model dry RH and 1.16 when RH=40 % is the model dry RH value. The overestimation by the models is believed to originate from the hygroscopicity parameterizations at the lower RH range which may not implement all phenomena taking place (i.e. not fully dried particles and hysteresis effects). This will be particularly relevant when a location is dominated by a deliquescent aerosol such as sea salt. Our results emphasize the need to consider the measurement conditions in such comparisons and recognize that measurements referred to as 'dry' may not be dry in model terms. Recommendations for future model-measurement evaluation and model improvements are provided.

## 1 Introduction

The effects of aerosol particles on the climate system are well known and appear as a consequence of the aerosol-radiation interaction (i.e., by scattering or absorption of solar radiation), and the aerosol-cloud interaction (when aerosols act as cloud condensation nuclei or ice nuclei and thereby change cloud microphysical and radiative properties; IPCC, 2013). Atmospheric aerosol particles are critical forcing agents in the climate system and, despite the increased number of studies in recent years, aerosol forcing remains (together with clouds) the largest uncertainty in climate change predictions (e.g., Ramanathan et al., 2001; IPCC, 2013; Regayre et al., 2018).

Aerosol optical properties, such as the wavelength-dependent light scattering coefficient, $\sigma_{\mathrm{sp}}(\lambda)$, are often measured under dry conditions (relative humidity (RH) below 40 %), as recommended by international protocols (e.g., WMO/GAW, 2016). However, aerosol particles can undergo hygroscopic growth and their optical properties are different at ambient conditions. The response of an aerosol particle to the surrounding RH is dependent on its size and solubility. Aerosol optical properties are thus dependent on RH: water uptake modifies particle size and chemical composition (and thus the complex refractive index) and this, in turn, affects the aerosol optical properties.

The scattering enhancement factor, $f(\mathrm{RH}, \lambda)$, is a key parameter that describes the change in particle light scattering coefficient $\sigma_{\mathrm{sp}}(\lambda)$ as a function of RH:

$$f(\mathrm{RH}, \lambda) = \frac{\sigma_{\mathrm{sp}}(\mathrm{RH}, \lambda)}{\sigma_{\mathrm{sp}}(\mathrm{RH_{dry}}, \lambda)}. \tag{1}$$

$f(\mathrm{RH}, \lambda)$ typically increases with increasing RH and is larger than 1 if particles do not experience significant restructuring when taking up water (Weingartner et al., 1995). The scattering enhancement factor is one way to represent aerosol hygroscopicity and its direct effect on particle light scattering (Titos et al., 2016).

There have been multiple measurement-based studies focused on investigating the scattering enhancement factor measured at different sites around the globe; Titos et al. (2016) compared $f(\mathrm{RH}, \lambda)$ at many of these as a function of dominant aerosol type. In general, they showed that clean marine aerosols exhibit higher $f(\mathrm{RH}, \lambda)$ than is measured at sites with anthropogenic

influence, consistent with other studies (e.g., Wang et al., 2007; Fierz-Schmidhauser et al., 2010a; Zieger et al., 2013). In addition to assessing $f(\mathrm{RH}, \lambda)$ as a function of dominant aerosol type, more detailed investigations have also been done. Quinn et al. (2005) utilized co-located chemistry and $f(\mathrm{RH})$ measurements to develop a parameterization relating organic mass fraction and water uptake based on measurements at sites in Canada, the Maldives and South Korea. Zieger et al. (2010) analyzed aerosol water uptake using nephelometer measurements of wet and dry scattering coefficient, aerosol size distribution, and Mie theory at the Arctic site Ny-Ålesund. Svalbard. At Melpitz (a rural site in Germany), Zieger et al. (2014) found a correlation between the scattering enhancement factor and the aerosol chemical composition, in particular with the inorganic mass fraction. This linear relationship was extended for organic-dominated aerosol with observations from a boreal site in Finland (Zieger et al., 2015). Results from seven years of aerosol scattering hygroscopic growth measurements at the rural Southern Great Plains site in the USA indicated higher growth rates in the winter and spring seasons, which correlated with a high aerosol nitrate mass fraction (Jefferson et al., 2017). Burgos et al. (2019) created an open access database of scattering enhancement factors for 26 sites, covering a wide range of aerosol types whose optical properties were measured both long-term and as part of field campaigns.

An accurate estimation of aerosol effects on climate by Earth system models (ESMs) requires a realistic representation of aerosols (aerosol size distribution, mixing state, and composition).[1] Models must also be able to simulate processes in the aerosol life cycle such as primary emissions, new particle formation, coagulation, condensation, water uptake, and activation to form cloud droplets among others. Water uptake by aerosols affects not only their optical properties but also their life cycle by changing their size which can impact processes such as wet and dry deposition, transport, and ability to act as cloud condensation and ice nuclei (Covert et al., 1972; Pilinis et al., 1989; Ervens et al., 2007). Representing aerosol processes and properties in ESMs poses a great challenge due to the diversity and complexity of atmospheric aerosols. ESMs have implemented special modules and treatments for aerosols and the estimates of aerosol radiative forcing and climate impacts will be influenced by the uncertainties associated with the description of these processes. However, a compromise must be achieved between sufficiently representative aerosol and atmospheric process representations and the resultant computational cost (Ghan et al., 2012).

The effect of harmonized emissions on aerosol properties in global aerosol models was analyzed by Textor et al. (2007), who found that the aerosol representation is controlled, to a large extent, by processes other than the diversity in emissions. This implies that the harmonization of aerosol sources has only a small impact on the simulated inter-model diversity of the global aerosol burden and optical properties. Results are largely controlled by model-specific representation of transport, removal, chemistry and aerosol microphysics.

Previous model studies have suggested that water associated with aerosol particles can lead to significant differences amongst model estimates, and the assumptions about water uptake can have a noticeable effect. For example, Haywood et al. (2008) used tandem-humidifier nephelometer measurements from an aircraft to assess the parameterization of aerosol water uptake by the Met Office Unified Model. They found that ambient aerosols were simulated as being too hygroscopic relative to observations

---

[1]Note that we are here using the more general term of Earth system model, while keeping in mind that other definitions (e.g. global climate models, general circulation models, transport models, etc.) are commonly used as well.

as a result of being modeled as composed solely of ammonium sulfate. Zhang et al. (2012) demonstrated that there are significant differences in simulated aerosol water content due to changes in a model's scheme to predict water uptake. Myhre et al. (2013) explored direct aerosol radiative forcing from a suite of models, showing that the primary source of differences among model estimates of the mass extinction coefficient was aerosol hygroscopic growth of sulfate aerosols. Similarly, Reddington et al. (2019) studied the sensitivity of the aerosol optical depth (AOD) simulated by the GLOMAP model to assumptions about water uptake. They found that the AOD decreased when using the $\kappa$-Köhler (Petters and Kreidenweis, 2007) water uptake scheme relative to the AOD calculated using the Zdanovskii–Stokes–Robinson approach (Stokes and Robinson, 1966a). Moreover, Latimer and Martin (2019) also found that the implementation of the $\kappa$-Köhler hygroscopic growth for secondary inorganic and organic aerosols reduced the bias that appears in the representation of aerosol mass scattering efficiency relative to when water uptake was based on the Global Aerosol Data Set (GADS).

The Aerosol Comparison between Observations and Models (AeroCom) project (Textor et al., 2006; Schulz et al., 2006; Kinne et al., 2006, https://aerocom.met.no) aims to analyze global aerosol simulations to enhance understanding of aerosol particles and their impact on climate. In this project, intercomparisons among global aerosol models and comparisons with observations of aerosol properties have been carried out. These types of model evaluations allow for the identification of sources of model diversity and determination of which modeled aerosol properties need improvement. The objective of tier III of the INSITU measurement comparison experiment within AeroCom phase III (https://wiki.met.no/aerocom/phase3-experiments), is to assess how well model simulations represent observations of aerosol water uptake by comparing a high-quality, long-term, in situ measurements dataset with the output of several global aerosol models and that is what was done here.

In this paper, we present a comparison among scattering enhancement factors modeled by 10 different ESMs and observations. Our objectives are (i) to use measurements as a reality check on model simulations, (ii) to assess differences amongst model estimates of aerosol hygroscopic growth and then (iii) to suggest some potential reasons for any observed discrepancies, both between models and measurements and amongst models. This is the first comparison carried out for a wide suite of site types (covering Arctic, marine, mountain, rural, urban and desert stations) and ESMs, and is possible due to a newly published observational dataset of aerosol hygroscopicity (Burgos et al., 2019). A short description of the measurement dataset is presented in Sect. 2, while Sect. 3 gives a brief description of the models and the main references related to them. Section 4 shows the results of the model-measurement comparison for 22 sites and we evaluate the influence of different model choices about chemical species and mixing states on this comparison. We explore the importance of temporal collocation for three sample sites where temporal collocation is possible and use the unique chemical composition at one of these sites to interpret model results in the context of the hysteresis phenomenon. Finally, we demonstrate the importance of the definition of the dry reference relative humidity for hygroscopicity studies.

## 2 Measurements

In this study, measured particle light scattering enhancement factors, $f(\mathrm{RH}, \lambda)$, from 22 different sites covering a wide range of site types (Arctic, marine, rural, mountain, urban and desert) are used. Note that all results here will be shown for $\lambda$=550 nm;

$\lambda$ will be omitted in the equations and variable names and only mentioned when necessary. Table 1 summarizes the station location and acronyms, while Fig. S1 (in supplementary material) shows a map with the location of these sites, color-coded by site type. The $f(\mathrm{RH})$ measurement data comes from the openly available scattering enhancement dataset described by Burgos et al. (2019). Four sites from Burgos et al. (2019) dataset were excluded in this current analysis, either because they

had a small upper size cut ($\mathrm{PM}_1$ or $\mathrm{PM}_{2.5}$, i.e., particulate matter with aerodynamic diameters less than 1 or 2.5 $\mu$m) or a very low number of data points ($N$<10). This scattering enhancement dataset was developed from dry and wet particle light scattering measurements made as part of field campaigns and long-term monitoring efforts by the USA Department of Energy Atmospheric Radiation Measurements (DoE/ARM), the USA National Oceanic and Atmospheric Administration Federated Aerosol Network (NOAA-FAN, Andrews et al., 2019), the Swiss Paul Scherrer Institute (PSI), and/or the Chinese Academy

of Meteorological Sciences (CAMS).

The scattering coefficients were measured simultaneously under two different conditions. First, under so-called dry or low-RH conditions (namely RH < 40 %), hereafter referred to as $\mathrm{RH}_{\mathrm{ref}}$, and measured with a reference nephelometer or DryNeph. Typically $\mathrm{RH}_{\mathrm{ref}}$ in the DryNeph will vary over the interval 0-40 % but this variation will depend on the characteristics of the site, e.g., at some marine sites like at GRW, the measurement system was not able to dry the aerosol below 50 % RH during

some months. Data with $\mathrm{RH}_{\mathrm{ref}}$ > 40% were not included in this study. Figure S2 presents the probability density function of the measured $\mathrm{RH}_{\mathrm{ref}}$ for all sites. Secondly, the scattering coefficients were measured scanning over a programmable range of RH values, mainly between 40 and 95 %, with a second humidified nephelometer or WetNeph (Sheridan et al., 2001; Fierz-Schmidhauser et al., 2010b). The RH in the WetNeph is termed $\mathrm{RH}_{\mathrm{wet}}$. The wide range of scanned $\mathrm{RH}_{\mathrm{wet}}$ values were typically achieved by passing the aerosol particles through a humidifier system before they entered the WetNeph. One possible limitation

of this approach is that the sample air may not equilibrate if the residence time in the elevated relative humidity downstream of the humidifier is too short (Sjogren et al., 2007). However, the measurements performed by PSI at the European sites JFJ, MHD, CES and MEL (see summary in Zieger et al., 2013) and HYY (Zieger et al., 2015) were all accompanied by optical closure studies using Mie theory together with measured size distribution and chemical composition and/or hygroscopic growth factors, which revealed no apparent bias due to too short residence times downstream of the humidifier.

In order to create a benchmark dataset for aerosol scattering enhancement, an identical process for data treatment was applied to all initial raw scattering coefficients, and data quality was assured by a thorough inspection of the scattering time series for each site (Burgos et al., 2019). The final dataset is composed of yearly files organized in three levels, containing scattering coefficients, hemispheric backscattering coefficients, and scattering enhancement factors for three wavelengths (450, 550, and 700 nm) and two particle size cuts (aerodynamic diameters lower than 10 and 1 $\mu$m). Level 1 contains the raw scattering data,

Level 2 the corrected scattering coefficients and calculated scattering enhancement factors, and Level 3 contains the calculated $f(\mathrm{RH}=85\,\% / \mathrm{RH}_{\mathrm{ref}})$. A detailed description of the data screening process and the corrections applied, the specific wavelengths and size cuts at each site, as well as the design and characteristics of the different instrument systems are given in Burgos et al. (2019) and references therein. As part of the observational dataset development, uncertainty in $f(\mathrm{RH})$ was also determined. The uncertainty in $f(\mathrm{RH})$ depends on the aerosol load, RH and hygroscopic growth, and was found to vary between 10 and

30 % for $\mathrm{PM}_{10}$. Table 4 in Burgos et al. (2019) presents a detailed description of the uncertainty as a function of these variables.

One of the strengths of the dataset is that it was developed using a homogenized data treatment - differences in data processing was one of the issues cited in Titos et al. (2016) hygroscopicity overview paper that limited absolute comparisons of $f$(RH) values reported in the literature. The homogenized data treatment facilitates the intercomparison of the stations included in the dataset as well as the comparison against global model output. A full description of the homogenization process is given

in Burgos et al. (2019), and a summary of the process is presented here. The homogenization starts with the light scattering raw data provided by each site manager. Standard corrections are applied to all raw data in an identical manner, and in-depth data screening is carried out to identify data during invalid periods or system malfunctions. Several corrections are applied to the valid data periods: angular truncation and illumination non-idealities, adjustment to standard temperature and pressure, particles losses, and a 10-minute moving average is applied to the dry scattering coefficient series (this step is specially relevant

for pristine sites). Finally, the scattering enhancement factors are reported at common $RH_{ref}$ and $RH_{wet}$ which eliminates potential discrepancies among $f$(RH) values due to choice of RH (Titos et al., 2016), and allows direct comparison between sites. In this study, we use Level 2 $f$(RH=85 %/$RH_{ref}$=40 %) at $\lambda = 550$ nm data from 22 stations (those with $PM_{10}$ size cut or whole-air measurements) (see Table 1 for information about the station names, IDs, and aerosol types). The dry value of particle light scattering coefficient used to retrieve the scattering enhancement factor can be a) measured with the DryNeph at

any $RH_{ref}$<40 %, or b) extrapolated to exactly $RH_{ref}$=40 %. We first present the model-measurement comparison results using DryNeph RH values extrapolated to $RH_{ref}$=40 %. This is followed by a discussion on the implications of making different assumptions about the DryNeph RH value for both measurements and models.

In this study we utilise the scattering enhancement at $RH_{wet}$=85% to parameterize aerosol hygroscopicity. Choosing $RH_{wet}$=85% ensures that the reported $f$(RH) value represents the aerosol in the fully deliquesced state (upper branch of the hysteresis loop).

Scattering enhancement at specified RH is a simple metric. There are other methods, of varying complexity, that may also be used to describe the aerosol scattering enhancement; Titos et al. (2016) presents a review of the various empirical parameterizations found in literature that have been used to describe the relationship of $f(\mathrm{RH}, \lambda)$ and RH. The most common other algorithm is the two-parameter, power law fit referred to as the $\gamma$-fit (Hänel and Zankl, 1979). While fitting over the whole range of RH observations can provide valuable additional information about hygroscopic growth (e.g., investigating the RH

ceilings often assumed in models or as a means to identify deliquescence transitions (Zieger et al., 2010; Titos et al., 2014a)) that level of complexity was not desired in this initial model measurement comparison.

## 3   Models

In this section, we present the ten models used in this study. We first provide a brief description of their main characteristics and relevant references, where detailed information on each model's parameterizations/assumptions can be found. The models

used are: Community Atmosphere Model version 5 (CAM5), Aerosol Two-dimensional bin module for foRmation and Aging Simulation (CAM-ATRAS), the CAM5.3-Oslo (CAM-OSLO) model, the Goddard Earth Observing System with the MERRA Aerosol Reanalysis (GEOS-MERRAero), the Georgia Institute of Technology-Goddard Global Ozone Chemistry Aerosol Radiation and Transport model (GEOS-GOCART), the GEOS-Chem (GEOS-Chem) model, the Tracer Model (TM5), the Oslo

chemistry-transport model (OsloCTM3), the European Centre for Medium-Range Weather Forecasts - Integrated Forecasting System model (ECMWF-IFS) run in the Copernicus Atmosphere Monitoring Service configuration, and the global general circulation model ECHAM6 with the SALSA module (ECHAM6.3-SALSA2.0). For simplicity, we will refer to these models as: CAM, ATRAS, CAM-OSLO, GEOS-Chem, GEOS-GOCART, MERRAero, TM5, OsloCTM3, IFS-AER, and SALSA, respectively.

Table 2 summarizes some of the most relevant characteristics of each model, such as meteorology, mixing states, species and size bins. Table 3 summarizes the parameterization of hygroscopic growth for the chemical components in each model and provides the growth values $g$(RH) at 90% so that the model assumptions can be more readily compared. The model data used in this study were provided within the tier III of the INSITU measurement comparison experiment of AeroCom phase III (https://wiki.met.no/aerocom/phase3-experiments) and are composed of aerosol absorption and extinction coefficients at RH = 0, 40, and 85 %. Models also provided the mass mixing ratios for the chemical constituents they simulated, which we use to assess the impact of composition on hygroscopicity. Model values of scattering coefficient were obtained by subtracting absorption coefficient from extinction coefficient. The models were run for the year 2010 and data at surface level from 22 locations (closest gridpoint to the observational data) have been extracted. Exact temporal collocation between measurements and models can only be achieved at three of the measurement sites (BRW, GRW, and SGP), which made measurements in 2010. The model output files provide data at either 1h, 3h, or daily resolution, while the measurement data is primarily at hourly resolution with some of the more pristine sites averaged to six-hourly resolution (see Tables 1 and 2 for details).

All models considered in this study take into account topography. However, a model's surface elevation for a given gridbox will represent an average of the topography within the given gridbox. Nonetheless, we have used the surface values provided by the models for all sites in this study. For sites located in complex terrain the model surface values may not be representative of the measurement site and this will be exacerbated by models with coarser resolution. For example, Schacht et al. (2019) noted that complex local terrain near ZEP may have impacted their modeling efforts. In this study there is one mountain site (JFJ) in the Swiss Alps with an altitude of 3580 m a.s.l. and seven more sites with elevations above 200 m a.s.l. (APP, FKB, HLM, NIM, PGH, UGR, and ZEP at 1100, 511, 525, 205, 1951, 680, and 475 m a.s.l., respectively). The remaining 14 stations are at elevations lower than 100 m a.s.l. It should be noted that elevation alone does not describe the wider topography; for example, UGR is surrounded by nearby mountains with elevations above 3000 m a.s.l. (Titos et al., 2014b); while PGH is located on the edge of the Indo-Gangetic Plain in the foothills of the Himalayas (Dumka et al., 2017).

## 3.1 CAM5

CAM5.3 is one of the versions from the CAM family models used in this study. The run we work with provided data at surface level with a grid resolution of 1.9º latitude x 2.5º longitude, and at hourly frequency. CAM5.3 uses the modal aerosol module which provides a compromise between computational resources and a sufficiently accurate representation of aerosol size distribution and mixing states. However, depending on the selected number of modes and aerosol species in each mode, it can still incur differences among models. This model uses the version with three lognormal modes, MAM3, which is described in detail in Liu et al. (2012b). As a brief description, MAM3 has Aitken, accumulation and coarse modes and it assumes

that: a) primary carbon is internally mixed with secondary aerosol, b) coarse dust and sea salt modes are merged, c) fine dust and sea salt modes are similarly merged with the accumulation mode, and d) sulfate is partially neutralized by ammonium. Hygroscopicity is based on $\kappa$-Köhler theory (Ghan et al., 2001), and the values used for the different aerosol components are listed in Table S3 of Liu et al. (2012b).

To represent the meteorological field, the nudging technique (Newtonian relaxation) has been used, with horizontal winds nudged towards ERA-Interim reanalysis, following Zhang et al. (2014). The present day (year 2000) anthropogenic emissions are prescribed using CMIP5 emission data (IPCC, 2013). Natural wind-driven aerosol (dust and sea salt) emissions are calculated online. CAM5.3 accounts for the following important processes that influence aerosols: nucleation, coagulation, condensational growth, gas- and aqueous-phase chemistry, emissions, dry deposition and gravitational settling, water uptake,

in-cloud and below-cloud scavenging, and production from evaporated cloud and rain droplets. Details on the representation of these processes can be found in the supplemental material of Liu et al. (2012a).

### 3.2   CAM-ATRAS

In this case, the CAM model is used but the aerosol module is changed to the Aerosol Two-dimensional bin module for foRmation and Aging Simulation (ATRAS). The run we work with provided data at surface level with the same grid resolution

(1.9º latitude x 2.5º longitude) as CAM5.3, and at hourly frequency. Meteorological nudging was used for temperature and wind fields in the free troposphere (<800 hPa) by using the MERRA2 (Modern-Era Retrospective Analysis for Research and Applications) data.

This model takes into account the following aerosol processes: primary aerosol emissions, gas- and aqueous-phase chemistry, nucleation, condensation and evaporation, secondary organic aerosols processes, dry and wet deposition, aerosol activation to

cloud droplets and water uptake. In this study, aerosol particles from 1 to $10\,\mu$m in dry diameter are represented with 12 size bins for sulfate, ammonium, nitrate, sea salt, dust, organic aerosol (OA), and black carbon (BC). The aerosol module as well as details and references for the aerosol processes treatment can be found in Matsui et al. (2014); Matsui (2017) and Matsui and Mahowald (2017). Related to to water uptake, $\kappa$-Köhler theory is used with the hygroscopicity parameter $\kappa$ for each species given in Matsui (2017).

### 3.3   CAM-OSLO

In this case, the aerosol module OsloAero5.3 is applied in the atmosphere model CAM5.3, which runs with a grid resolution of 0.9º latitude x 1.25º longitude. A thorough description and general modelling and validation results from this aerosol module used in the atmospheric component CAM5.3-Oslo of the Norwegian Earth System Model (NorESM1.2) have been published by Kirkevåg et al. (2018).

For aerosols, the model represents sulfate, black carbon, primary and secondary organic aerosols, sea salt and mineral dust. The following processes are taken into account: nucleation, coagulation, condensational growth, gas- and aqueous-phase chemistry, emissions, dry deposition and gravitational settling, water uptake, in-cloud and below-cloud scavenging, and cloud processing. Unlike (e.g.) MAM3, this aerosol module makes use of a "production tagged" method to calculate aerosol size and chemical

composition. It describes a number of "background" log-normal modes that can change their size distribution due to condensation, coagulation, and cloud processing. A detailed offline size-resolving model carries out the corresponding aerosol micro-physical calculations, and a selection of results are stored in lookup tables. Hygroscopicity is estimated for each particle size and type by the use of the volume mixing rule for internal mixtures, adding (by condensation) water as a function of RH according to Köhler theory. In CAM-OSLO, optical parameters are found by interpolation in look-up tables at the actual RH in each grid-box and time. The model data is output at hourly frequency.

## 3.4  GEOS-Chem

GEOS-Chem is a community global three-dimensional Eulerian chemistry-model originally described in Bey et al. (2001) with updates that are described in http://acmg.seas.harvard.edu/geos/geos_chem_narrative.html (last accessed 28 November 2019). Here we use version 10-01 of the model. GEOS-Chem is driven by assimilated meteorological observations from the Goddard Earth Observing System (GEOS) of the NASA Global Modeling and Assimilation Office (GMAO). For this work, we use the GEOS fields version 5.2.0 degraded from the native resolution to the 2° x 2.5° simulation grid and 47 levels, for computational expediency. For anthropogenic emissions we use EDGAR 4.2 complemented with regional inventories where available (US, Canada, Mexico, Europe and East Asia).

The aerosol module employs a bulk mass approach for sulfate-nitrate-ammonium system and for BC and OA. Soil dust and sea salt are simulated with a sectional approach having four and two size bins, respectively. The aerosol optical properties are calculated from the simulated aerosol mass assuming log-normal size distribution with parameters taken from OPAC (Optical Properties of Aerosols and Clouds, Hess et al., 1998) and updated by Jaeglé et al. (2011) and Heald et al. (2014), adopting an external mixing representation. The hygroscopic growth factors are taken from Chin et al. (2002).

## 3.5  GEOS-GOCART

The Goddard Chemistry, Aerosol, Radiation, and Transport module (GOCART) (Chin et al., 2002, 2009) was implemented in the NASA GEOS global Earth system model to simulate aerosol processes of sources, sinks, transport, and transformation (Colarco et al., 2010; Bian et al., 2013, 2017). For this study, the aerosol species included are sulfate, dust, organic aerosol (OA), BC, and sea salt. The model is "replayed" from the MERRA meteorological analyses at the same spatial resolution produced by the NASA Global Modeling and Assimilation Office (Rienecker et al., 2011). Every 6 h the model dynamical state (winds, pressure, temperature, and humidity) is set to the balanced state provided by MERRA and then a 6 h forecast is performed until the next analysis is available. The GEOS model is run with a grid resolution of 0.5° latitude x 0.625° longitude and with 72 vertical layers from surface up to 0.01 hPa (about 85 km). Aerosols are considered to have different degrees of hygroscopic growth with ambient RH (with the exception of dust). The hygroscopic growth follows the equilibrium parameterization of Gerber (1985) for sea salt and OPAC (Hess et al., 1998) for other aerosols.

### 3.6 GEOS-MERRAero

The GEOS Earth System Model is a weather- and climate-capable model which includes atmospheric circulation and composition, as well as oceanic and land components. This model includes the same aerosol transport module based on the GOCART (Chin et al., 2002; Colarco et al., 2010) that is used in the previously described GEOS-GOCART. The specific version of GEOS used in this study also includes assimilation of bias-corrected Aerosol Optical Depth (AOD) from the Moderate Resolution Imaging Spectroradiometer (MODIS) sensors. This is the so-called MERRAero aerosol reanalysis (Buchard et al. (2015)). Driven by the MERRA meteorology, MERRAero was run at a global 0.5 x 0.625 latitude-by-longitude horizontal resolution with 72 vertical layers and 3-hour frequency. The data assimilation step provides a direct observational constraint on the simulated 550 nm AOD, but absorption, speciation and vertical distribution remain largely driven by the background simulation. Optical properties of the aerosols are primarily based on Mie calculations using the particles properties as in Chin et al. (2002) and Colarco et al. (2010) with spectral refractive indices and hygroscopic growth parameterizations primarily from the OPAC database (Hess et al., 1998). The Gerber growth curve (Gerber, 1985) is used for sea salt.

### 3.7 OsloCTM3

OsloCTM3 is a chemistry-transport model, described in detail in Lund et al. (2018). The model includes several updates with regards to its predecessor, OsloCTM2, particularly in the convection, advection, proto-dissociation, and scavenging schemes. OsloCTM3 is a global three-dimensional transport model that is driven by 3h offline meteorological forecast data from IFS ECMWF and CEDS emissions as described in Hoesly et al. (2018). With respect to aerosols, it includes BC, primary and secondary organic aerosols, sulfate, nitrate, dust and sea salt and its aerosol module is inherited from OsloCTM2, with the main updates described in Søvde et al. (2012) and Lund et al. (2018). The hygroscopic growth for sulfate, nitrate and sea salt follows Fitzgerald (1975), and for organic aerosols from fossil fuel emissions and of secondary origin from Peng et al. (2001), and finally Magi and Hobbs (2003) for biomass burning aerosols, see further description in Myhre et al. (2007). The parameterization from Fitzgerald (1975) on hygroscopic growth for inorganic aerosols has been shown to be very similar to using Köhler theory in OsloCTM3 (Myhre et al., 2004). The run used in this study has a grid resolution of 2.25º latitude x 2.25º longitude and daily frequency output was provided.

### 3.8 TM5

The Tracer Model 5 (TM5) is an atmospheric chemistry and transport model. The version used for this study is an update of the model described by van Noije et al. (2014). Essentially the same version was used to carry out the Tier I experiment of the INSITU project in 2016. For the study presented here, additional diagnostics were included in the model to assess the hygroscopic growth at varying relative humidity.

TM5 uses a regular grid with a horizontal resolution of 3º longitude x 2º latitude and 34 vertical levels. At high latitudes, the number of grid cells in the zonal direction is gradually reduced towards the poles. Dry deposition velocities and the emissions of DMS, sea salt and mineral dust are calculated on a 1º x 1º surface grid, and subsequently coarsened to the atmospheric

grid. The hygroscopic growth of the soluble modes follows the description in Vignati et al. (2004). For pure sulfate-water particles the water uptake is calculated using the parameterization from (Zeleznik, 1991). When sea salt is present in the soluble accumulation or coarse modes, the water uptake is calculated using the ZSR method (Stokes and Robinson, 1966b; Zdanovskii, 1948). Below relative humidities of 45 %, sea salt is assumed to be dry. Additional water uptake in the presence of ammonium-nitrate in the soluble accumulation mode is calculated using EQSAM (Metzger et al., 2002). BC, OA and dust do not influence the water uptake. For calculating the aerosol optical properties at relative humidities other than ambient conditions, additional diagnostic calls to M7 and EQSAM have been included to calculate the water uptake in the relevant modes at these RH values. Apart from the water content, all other aerosol components are kept at their levels calculated at ambient conditions.

## 3.9 IFS-AER

The European Centre for Medium-Range Weather Forecasts (ECMWF) Integrated Forecasting System (IFS), also used for numerical weather prediction, includes an optional aerosol module (AER). This is described in Morcrette et al. (2009), and an update regarding its parameterizations for aerosol sources, sinks and chemical production is provided in Rémy et al. (2019). Successive versions of this model, including the aerosol module, are used operationally to produce global analyses and 5-day forecasts for the Copernicus Atmosphere Monitoring Service. The version used here, however, does not correspond precisely to any operational version, and is based on cycle 43r1 but with a number of experimental additions - most notably an early version of the nitrate and ammonium aerosol scheme that is described in Bozzo et al. (2019). The configuration corresponds closely to the ECMWF-IFS-CY43R1-NITRATE-DEV submission to the AeroCom Phase III 2016 control experiment. In this configuration, the model runs with a grid resolution of approximately 40km. The data files provided have 3h frequency. Hygroscopic growth follows the description of Bozzo et al. (2019) for sulfates, sea salt and organic aerosols. This includes the parameterization of Tang (1997) for sea salt, and Tang and Munkelwitz (1994) for sulfates. The species taken into account are sea salt, desert dust, hydrophilic and hydrophobic OM, and BC and sulfate, nitrate and ammonium.

## 3.10 SALSA

SALSA is the sectional aerosol module that has been coupled to the ECHAM-HAMMOZ aerosol-chemistry-climate model framework. The model version used in this study was ECHAM6.3-HAM2.3-MOZ1.0. The detailed description of SALSA along with the details of its implementation and evaluation against several types of observations have been presented by Kokkola et al. (2018). The SALSA module describes aerosol size distribution with 10 size classes in size space which include two parallel externally mixed size classes for insoluble and soluble aerosol, thus tracking 17 size classes covering dry diameters from 3 nm to 10 $\mu$m. It simulates all relevant atmospheric aerosol processes including aerosol-cloud interactions. Simulated compounds are sulfate, organic aerosols, BC, sea salt, dust and water. The hygroscopic growth in SALSA is calculated according to the Zdanovskii-Stokes-Robinson (ZSR) equation described in Stokes and Robinson (1966b) assuming that the soluble fraction of particles is always in liquid phase. Simulations were run with T63 spectral resolution (approx 1.9º latitude x 1.9º longitude), with 47 vertical levels and hourly output frequency.

## 3.11 Model main characteristics: hygroscopic growth, size distribution, chemical composition, and mixing state

In order to have a complete vision of the main traits of the models used in this study, we summarize here some of their characteristics and try to group them when possible to facilitate the analysis of the results in the following section. The aerosol size distribution, chemical species, mixing state and assumed hygroscopicity of each species are essential to predict the enhancement in aerosol light scattering. The mixing state, species and the number of size bins for each of the models are provided in Table 2, while Table 3 presents the details about the hygroscopic parameterization and coefficients used for each chemical constituent.

The models assign the chemical species to one or more size bins as described in Table 2. The size bins are typically assigned modal parameters to account for a range of particle sizes. To properly assess the impacts of the disparate approaches to size distribution for the different species would require synthesizing the size assumptions onto a common diameter grid (e.g., Mann et al., 2014). Such an approach is outside the scope of this paper and, therefore, we will not consider assumptions related to particle size in evaluating water uptake differences amongst models. Such an effort could be of value to explore in future work.

With regards to chemical constituents, all models consider five basic species: sulfate, dust, sea salt, BC, and OA. Five models also include nitrate and ammonium (ATRAS, CHEM, OsloCTM3, TM5, and IFS-AER). In addition, TM5 includes methane sulfonic acid (MSA). Figure S4 in the supplemental material shows that, for each species simulated by the models, there are both similarities and differences at the different sites. For example, for some sites (e.g., GRW, MHD, PGH and NIM) the modeled chemistry is quite consistent across models. In contrast, at coastal sites in North America (PYE, THD, PVC and CBG) the contribution of sea salt can be quite variable, possibly depending on where in each model's gridbox the site is located. The GEOS family of models tend to simulate a larger contribution from dust at individual sites relative to other models - this is most obvious at the Arctic sites BRW and ZEP, but occurs at other sites as well.

In addition to differences in simulated chemistry, there are some differences in model assumptions about water uptake for the different species (see Table 3). The modeled hygroscopic growth in the ten models considered in this study can be either calculated by means of direct parameterization (e.g., GEOS-family models, OsloCTM3, TM5, and IFS-AER), methods based on different theories (e.g., $\kappa$-Köhler theory (Petters and Kreidenweis, 2007; Ghan et al., 2001) used by the CAM-family models, Zdanovskii-Stokes-Robinson (ZSR, Stokes and Robinson, 1966a) equation implemented in SALSA), or thermodynamic equilibrium models (e.g., EQSAM (Metzger et al., 2002) used by TM5 for nitrate). Some models provided hygroscopicity factors in terms of $g(\mathrm{RH} = 90\,\%)$ and others in terms of $\kappa$; the $\kappa$-values were converted to $g(\mathrm{RH} = 90\,\%)$ using $g(\mathrm{RH}) = (1 + \kappa * \mathrm{RH}/(1 - \mathrm{RH}))^{1/3}$ (see Petters and Kreidenweis, 2007, here ignoring the Kelvin effect). Note: $g(\mathrm{RH})$ is analogous to $f(\mathrm{RH})$, but represents the aerosol diameter enhancement due to water uptake instead of the scattering enhancement which is an optical property. A $g(\mathrm{RH})$ value of 1.0 indicates no hygroscopicity/water uptake, while increasing values of $g(\mathrm{RH})$ correspond to higher growth due to water uptake. The parameter $\kappa$ is an indicator of the water uptake for different chemical species.

All models assume similar hygroscopicity for sea salt, with $g$(RH) values ranging from 2.25-2.4, except MERRAero and GEOS-GOCART which utilize lower values (1.90-2.17 depending on the size bin). Sulfate hygroscopicity among models is quite homogeneous, with values ranging from 1.64-1.9. Black carbon is only considered to grow in the GEOS-family models. Organic aerosols are assumed to be non-hygroscopic or have low hygroscopicity except in the GEOS-family models and IFS-AER. Dust is assumed to be non-hygroscopic by most models, but CAM and CAM-Oslo consider $g$(RH) values of 1.17. The models that include nitrate and ammonium assume similar hygroscopicity for these two components, ranging from 1.64 to 1.87. In summary, then, one common trait of the three GEOS-family models is that they assign high hygroscopic values to all components, while the rest of the models assume black carbon, organics and/or dust will undergo little or no hygroscopic growth.

Previous studies have also evaluated the sensitivity of modeled aerosol optical properties to the mixing state assumptions. Curci et al. (2015) found significant differences in simulated ambient AOD between internally and externally mixed assumptions, while Reddington et al. (2019) found that simulated ambient AOD is relatively insensitive to mixing state assumptions, and suggested the bigger impact found by Curci et al. (2015) was due, mainly, to the different calculations of the aerosol number size distribution. Neither study specifically address the effect of the mixing state assumption on water uptake. The models used in this study utilize a variety of assumptions about mixing state as specified in Table 2.

## 4   Results

In this section we present the results showing the comparison between in situ measurements and the ten models described in the previous sections. We first provide a general comparison of scattering enhancement measured at 22 sites in the Burgos et al. (2019) dataset with model outputs. For this analysis, temporal collocation of model and measurement data is made on a climatological basis. Model output for the simulation year 2010 is selected only from those months where measurement data is available (regardless of the year the measurements were made). We included all model data for each month for a given site regardless of the number of measurement data points in that month and for that site. Analysis (not shown) requiring a constraint on the number of measurements in a month in order to include model simulations for that month suggested that our approach had minimal impact on the results. By selecting the entire month from the model dataset, the impact of interannual variability is minimized. An illustration of the possible impact of the difference between model and observational years can be found in the supplemental materials for the site SGP, which has the longest period of measurements (see Fig. S3). In Sect. 4.2 we perform a more detailed analysis for three sites that measured during 2010, and thus allow an exact temporal collocation with the models, collocating for day and month of the year 2010.

### 4.1   Comparison of modeled vs. measured $f$(RH)

Figure 1 shows the box and whisker plots of the particle light scattering enhancement factor $f$(RH=85 %/RH$_{\text{ref}}$=40 %), where the dry reference RH is taken at RH$_{\text{ref}}$=40 %, for both the measurements and models. Note that models CAM-OSLO and MERRAero have fewer extracted sites (18 and 21, respectively) than the available measurement stations. These models pro-

vided data extracted at site locations, rather than the full global simulation and four station locations (CBG, FKB, HLM, and LAN) were not requested from CAM-OSLO at the time of their model run and one (LAN) was not requested from MERRAero at the time of their run. The box edges represent the $25^{th}$ and $75^{th}$ percentiles, with a line for the median ($50^{th}$ percentile). The whiskers shows the range of the data expanding from the percentile $10^{th}$ to the $90^{th}$. The gray shaded area indicates the range of the $25^{th}$ to $75^{th}$ percentiles of the measurements and is plotted to facilitate comparison with the modeled values. This area represents the temporal variability over the time period of the $f$(RH) measurements for each site and does not include measurement error. The number of measurements for each individual site is provided in the top right corner of each plot. As noted above, the model statistics shown represent the same months as the measurements, but the measurement year may not match the model year. For example, MHD has measurements during January and February of 2009, so model data shown for MHD has been restricted to January and February for model year 2010. The sites are organized by site type: Arctic (BRW, ZEP), marine (CBG, GRW, GSN, MHD, PVC, PYE, THD), mountain (JFJ), rural (APP, CES, FKB, HLM, HYY, LAN, MEL, SGP), urban (HFE, PGH, UGR) and desert (NIM).

In general, the top 10 panels (Fig. 1 a-j), comprising the Arctic, marine and mountain sites, and the desert site (Fig. 1 v) tend to exhibit the best agreement among the models and the measurements (i.e., more models fall within the shaded area). These sites tend to be the furthest away from local sources and may be more representative of a larger area. Two sites (CBG and PVC) both on the north-eastern coast of North America (CBG is in Nova Scotia and PVC in coastal Massachusetts) are less well simulated; in both cases the models tend to simulate larger scattering enhancement than is observed. Titos et al. (2014a) showed that there were significant differences in $f$(RH) at PVC depending on whether the sample air was urban influenced or predominantly marine. The rural and urban sites (Fig. 1 k-u) tend to exhibit lower scattering enhancement than is simulated by the models. In this second group, the sites CES and MEL are the exception, with most of the models falling in the shaded area, and occasionally below the shaded area.

Overall, high variability among the models is observed. The CAM-family models (ATRAS, CAM, and CAM-OSLO) exhibit differences among themselves and also, in general, large variability of $f$(RH) values within each model. In contrast, the three GEOS models (GEOS-Chem, GEOS-GOCART and MERRAero), OsloCTM3, and IFS-AER exhibit similar predicted scattering enhancement values and quite narrow variability in $f$(RH) within each model. One possible explanation for the fact that GEOS-family models generally show lower median values of $f$(RH) could be that they simulate a larger relative contribution of dust to the aerosol load (see Fig. S4 in supplementary material) which is considered to be non hygroscopic. This could explain the results found at the Arctic sites as well as GSN, JFJ, APP, MEL, SGP and UGR. However, the GEOS-family models also simulate lower $f$(RH) values for some other sites (e.g., GRW, MHD, PVC, THD and CES) where they don't simulate a large contribution from dust. Additionally, OsloCTM3 and IFS-AER do not simulate enhanced dust contributions so dust is unlikely to be the sole explanation. TM5 and SALSA exhibit the largest variability within their results, as can be seen at some rural (e.g., APP, CES, HYY, and SGP) and urban sites (HFE, PGH, and UGR).

In general, most of the models tend to overestimate $f$(RH) at almost all site types. There are several sites that most models consistently overestimate, for example: CBG, APP, FKB, HYY, LAN, PGH and UGR. For some sites this may be due to complex topography and emissions sources that are not adequately captured by the models. For example, Granada (UGR)

is surrounded by mountains and is impacted by desert dust from the Saharan desert and black carbon originating from local emissions (e.g., traffic and biomass burning, Titos et al., 2017). Similarly, PGH is in the foothills of the Himalayan range and is influenced by local and transported aerosol plumes (Dumka et al., 2017), and LAN is a polluted background station representative of the Yangtze River Delta conditions, influenced by anthropogenic emissions and dust (Zhang et al., 2015).

However, there is no clear pattern in the chemistry simulated at each site (e.g., Fig. S4 in supplemental materials) that would explain this overestimation.

The data shown in Fig. 1 can be visualized in a different way in order to more readily see the relation between modeled and measured data for each model rather than for each site. Figure 2 shows the mean and standard deviations of the modeled versus measured $f(\text{RH}=85\,\%/\text{RH}_{\text{ref}}=40\,\%)$ for each model, color-coded by site type. The one to one relationship is indicated by a solid

black line and the gray dashed lines represent 30 % uncertainty bounds which is the maximum uncertainty of the measurements as described in Burgos et al. (2019). The CAM-family models, TM5 and SALSA exhibit a tendency to overestimate $f(\text{RH})$. The figure also shows a wide diversity between modeled and measured $f(\text{RH})$ for the different models. For example, the CAM-family models and TM5 exhibit a wider range in $f(\text{RH})$ relative to the GEOS-family models and IFS-AER, which exhibit very little range in $f(\text{RH})$.

The other models mostly fall within the 30 % interval of (upper) measurement uncertainty estimate (Burgos et al., 2019). CAM, CAM-Oslo, and OsloCTM3 are the models that most accurately estimate $f(\text{RH})$ at all site types, with the simulated results falling closest to the 1:1 black line and being within the 30 % interval. The Pearson correlation coefficient is also shown in the left top corner of each panel. The best correlations are found for CAM-Oslo, CAM, and OsloCTM3 with $r = 0.78$, 0.71, and 0.72, respectively. The GEOS-family models have correlation coefficients close to 0.5, while SALSA exhibits negative

correlation with the measurements.

Previous studies (Burgos et al., 2019; Titos et al., 2016) found the largest values of $f(\text{RH})$ for Arctic and marine sites and lowest for urban, desert and polluted sites. CAM and TM5 (and to a lesser extent CAM-OSLO) appear to replicate the observed pattern of the Arctic and marine sites having higher $f(\text{RH})$ than other sites. ATRAS and SALSA are similar in that they tend to simulate higher $f(\text{RH})$ values for marine, rural, and urban sites and lower for Arctic locations, with ATRAS predicting the

highest hygroscopicity at rural sites. The GEOS-family and IFS-AER do not exhibit a large enough range in simulated $f(\text{RH})$ to determine if some site types are more hygroscopic than others.

It is useful to consider what causes the discrepancies between models and observations. Potential explanations for the model overestimates of $f(\text{RH})$ may be related to model assumptions about chemistry (e.g., the species included, hygroscopicity parameterizations for those species, assumptions about hysteresis, mixing state, etc.) or size distribution. We have already

noted that it is beyond the scope of this paper to consider the impact of aerosol size distribution on scattering enhancement, but below we discuss hygroscopicity in relation to hysteresis, mixing state, hygroscopcity parameterization and chemical composition. Table 3 summarizes the parameterizations used as well as the hygroscopic growth factors, g(RH), at RH=90% and $\kappa$ parameters so that the model assumptions of hygroscopic growth can be more directly compared.

A deliquescent aerosol can exist in the liquid and solid phases at the same RH, an effect known as hysteresis (Orr et al., 1958).

This means that, below its deliquescence RH but above its efflorescence RH, the corresponding scattering will be different

depending on whether it is in a liquid or dry state. Deliquescent aerosols are typically inorganic species such as ammonium sulfate and sodium chloride. Modelling hysteresis is complex as the behavior differs for aerosols of mixed composition, relative to single component particles. The hysteresis effect is unlikely to be the cause of differences amongst the models as it has only been accounted for by two of the models considered in this study (CAM and CAM-Oslo). Moreover, $f$(RH) was calculated

at RH=85 % to minimize discrepancies due to hysteresis because at that RH the particles will have undergone deliquescence. However, models may make different assumptions about water uptake at low RH which will affect $f$(RH) by impacting the denominator of the scattering enhancement equation, which will be of importance of strongly deliquescent aerosol. This is explored in more detail in sections 4.3 and 4.4.

The mixing state is another model assumption that could play a role in the observed differences amongst models. Curci et al.

(2015) reported that aerosol optical properties calculated from bulk aerosol models which assume external mixing may be inherently different from the optical properties calculated from more detailed microphysical models which assume internal mixing. In contrast, Reddington et al. (2019) found modeled aerosol optical properties to be insensitive to mixing state and suggested the differences described in the Curci et al. (2015) study were more related to assumptions about size distribution than mixing state. In this study, a commonality among the models exhibiting low variability in $f$(RH) (e.g., the GEOS-family

models and IFS-AER), is that they assume an external mixing state (Table 2). SALSA, however, also assumes an externally mixed aerosol but does not exhibit the narrow range in $f$(RH) seen for the other models making this assumption. This suggests that mixing state assumptions may not be the reason behind these differences, although we are unable to evaluate this further.

The role played by the different parameterizations of aerosol water uptake has also been studied (Reddington et al., 2019; Latimer and Martin, 2019). Reddington et al. (2019) demonstrates that simulated AOD is sensitive to this assumption. Their

results show that using the $\kappa$-Köhler theory to describe hygroscopicity decreases AOD significantly relative to the AOD simulated when the ZSR equation is used calculate aerosol water uptake. A comparison of SALSA (which uses ZSR) with the CAM-family of models (which use $\kappa$-Köhler) in Fig. 1 does not reveal a consistent pattern; sometimes the $f$(RH) is higher for SALSA and sometimes for one or more of the CAM-family models. Since there are other differences amongst these models as well (e.g., simulated chemistry and size), it is impossible to assess the impact of these two different hygroscopicity parameter-

izations here. Latimer and Martin (2019) shows significant differences in mass scattering efficiency when $\kappa$-Köhler theory is used rather than GADS (Global Aerosol Dataset) to parameterize water uptake; they find that GADS results in an overestimate of mass extinction efficiency relative to $\kappa$-Köhler. The GADS parameterization is discussed in more detail below.

As noted in Section 3, the hygroscopicity values are generally quite similar for sea salt, sulfate, and dust for all models. There are, however, large differences for BC, POA and SOA amongst the models. The GEOS-family of models assign significantly

higher growth for these three species than assumed by the other models. This may, in fact, be the explanation for the narrow range of $f$(RH) exhibited by the GEOS-family of models - regardless of the simulated composition there will always be a large amount of water uptake. In contrast, the other models can simulate a wider range of $f$(RH), i.e., from low to high $f$(RH), as the proportions of the chemical constituents shift.

The GEOS-family models all use GADS by Köpke et al. (1997) (or OPAC by Hess et al., 1998, which uses essentially the

same values) to parameterize hygroscopicity. This simplified aerosol property model provides size and hygroscopic growth

parameters of six components (for various size ranges) at selected RH values, where models often use linear interpolation. Zieger et al. (2013) and, more recently, Latimer and Martin (2019) have shown that OPAC can be problematic for modeling hygroscopicity as it results in an overestimate of $f$(RH) at low and intermediate RH. Our analysis suggests another implication of that overestimate - the inability to simulate the range of scattering enhancement factors observed by measurements.

Our study provides the opportunity to challenge the models with a composition-based parameterization of $f$(RH) using the model-simulated chemistry to constrain model estimates of $f$(RH). Previous experimental field work has shown that aerosol hygroscopicity can be parameterized as a function of aerosol composition (Quinn et al., 2005; Zhang et al., 2015; Zieger et al., 2015) without any a priori assumptions about species-dependent water uptake. The simplest parameterization by Quinn et al. (2005) utilizes measured sulfate and organic aerosol mass concentrations to estimate organic mass fraction

(OMF=OA/(OA+sulfate)) and relates OMF to observations of $f$(RH) at three sites (CBG, KCO, GSN). They find that low OMF tends to result in high $f$(RH) and vice versa. More recent efforts (Zhang et al., 2015; Zieger et al., 2015) also applied the simple Quinn parameterization but determined that, for their sites, a more complete chemical characterization (i.e., considering more species) resulted in better correlation between observed chemical composition and $f$(RH). Here, we only compare with the simple Quinn parameterization because there is a disconnect between the measured species considered in the enhanced

parameterizations and the components simulated by the models. Figure 3 shows $f$(RH=85 %/RH$_{\text{ref}}$=40 %) as a function of the OMF simulated by each model. Each point represents one site, color-coded by site-type. Lines representing the relationship between OMF (as defined above) and $f$(RH) observed at different sites by Quinn et al. (2005), Zieger et al. (2015) and Zhang et al. (2015) are displayed as different lines on the figure. Note that the fit lines from Zieger et al. (2015) and Zhang et al. (2015) only represent their fits based on organics and sulfate rather than the relationships they developed using more detailed

chemistry.

Several things can be observed in Fig. 3. First, the models consistently simulate lower OMF values for marine and Arctic sites relative to those simulated for rural, urban, mountain and desert sites. However, those lower OMF values do not correspond to higher $f$(RH) for all models. The CAM-family models, OsloCTM3, and TM5 exhibit similar behavior to the Quinn et al. (2005) parameterization, with $f$(RH) inversely related the OMF. In contrast, the GEOS-family of models, and IFS-AER exhibit

no relationship between OMF and $f$(RH) and SALSA simulates a positive relationship (opposite to what is observed). The models that best reproduce the observed relationship between $f$(RH) and OMF are those that assume lower hygroscopicity for organics - this allows these models to simulate a wider range of $f$(RH) than if organic is assumed to have similar hygroscopicity characteristics as other considered species.

## 4.2   Investigating the importance of temporal collocation at BRW, GRW, and SGP

Temporal collocation of model data with observational data is an important aspect in model-measurement evaluation exercises (Schutgens et al., 2016). The model runs were conducted to simulate the year 2010 and three sites provide data covering almost that entire year. These sites exhibit distinct differences in their prevalent aerosol type: BRW, an Arctic site, GRW, a marine site, and SGP, a rural site. Temporal collocation has been carried out (Fig. 4) by selecting only those model data sampled at the same hour, day, and month (only day and month for OsloCTM3 and GEOS-GOCART models) with valid measurement data.

Because the focus in this section is to study the importance of temporal collocation, no threshold on number of data points within each month was required; the number of data points in each month are provided in Fig. 4 to give an indication of the representativeness (or lack thereof) of the monthly value.

Figure 4 shows, in the left column, the annual cycle (monthly medians) of the scattering enhancement factor for $f$(RH=85 % /
RH$_{\text{ref}}$=40 %). The black lines represent the observations (solid line: year 2010 only, dashed line: all available measurements, gray area: interquartile range of all measurements), and the colored lines the estimates by the different models. The observations from 2010 do not show obviously different characteristics compared to the climatology of the entire dataset for each site. The exceptions are for BRW in the latter half of the year where the 'all data' climatology is $\sim$12 % lower than the 2010 values, and for SGP where August and October exhibit monthly 2010 values lower than the climatological values (28 % and 20 %,
respectively). In general, the variability in the measured monthly $f$(RH) is significantly narrower than the range of $f$(RH) simulated by the models, suggesting exact collocation in time will have a limited impact on the overall model-measurement comparison. Using all observational data allows extension of the comparison to additional months which were not covered in 2010. Figure S3 shows the annual cycle in $f$(RH) for each individual year of measurements at SGP, the site with the longest time coverage (1999-2016); just 3 out of 18 years exhibit deviations from the climatological values larger than 50 %, suggesting
the climatological values are a reasonable proxy for comparison with model values.

Measurements at GRW and SGP do not exhibit a marked seasonal cycle in $f$(RH) , although the $f$(RH) observed at GRW exhibits slightly lower values during April, May, and June. The seasonal cycle appears to be much larger for BRW, with larger values occurring in the second half of the year. None of the models reproduce the observed annual cycle at BRW; some models (ATRAS, CAM-Oslo, GEOS-GOCART, GEOS-Chem, MERRAero, OsloCTM3, IFS-AER, and SALSA) are better in the early
part of the year and fall within the observed interquartile range, while CAM is closer to the observations in the latter part of the year. TM5 exhibits a clear bias towards larger values at BRW. At GRW, only CAM-Oslo reproduces the slightly lower values observed in late spring and early summer, though it is biased towards larger values. TM5, again, shows the largest bias with respect to the measurements. The rest of the models agree better in terms of magnitude of $f$(RH), but do not track the observed seasonal cycle. At SGP, most models reproduce the lack of seasonal cycle suggested by the observations. Only
ATRAS indicates a strong seasonal cycle which is not observed in these co-located measurements, although Jefferson et al. (2017) report a seasonal cycle for observed $f$(RH) at SGP similar to that simulated by ATRAS in shape but with a $f$(RH) narrower range. SALSA and TM5 both overestimate the observed $f$(RH). For SGP, the GEOS-family models, OsloCTM3 and IFS-AER fall within the observed interquartile range throughout the year.

This modeled seasonality (or lack thereof) is easier to quantify using Taylor diagrams as discussed below. To the right of each
annual cycle plot in Fig. 4 there is a Taylor diagram (Taylor, 2001) showing the skill of the models for these three sites when the model results are collocated both in time and space with the measurements. Taylor diagrams are used to provide a concise statistical summary on how well models match measurements in terms of standard deviation (represented by the radial distances from the origin to the points) and correlation coefficient (represented by the angle from the normal). Black symbols represent the in situ measurements and colored symbols represent the different models in our study. Note that standard deviation and
correlation coefficient have been calculated from all the collocated instantaneous values. The correlation coefficients are quite

low, suggesting that the models do not capture the monthly variability seen in the measurements. The correlation coefficients are only larger than 0.3 for GEOS-GOCART ($r$=0.38 at BRW ), and OsloCTM3 ($r$=0.36, 0.3 at GRW and SGP, respectively). Negative correlation coefficients are also found for some models at the three sites. The models exhibit a fairly wide range of standard deviations, SD (between 0.1 and $\sim$0.7, depending on model and site), with values both above and below the SD

observed for the measurements. The standard deviation is largest ($>$0.4) for CAM at the three sites and for TM5 at BRW and SGP. The Taylor diagrams suggest a lack of skill in the models at simulating the seasonality and variability of observed aerosol hygroscopicity even when the data are exactly temporally collocated.

Changes in both aerosol composition and size can cause changes in scattering enhancement (e.g., Zieger et al., 2010; Titos et al., 2014a). Such changes could be driven by annual circulation changes bringing different air masses to a site (Sherman et al.,

2015) and/or by normal variability in sources over the year. Both direct measurements of aerosol size distribution and indirect proxies such as the scattering Ångström exponent suggest there are seasonal shifts in aerosol size at these three sites (e.g., Quinn et al., 2002; Marinescu et al., 2019; Pio et al., 2007). Similarly, aerosol composition shifts as a function of season have also been reported for these sites (e.g., Quinn et al., 2002; Parworth et al., 2015; Logan et al., 2014). An in-depth evaluation of observed and modeled seasonal composition cycles at the 22 sites considered in our study is outside the purview of this

paper. However, we can look beyond the annual mass mixing ratio comparisons (Fig. S4, discussed in the previous section) to differences in the modeled monthly composition which may contribute to the variability in the modeled seasonal $f$(RH) shown in Fig. 4. Figures S5, S6 and S7 show the monthly variation in mass mixing ratio for the ten models considered in this study and for the year 2010 for these three sites. There is a fair amount of variability amongst the models in the simulated aerosol components at BRW and SGP. The variability in model chemistry for BRW and SGP suggests that at least some (if not all) of

the models are simulating substantially different chemistry than is observed at those two sites.

While it is beyond the scope of this paper to do a detailed comparison of measured and modeled chemistry for all sites, some observations can be made. At SGP, Jefferson et al. (2017) note the importance of nitrate in determining $f$(RH), but many models do not include nitrate (see Table 2). From those models considering nitrate, only ATRAS, GEOS-Chem and TM5 show a marked annual cycle in nitrate, but only ATRAS simulates a $f$(RH) annual cycle at SGP which could just as easily be related

to the OMF seasonal cycle as that of nitrate.

The models tend to simulate more consistent chemical composition at GRW. The temporal cycle of chemical constituents at GRW is dominated by sea salt (see Fig. S6) with the aerosol being almost entirely composed of sea salt in the winter months. This is consistent with observations of aerosol chemical composition in the region (Pio et al., 2007) and suggests perhaps wind-driven sea salt emissions are better parameterized than other aerosol species. Despite the similar estimates of chemical

composition among the models at GRW, Fig. 4 shows that some models (TM5, CAM and CAM-Oslo) simulate significantly higher $f$(RH=85 % / RH$_{\text{ref}}$=40 %) at GRW throughout the year. Because the chemistry simulated is generally consistent across the models and because models assume very similar hygroscopic growth for sea salt at high RH (Table 3), some other factor is causing these three models to be biased high. One possibility, which was alluded to previously, is how water uptake is modeled at low RH. Figure S8 shows that the models that exhibit the least growth between 0 % and 40 % RH are the models that

simulate the highest $f$(RH) in Fig. 4. In the next section we explore this for the specific case of sea salt hygroscopicity.

### 4.3 Graciosa (GRW) as a test case for modeled sea salt hygroscopicity

The unique characteristics of individual sites can be helpful to understand some features of the models. In this section we focus on the marine site GRW because all models simulate that the aerosol consists almost entirely of sea salt during winter months (see Fig. S6 in the supplementary material). Figure 5 presents $f$(RH) with $RH_{ref}$=0 % as a function of RH for the models for

cases when the models simulated a sea salt mass fraction larger than 95 %. Here the model values at additional specified RH values (RH=55, 65, and 75 %) are included when available. The figure also shows the observational data and theoretical curves for inorganic sea salt (Zieger et al., 2010) and NaCl. The theoretical curves were calculated using Mie theory (as described in Zieger et al., 2013) and the revised hygroscopic growth factors of inorganic sea salt and NaCl determined by Zieger et al. (2017). The particle size distribution needed for the Mie calculations was taken from Salter et al. (2015) for inorganic sea salt

with a water temperature of 20°C.

From Fig. 5 (left panel), it can be seen that five models (GEOS-Chem, OsloCTM3, TM5, IFS-AER and SALSA) assume that sea salt has the same hygroscopic growth as NaCl. In particular, at low RH, TM5 reproduces the theoretical NaCl behavior, with no hygroscopic growth up to RH=45 %. GEOS-Chem, IFS-AER and SALSA simulate some hygroscopic growth at RH=40 %, probably due to extrapolation of the hygroscopic growth below 40 %. Above 40 % RH, GEOS-Chem, TM5, and SALSA exhibit

the same curvature as the Mie model for NaCl on the upper part of the hysteresis loop. SALSA predicts slightly larger values for all relative humidities, which could point towards smaller model particle sizes (e.g., Zieger et al., 2013). This figure thus suggests that GEOS-Chem, IFS-AER and SALSA are most likely modeling sea salt as NaCl without assuming the aerosol to be solid at RH=40 %; this is in contrast to TM5 which assumes sea salt to be dry below 40 %. This explains one of the features seen in our previous results, namely, TM5 mostly overestimating Arctic and marine sites (Fig. 2). This is consistent with TM5

considering sea salt aerosol at 40 % RH to be fully crystallized (solid). A dry sea salt particle will be smaller and scatter less than the same particle with associated water. Thus the dry particle will exhibit a larger $f$(RH) because the denominator in Equation 1 will be smaller.

Zieger et al. (2017) have shown that inorganic sea salt exhibits different characteristics than NaCl. For inorganic sea salt, the expected value of $f$(RH=40 %) is around 1.2 for the lower branch (hydration curve) and around 1.7 for the upper branch

(dehydration curve, if efflorescence is not taken into account). With these values in mind, Fig. 5 (right panel) shows that CAM and CAM-Oslo (which are the only models implementing the hysteresis effect) exhibit values closer to the hydration curve, while ATRAS, MERRAero and GEOS-GOCART simulate values closer to the dehydration curve. In this hysteresis RH range, the model values for ATRAS, CAM, CAM-Oslo, MERRAero and GEOS-GOCART are always somewhere between the hydration and dehydration curves. At higher RH (e.g., RH=85 %) ATRAS exhibits a lower scattering enhancement factor

than is observed for inorganic sea salt, while CAM and CAM-Oslo show larger scattering enhancement factors than observed. MERRAero and GEOS-GOCART are the models that best match observed sea salt scattering enhancement. Moreover, CAM-Oslo shows the sharpest increase between RH=75-85 %, due to the fact that the hygroscopicity (and thus also $g$(RH)) has a discontinuous increase, which follows from this model's implementation of the hysteresis effect.

These results can be evaluated in the context of the hygroscopic growth factors the models assume for sea salt given in Table 3. The expected growth factor for NaCl at RH=90 % should range between g(90 %)=2.29-2.4. This is consistent with the factors used in GEOS-Chem, OsloCTM3, IFS-AER, and SALSA. GEOS-GOCART and MERRAero assume the lowest growth factor for sea salt at RH=90 % (1.9-2.17), which is consistent with the curves observed in Fig. 5, which are close to the theoretical curves for inorganic sea salt. Finally, the three CAM-family models assume g(RH=90 %)=2.25-2.28, values between those of inorganic sea salt (2.11) and NaCl (2.29-2.4). In accordance with this, CAM and CAM-Oslo simulate curves between those expected for inorganic sea salt and NaCl, while ATRAS exhibits slightly lower values than the inorganic sea salt curve.

## 4.4 The importance of defining the dry reference RH

The previous section has shown the importance of growth assumptions at low RH specifically for a deliquescent sea salt dominated aerosol. What happens at low RH is also important in considering $f(\text{RH})$ for other aerosol types and for model/measurement comparisons. Here, we consider the importance of defining the dry reference RH in general.

Based on recommendations from WMO/GAW (WMO/GAW, 2016), experimentalists try to maintain sampling conditions for 'dry' aerosol optical properties at RH<40 % and, as a first approximation, consider RH values below 40 % to be 'dry'. Measuring at dry conditions enables a comparison of aerosol properties across locations while minimizing the confounding effects of water. Making measurements at low RH is not without issues. Changing the conditions of the aerosol from ambient to RH<40 % can potentially result in the loss of volatile species such as nitrate and some organics (Bergin et al., 1997). Further, depending on the site environment, it can be difficult to maintain the sample conditions such that $\text{RH}_{\text{ref}}$<40 % (see Fig. S2 in the supplementary material). In fact, seasonal changes in ambient temperature and ambient RH can be reflected in the resulting measurement RH.

Complicating the picture is that some types of aerosol particles (e.g., sea salt, sulfuric acid or organic aerosol) will take up water at RH values below 40 %. Figure S9 provides a selection of the scattering enhancement as a function of RH for five sites covering multiple airmass types in Europe (based on Fig. 5 from Zieger et al., 2013). At all of these sites the $\sigma_{\text{sp}}(\text{RH}_{\text{dry}})$ was maintained at RH<30 % and often less than 20 %. These curves, obtained using tandem nephelometer humidograph measurements demonstrate that as RH increases, $f(\text{RH})$ has a tendency to also increase for almost all airmass types depicted. This is true even below RH=40 %. Further, the plots show that $f(\text{RH})$ depends on aerosol type, with cleaner and/or maritime air masses typically exhibiting higher enhancements than more polluted air masses. The magnitude of the enhancement at relatively low RH can be significant, for example, the humidogram for a non-sea salt event measured in the Arctic (see blue curve in Fig. S9 marked by an arrow) shows that particle light scattering increases by approximately 25 % due to water uptake at $\text{RH}_{\text{ref}} = 40$ % relative to dry scattering. For the sea salt event at the same site (dark blue line with markers), the hygroscopic growth is lower, but still observable. The water uptake at low RH even by pure inorganic sea salt has been confirmed by several independent methods (see Fig. 5).

When modelers are asked to provide simulations of aerosol optical properties at dry conditions, they typically will provide output at RH=0%. Depending on model assumptions about aerosol hygroscopicity and the types of aerosol particles studied, this can create large discrepancies between modeled and measured estimates of aerosol hygroscopicity. While the discussion of

Sections 4.1 and 4.2 focused on comparisons with model simulations at RH=40 % and measurements with $RH_{ref}$ extrapolated to 40 %, Section 4.3 shows that models exhibit differences between 0 and 40 % RH for the specific case of sea salt aerosol. Thus, it is useful/instructive to evaluate the impact of comparing the choice of $RH_{ref}$=0 % with that of $RH_{ref}$=40 %.

Figure 6 demonstrates the impact of the choice of $RH_{ref}$ on the comparison of observations and models. The figure shows the probability distribution function of the ratio between the modeled and measured $f(RH)$, for each model for two $RH_{ref}$ conditions. Each distribution takes into account all sites and the full periods of measurements, calculating the ratios between the model monthly median values of $f(RH)$ and the monthly median $f(RH)$ values for each site. A ratio larger than one appears for those models that tend to overestimate measurements.

The blue distributions in Fig. 6, which are for reference $RH_{ref}$=40 %, summarize the data that have been shown in sections 4.1 and 4.2. For most models, the peak of the blue curve is near, but above 1, indicating relatively good agreement between models and measurements, albeit with a slight bias toward higher hygroscopicity than is observed. The high variability in simulated $f(RH)$ observed for TM5 and ATRAS is reflected in the width of the histograms for those two models, while the low variability for some other models is indicated by narrow histograms.

The gray distribution in Fig. 6 represents the $f(RH)$ model-measurement ratio where $RH_{ref}$=0 % (for the model) and $RH_{ref}$ is at dry conditions (for the measurements), meaning measurement $RH_{ref}$ can be any value below 40 % - whatever the actual measurement condition was (see Fig. S2). Model overestimation is found to be larger when $RH_{ref}$ is set to 0 % for the GEOS-family models (GEOS-Chem, GEOS-GOCART, MERRAero), IFS-AER and SALSA and, to a lesser extent, for ATRAS and CAM-OSLO. A recent study by Latimer and Martin (2019) show a positive bias in the GEOS-Chem model for the GADS hygroscopicity paramterization which appears to be more significant at low (RH<35 %) conditions. This finding is consistent with the results shown in Fig. 6 for GEOS-Chem model.

The ratio of the modeled $f(RH)$ to measured $f(RH)$ when $RH_{ref}$=0 % is 1.64, and it decreases to 1.15 when using $RH_{ref}$=40 %. The implication is that the models that exhibit such large differences between $RH_{ref}$=0 % and $RH_{ref}$=40 % conditions are simulating significant hygroscopic growth between 0-40 % RH. Such growth would often not be seen by the measurements because the measurements are rarely (if ever!) that dry. In contrast, CAM and TM5 exhibit very little difference in their $f(RH=0\%)$ and $f(RH=40\%)$ histograms. This suggests these two models assume little growth below RH=40 % and this is seen in Fig. 5 for the specific case of sea salt. In particular, MAM in CAM model assumes that if RH<35 % the aerosol particles have fully crystallized (are in solid state) and have not taken up water.

The comparison presented in Fig. 6 highlights the differences in the model hygroscopicity parameterizations at the lower RH range (e.g. not fully dried particles and hysteresis effects). The discrepancy in $f(RH)$ for the two $RH_{ref}$ conditions presented in Fig. 6 is consistent with the hygroscopic growth simulated between RH=0 and 40 % (i.e., $f(RH=40\%/RH=0\%)$), shown in Fig. S10. This finding is further supported by the minimal shift in the $f(RH)$ probability distribution function when the two $RH_{ref}$ values are considered (Fig. S11).

This difference between the comparison at $RH_{ref}$=0 % and $RH_{ref}$=40 % may also explain the results of Gliss et al. (2019). They performed model/measurement comparisons for both in situ scattering and aerosol optical depth (AOD). For their in situ scattering comparison, 'dry' scattering measurements were compared with model simulations reported at RH=0 %; they

found that the ensemble model value underestimated the observed scattering by 33 %. In contrast, for the AOD comparisons, which were at ambient conditions for both models and measurements, the ensemble model value underestimated only by approximately 20 % (10-33 % depending on the source of AOD data). Thus, Gliss et al. (2019)'s larger model underestimate for in situ scattering than AOD may be due, at least in part, to the disconnect between the model and measurement definition of 'dry', although obviously other factors may also play a role. The results from this study and Gliss et al. (2019) imply that models would need to simulate higher aerosol loads and surface concentrations (or higher mass extinction coefficients) along with a reduced $f(\mathrm{RH})$ to reduce the overall bias between models and measurements. This type of comparison demonstrates the usefulness of evaluating models against a variety of independent atmospheric observations - here it suggests further exploration of the role of hygroscopic growth across a range of RH values is warranted.

## 5 Conclusions

This works presents the first comprehensive model-measurement evaluation exercise for aerosol hygroscopicity and its effect on light scattering (22 sites, 10 Earth system models). Model simulations of the scattering enhancement factor $f(\mathrm{RH})$, for the year 2010 were compared to spatially collocated measurements. The models exhibited large variability and diversity in the simulated $f(\mathrm{RH})$, but tended to overestimate $f(\mathrm{RH})$ relative to the measurements when the reference relative humidity is $\mathrm{RH_{ref}}$=40 %. The mean ratio between modeled $f(\mathrm{RH})$ and measurements is 1.15. The GEOS-family models and IFS-AER tend to simulate a narrow range of $f(\mathrm{RH})$ relative to the other models. Hygroscopic growth factors for the different simulated chemical species vary among the models and we attribute the narrow range in $f(\mathrm{RH})$ to the high growth factors the GEOS-family models and IFS-AER assume for all species except dust, which limits the range of $f(\mathrm{RH})$ those models can simulate. The chemical composition simulated by each model was compared and exhibited both similarities and differences across the sites studied. The GEOS-family of models tend to simulate more dust at many sites than the other models. The simulated chemistry was used to compare the modeled relationship between organic mass fraction and $f(\mathrm{RH})$ with various results from observational field campaigns. Models which assumed little to no hygroscopic growth for organic aerosol were better able to reproduce the observed relationship than models which assumed high growth factors. It was possible to explain some of the variability in model $f(\mathrm{RH})$ at a marine site by comparing the simulated $f(\mathrm{RH})$ when models simulated an aerosol dominated by sea salt. Model assumptions about water uptake at low RH were a significant factor, but different assumptions about the hygroscopicity of sea salt also played a role. Some models assumed the hygroscopicity of sea salt could be represented by NaCl, while others assumed water uptake characteristics more similar to the observed hygroscopicity of inorganic sea salt. Overall, all models fail to capture the annual cycle of observed $f(\mathrm{RH})$ at three sites representing distinct regimes (Arctic, rural, and marine) when it was possible to also temporally collocate the observations. Temporal collocation did not appear to improve the comparison of model simulations and observations relative to the comparison with multi-year climatological values. The diversity of the models tended to be larger than the variability in the observed long-term climatology at these three sites. Agreement between models and measurements was strongly influenced by the choice of $\mathrm{RH_{ref}}$. Better agreement between observations and models is found when $\mathrm{RH_{ref}}$=40%. In addition, some models exhibited unexpectedly large differences in

$f$(RH) at low RH (i.e., modeled scattering enhancement was significantly different for $RH_{ref}$=0 % and $RH_{ref}$=40 %), pointing to the sensitivity in the model parameterization of hygroscopic growth at low RH (e.g., effects of particle hysteresis). This was explicitly demonstrated for the modeled sea salt component, but may also be relevant for other species which exhibit hysteresis. To address this for future evaluations, models and measurements should be compared at similar RH conditions. For example, models could calculate $f$(RH) at the same variable RH conditions as the measurements. This type of study will make the model-measurement comparison more challenging since the same RH conditions should be matched and measurement conditions can vary widely with site and season. Alternatively, if measurements could better control their reference RH, both keeping it below 40 % and maintaining a narrower distribution of $RH_{ref}$, there would be less uncertainty in the model/measurement comparisons. Caution must always be taken when changing the measurement conditions - semivolatile species may volatilize with decreasing RH, inducing a negative artifact. While such losses are known and characterized for some species such as ammonium nitrate, we are still far away from a quantitative understanding such effects for semi-volatile organic species.

Based on the results presented here there are several topics that should be explored. One is to evaluate whether the gamma fit parameter is a more robust indicator for model/measurement comparisons than $f$(RH). Doing so would require model and measurement scattering data over a range of RH conditions. Another avenue is related to the $f$(RH) dependence on both chemical composition of the particles and particle size. Measured chemistry and size data collocated with scattering enhancement measurements at the sites where that information is available could be used in future work to assess modeled simulations of these factors and their impact on modeled scattering enhancement. The diversity in simulated chemical composition at many of the sites suggests this should be pursued. Comparison of size distributions is more challenging due to the variety of methodologies used by the different models to represent aerosol size. Evaluating model size distributions with measurements is a step beyond that and would require integration of measurements from several instruments to get a complete size distribution covering the full range of aerosol sizes simulated by the models. Another challenging task on the measurement side is to measure the scattering at RH > 85 % (e.g., 90-100 %) where the steepest hygroscopic growth happens and where models introduce large diversity in $f$(RH) due to assumptions on sub-grid scale humidity fluctuations and cloud versus cloud-free conditions.

Finally, we recommend that models update their hygroscopic growth parameterization for sea salt by assigning a lower and more realistic hygroscopic growth factor rather than assuming sodium chloride to be representative of sea salt. Models should also, if possible, explicitly provide $f$(RH) at specified RH values for pure components (i.e. for the sulfate or organic components) separately, which can then be compared to theory and observations. In addition, to further evaluate the influence of mixing state and particle size, a new multi-model experiment with a common hygroscopicity scheme would be desirable (e.g., within AeroCom).

*Code and data availability.* The measurement data behind this study is already publicly available (see Burgos et al., 2019). The entire dataset, incl. the corresponding model data, and analysis code is available at the Bolin Data Centre (https://bolin.su.se/data/burgos-2020-esm).

*Author contributions.* M.B. performed the model-measurement evaluation. M.B., E.A., G.T., and P.Z. designed study and wrote the paper. A.B., H.B., V.B., G.C., A.K., H.K., A.L., H.M., G.M., C.R., N.S., T.N. and K.Z. designed and performed model calculations. L.A., U.B., A.J., J.S., J.S., E.W., G.T., and P.Z. provided measurement data. All authors read and commented on the manuscript.

*Competing interests.* The authors declare no competing interests.

*Acknowledgements.* This work was essentially supported by the Department of Energy (USA) under the project DE-SC0016541. The JFJ measurements and the work by P.Z., U.B. and E.W. were financially supported by the ESA project Climate Change Initiative Aerosol cci (ESRIN/Contract No. 4000101545/10/I-AM), the Swiss National Science Foundation (Advanced Postdoc.Mobility fellowship; Grant No. P300P2_147776), and by the EC-projects Global Earth Observation and Monitoring (GEOmon, contract 036677) and European Supersites for Atmospheric Atmospheric Aerosol Research (EUSAAR, contract 026140). We thank the China Meteorological Administration for their
continued support to Lin'an Atmospheric Background Station; National Scientific Foundation of China (41675129), National Key Project of Ministry of Science and Technology of the People's Republic of China (2016YFC0203305 & 2016YFC0203306), Basic Research Project of Chinese Academy of Meteorological of Sciences (2020KJ001) . It was also supported by the Innovation Team for Haze-fog Observation and Forecasts of MOST and CMA.

CAM5.3-Oslo model development and simulations for this study were supported by the Research Council of Norway (grant nos. 229771 and
285003), by Notur/NorStore (NN2345K and NS2345K), and by the Nordic Centre of Excellence eSTICC (grant no. 57001). We acknowledge the Academy of Finland Projects 317390 and 308292. The ECHAM-HAMMOZ model is developed by a consortium composed of ETH Zurich, Max Planck Institut für Meteorologie, Forschungszentrum Jülich, University of Oxford, the Finnish Meteorological Institute and the Leibniz Institute for Tropospheric Research, and managed by the Center for Climate Systems Modeling (C2SM) at ETH Zurich.

KZ was supported by the Office of Science of U.S. Department of Energy. KZ thanks Steve Ghan for the help and support on the CAM5
AeroCom submission.

HM acknowledges funding from the Ministry of Education, Culture, Sports, Science, and Technology and the Japan Society for the Promotion of Science (MEXT/JSPS) KAKENHI Grant Numbers JP17H04709, JP16H01770, JP19H04253, JP19H05699, and JP19KK0265, the MEXT Arctic Challenge for Sustainability (ArCS) projects, and the Environment Research and Technology Development Fund (2–1703) of Environmental Restoration and Conservation Agency.
We thank Mian Chin (NASA Goddard) and the AeroCom community for valuable discussions.

# 6 Tables

**Table 1.** General site information. The median $RH_{ref}$ refers to the relative humidity inside the (dry) reference nephelometer, while the temporal resolution refers to measured values of $f(RH)$. More details and references on the sites can be found in Burgos et al. (2019).

| Station ID | Station Name, Country | Latitude (°) | Longitude (°) | Site Type | Median $RH_{ref}$ (%) | Temporal Resolution (h) |
|---|---|---|---|---|---|---|
| BRW | North Slope of Alaska, USA | 71.3 | -156.6 | Arctic | 6.8 | 6 |
| ZEP | Zeppelin, Norway | 78.9 | 11.9 | Arctic | 11.6 | 6 |
| JFJ | Jungfraujoch, Switzerland | 46.6 | 8 | Mountain | 5.2 | 3 |
| CBG | Chebogue Point, Canada | 43.8 | -66.1 | Marine | 28.2 | 1 |
| GRW | Graciosa, Portugal | 39.1 | -28 | Marine | 28.5 | 1 |
| GSN | Gosan, S. Korea | 33.28 | 126.2 | Marine | 33.0 | 1 |
| MHD | Mace Head, Ireland | 53.3 | -9.9 | Marine | 26.4 | 3 |
| PVC | Cape Cod, USA | 42.1 | -70.2 | Marine | 24.0 | 1 |
| PYE | Point Reyes, USA | 38.1 | -123 | Marine | 28.9 | 1 |
| THD | Trinidad Head, USA | 41.1 | -124.2 | Marine | 28.8 | 1 |
| APP | Appalachian State, USA | 36.2 | -81.7 | Rural | 13.6 | 1 |
| CES | Cabauw, Netherlands | 52 | 4.9 | Rural | 13.3 | 3 |
| FKB | Black Forest, Germany | 48.5 | 8.4 | Rural | 21.5 | 1 |
| HLM | Holme Moss, UK | 53.5 | -1.9 | Rural | 27.6 | 1 |
| HYY | Hyytiälä, Finland | 61.9 | 24.3 | Rural | 28.2 | 3 |
| LAN | Lin'an, China | 30.3 | 119.7 | Rural | 12.2 | 1 |
| MEL | Melpitz, Germany | 51.4 | 12.9 | Rural | 10.7 | 3 |
| SGP | Southern Great Plains, USA | 36.6 | -97.5 | Rural | 18.3 | 1 |
| HFE | Shouxian, China | 32.6 | 116.8 | Urban | 22.4 | 1 |
| PGH | Nainitial, India | 29.4 | 79.5 | Urban | 30.4 | 1 |
| UGR | Granada, Spain | 37.2 | -3.6 | Urban | 15.9 | 1 |
| NIM | Niamey, Niger | 13.5 | 2.2 | Desert | 9.4 | 1 |

**Table 2.** Summary of main characteristics implemented by each model. Model main reference, meteorology, mean RH value (subgrid variability considered), mixing state (black carbon), and species (number of size bins). In Meteorology column: GMAO = Global Modeling and Assimilation Office. In Mixing State column: E = external, I = internal. In Species and size bins column: BC = black carbon, OA = organic aerosol, MSA = methane sulfonic acid.

| Model (temporal resolution) | Main Reference | Meteorology | Mean RH (subgrid variability) | Mixing State | Species (size bins) |
|---|---|---|---|---|---|
| ATRAS (1h) | Matsui (2017) | Nudged to MERRA | clear-sky (no) | I | sulfate, dust, sea salt, BC, and OA, nitrate, ammonium (12) |
| CAM (1h) | Liu et al. (2012b) | Nudged to ERA interim | clear-sky (no) | I | sulfate, dust, sea salt, BC, and OA. 3 modes: Aitken, accumulation and coarse |
| CAM-OSLO (1h) | Kirkevåg et al. (2018) | Nudged to ERA interim | grid (no) | I, E | sulfate, dust, sea salt, BC, and OA (distributed in 12 modes) |
| GEOS-Chem (1h) | Bey et al. (2001) | GEOS5 version of NASA GMAO | grid (no) | E | sulfate, nitrate, ammonium, BC, and OA (bulk-mass), dust (4), sea salts (2) |
| GEOS-GOCART (24h) | Chin et al. (2002) | MERRA reanalysis | grid (no) | E | sulfate, dust and sea salt (5), BC and OA (2) |
| MERRAero (3h) | Buchard et al. (2015) | MERRA reanalysis | grid (no) | E | sulfate, dust and sea salt (5), BC and OA (2) |
| OsloCTM3 (24h) | Lund et al. (2018) | Offline meteorology from IFS ECMWF | grid (no) | I for hydrophilic BC | sulfate, dust (8), sea salt (8), BC, primary and secondary OA, nitrate (2), and ammonium (2) |
| TM5 (1h) | van Noije et al. (2014) | Offline, ERA-Interim | clear-sky (no) | I,E | sulfate, dust, sea salt, BC, and OA (7), ammonium nitrate, and MSA |
| IFS-AER (3h) | Morcrette et al. (2009) | online, initial conditions NWP analysis | grid (no) | E | sulfate, dust (3), sea salt (3), OA and BC (2), nitrate and ammonium |
| SALSA (1h) | Kokkola et al. (2018) | Nudged to ERA interim | clear-sky (no) | E | sulfate, dust, sea salt, OA, and BC (10) |

**Table 3.** Summary of the hygroscopic growth parameterization used by each model and the hygroscopic growth factor ($g$(RH), defined as the wet divided by the dry particle diameter) values for the main chemical species at RH=90 %. For models which use the $\kappa$-Köhler parameterization (given in squared brackets), we state $g(RH)$ calculated using $g(RH) = (1 + \kappa * RH/(1 - RH))^{1/3}$ (see Petters and Kreidenweis, 2007, here ignoring the Kelvin effect) for comparison reasons. Note, all models use different size parameterizations which vary in particle size and resolution (see Table 2 and Sect. 3). Boxes with hyphen note that this component is not considered by the model. *Only for hydrophilic fraction. ** Either as nitrate or sulphate. ***Parameterizations by Vignati et al. (2004)

| Model | Hygroscopicity parameterization | SS | SO$_4$ | BC | OA POA | OA SOA | NO$_3$ | NH$_4$ | Dust |
|---|---|---|---|---|---|---|---|---|---|
| ATRAS | Based on $\kappa$-Köhler theory | 2.25 [$\kappa = 1.16$] | 1.87 [$\kappa = 0.61$] | 1.0 [$\kappa = 10^{-10}$] | 1.24 [$\kappa = 0.1$] | 1.24 [$\kappa = 0.1$] | 1.87 [$\kappa = 0.61$] | 1.87 [$\kappa = 0.67$] | 1.0 [$\kappa = 0.001$] |
| CAM | Based on $\kappa$-Köhler theory | 2.25 [$\kappa = 1.16$] | 1.77 [$\kappa = 0.507$] | 1.0 [$\kappa = 10^{-10}$] | 1.24 [$\kappa = 0.1$] | 1.31 [$\kappa = 0.14$] | - | - | 1.17 [$\kappa = 0.068$] |
| CAM-OSLO | Based on $\kappa$-Köhler theory | 2.28 [$\kappa = 1.2$] | 1.77-1.80 [$\kappa = 0.507 − 0.534$] | 1.00 [$\kappa = 5 * 10^{-7}$] | 1.31 [$\kappa = 0.14$] | 1.31 [$\kappa = 0.14$] | - | - | 1.17 [$\kappa = 0.069$] |
| GEOS-Chem | Modified GADS/OPAC | 2.38 | 1.64 | 1.4 | 1.64 | 1.64 | 1.64 | 1.64 | 1.0 |
| GEOS-GOCART | Modified GADS/OPAC | 1.90-2.17 | 1.8 | 1.4* | 1.6 | 1.6 | - | - | 1.0 |
| MERRAaero | Modified GADS/OPAC | 1.90-2.17 | 1.8 | 1.4 | 1.64 | - | - | - | 1.0 |
| OsloCTM3 | Own development (see Sect. 3.7) | 2.31-2.39 | 1.72 | 1.0 | 1.46 | 1.46 | 1.82 | ** | 1.0 |
| TM5 | Own development (see Sect. 3.8) | *** | *** | 1.0 | 1.0 | - | *** | *** | 1.0 |
| IFS-AER | Own development (see Sect. 3.9) | 2.36 | 1.73 | 1.0 | 1.64 | - | 1.7 | 1.73 | 1.0 |
| SALSA | Own development (see Sect. 3.10) | 2.4 [$\kappa = 1.46$] | 1.9 [$\kappa = 0.68$] | 1.0 [$\kappa = 0$] | 1.5 [$\kappa = 0.3$] | - | - | - | 1.0 [$\kappa = 0$] |

Note, that the sea salt (SS) component is often assumed to have the same hygroscopic growth as sodium chloride. However, it has been recently shown that pure inorganic sea salt has a 8-15% lower hygroscopic growth than sodium chloride (Zieger et al., 2017).

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

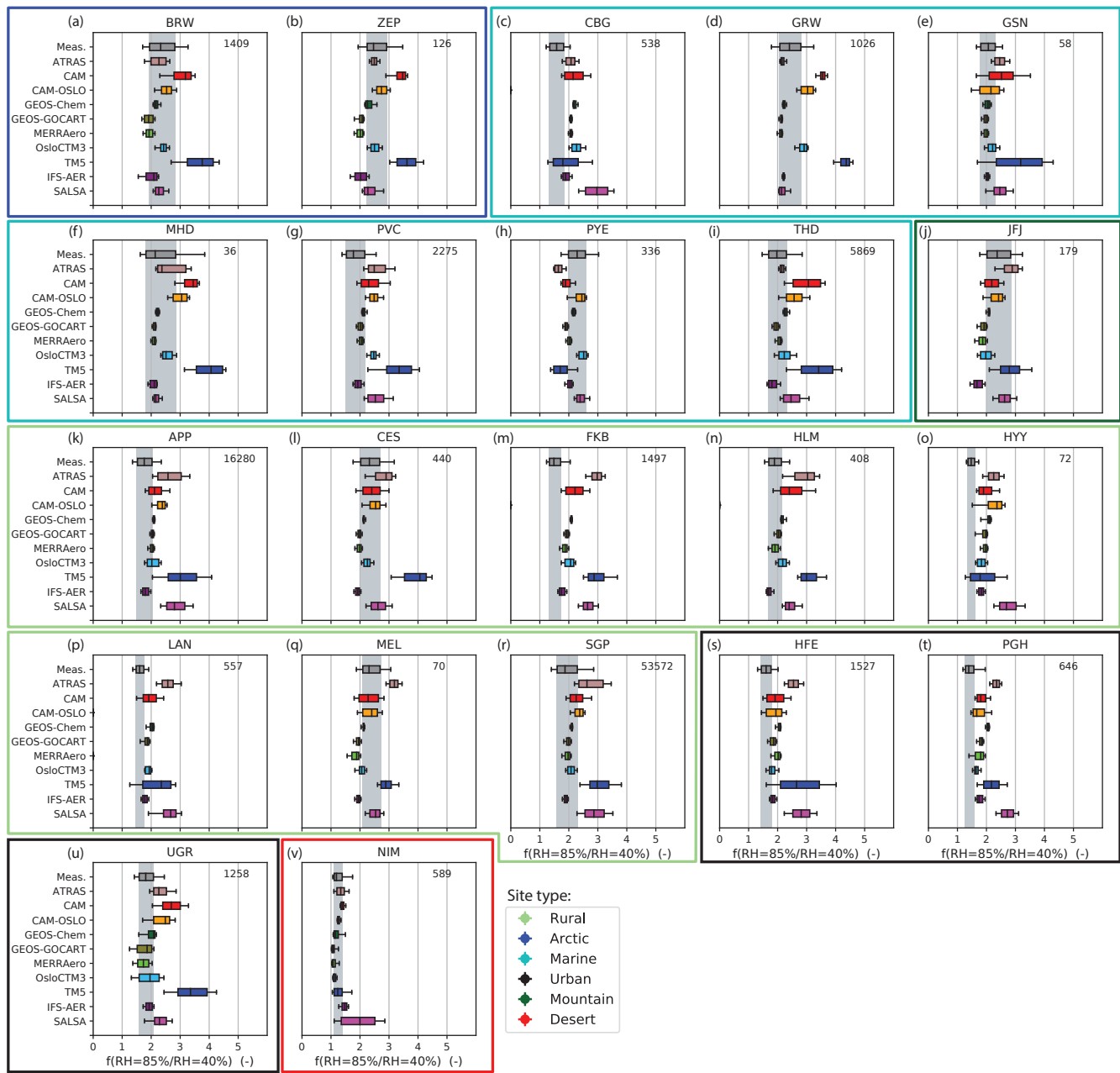

**Figure 1.** The scattering enhancement $f$(RH=85 %/RH$_{ref}$=40 %) at $\lambda = 550$ nm as measured and predicted by the various models for all investigated sites (panel **(a)** - **(v)**). The box edges represent the 25$^{th}$ to the 75$^{th}$ percentile (the gray underlying area represents the quartiles for all measurements), with the center line indicating the median. The whiskers show the range of the data extending from the percentiles 10th to 90th. The number in the top right corner indicates the number of available measurements at each site (temporal resolution shown in Table 1). The colored boxes grouping the different sets of plots indicate the site type.

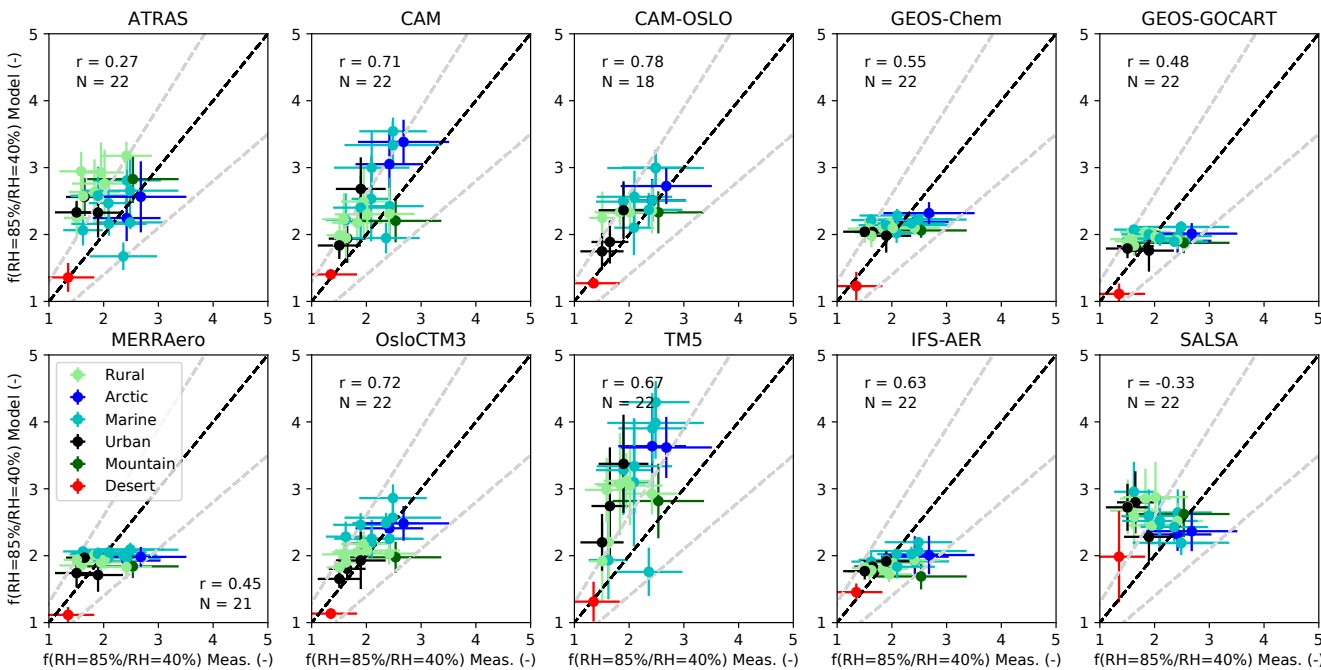

**Figure 2.** Simulated versus measured $f$(RH=85 %/RH$_{ref}$=40 %) at $\lambda = 550$ nm for each model color-coded by site type: blue for Arctic, cyan for marine, dark green for mountain, light green for rural, black for urban, and red for desert sites (panel **(a)** - **(j)**). The Pearson correlation coefficient (r) and the number of sites are indicated for each panel. The dashed black line shows the 1:1-line and gray dashed line shows the upper estimate of measurement uncertainties.

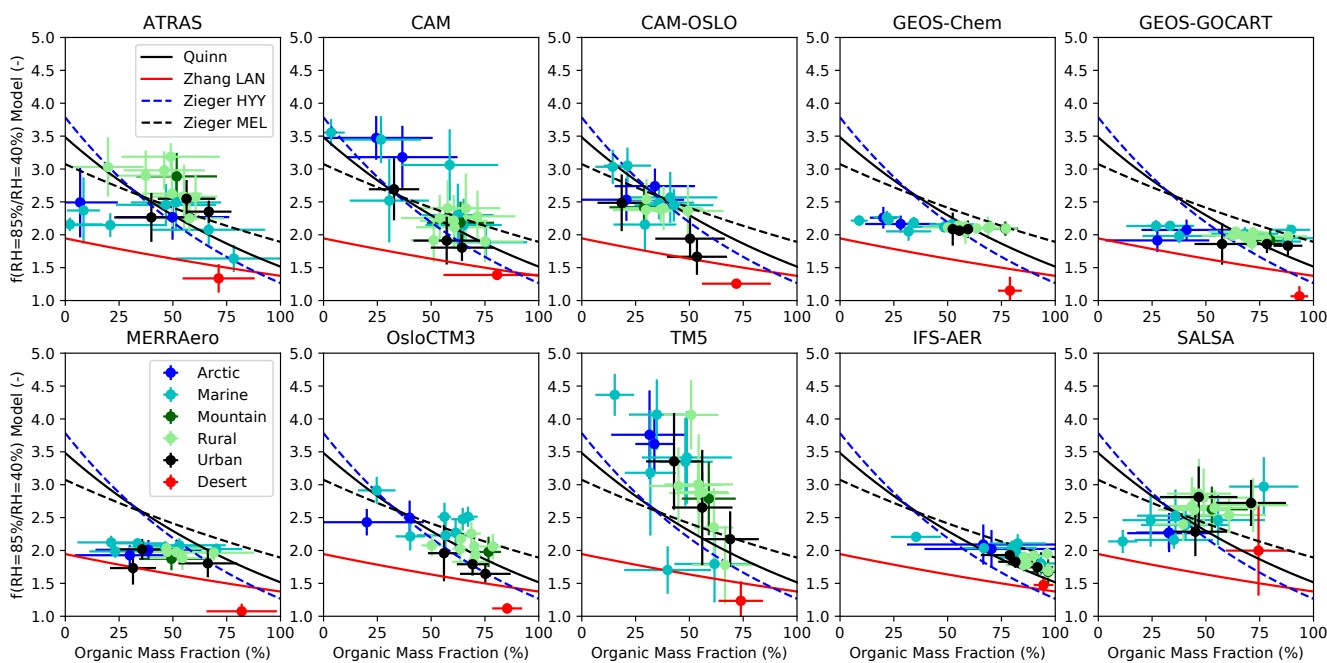

**Figure 3.** $f$(RH=85 %/RH$_{ref}$=40 %) vs. organic mass fraction for each model considered in this study. Each point represents one site, which are color-coded by site type. Parameterizations by Quinn et al. (2005), Zhang et al. (2015), and Zieger et al. (2015) represented by the solid and dotted lines.

.

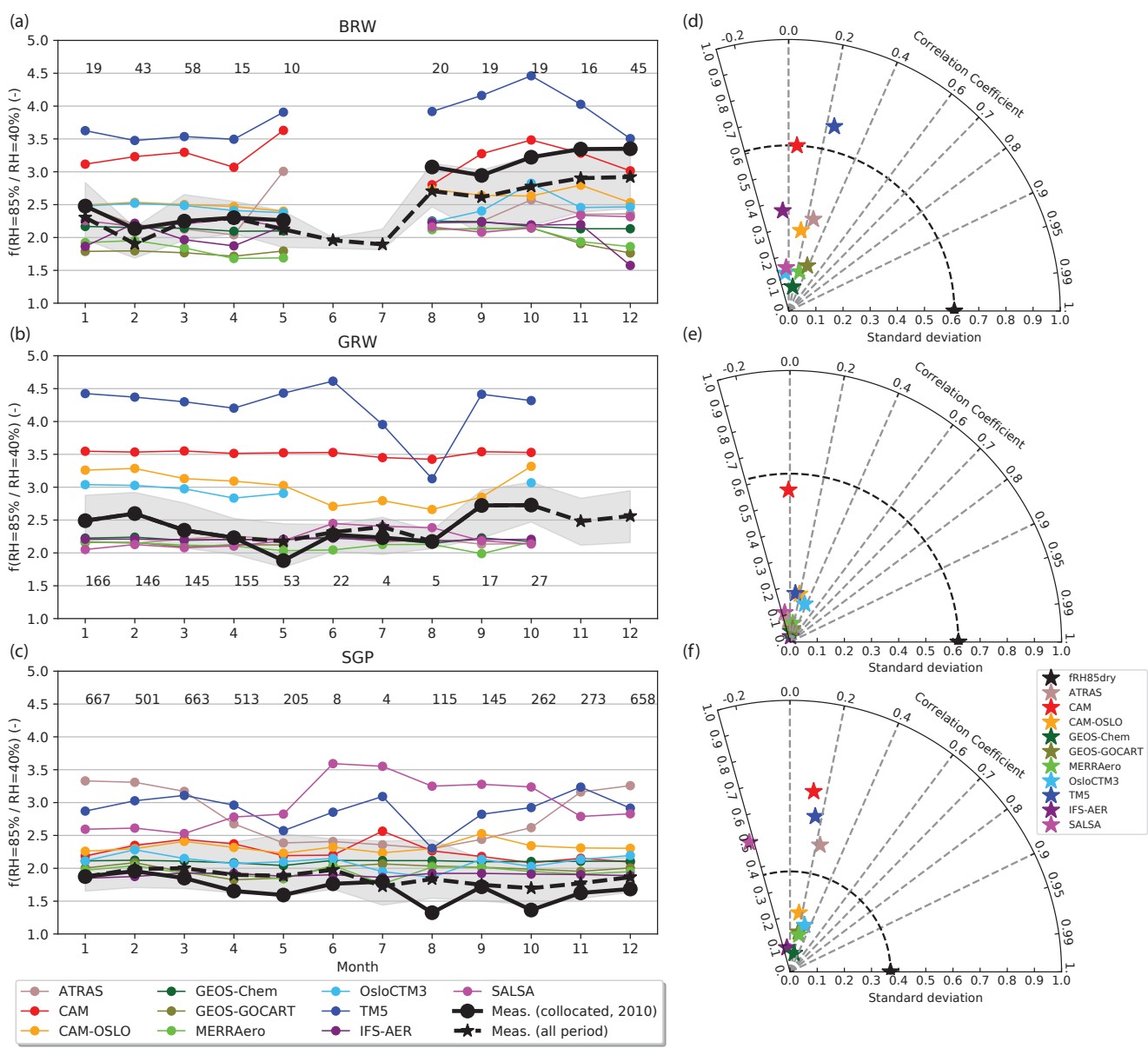

**Figure 4.** Comparison of $f$(RH=85 %/RH$_{ref}$=40 %) at $\lambda = 550$ nm for 2010: Barrow (Arctic site), Graciosa (marine site), and Southern Great Plains (rural site). **(a)-(c)** Annual cycles of the median $f$(RH=85 % /RH$_{ref}$=40 %) as measured (black line) and as predicted by the models (colored lines) collocated for 2010. The black dashed line and gray underlying area represent the median and range for the entire dataset. The numbers of data points in each month are also indicated. **(d)-(f)** Taylor diagrams showing the correlation coefficients and standard deviations of $f$(RH=85 % /RH$_{ref}$=40 %) for measurements (black symbols) and models (colored symbols, see legend).

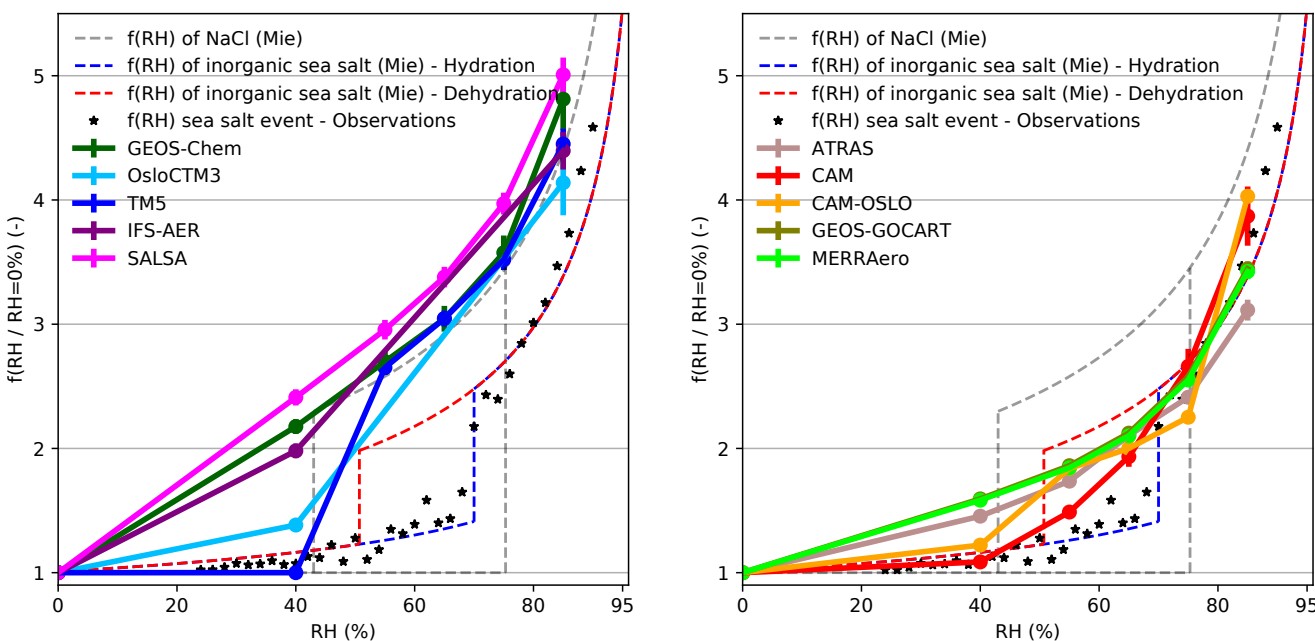

**Figure 5.** The scattering enhancement factor $f(\text{RH})$ vs. RH for sea salt dominated aerosol at Graciosa (GRW) as predicted by the different models (left panel: GEOS-Chem, OsloCTM3, TM5, IFS-AER, SALSA. Right panel: ATRAS, CAM, CAM-Oslo, GEOS-GOCART, and MERRAero). The model data is shown for cases when the predicted sea salt mass fractions was larger than 95 %. For comparison, the expected values for $f(\text{RH})$ of (i) NaCl determined by Mie modelling, (ii) for inorganic sea salt determined by Mie modeling based on H-TDMA sea salt chamber measurements of Zieger et al. (2017) are shown. The dashed blue and red lines show the corresponding hydration and dehydration line, respectively. Field measurements of $f(\text{RH})$ for pristine sea salt aerosol are shown as black stars (taken from Zieger et al., 2010).

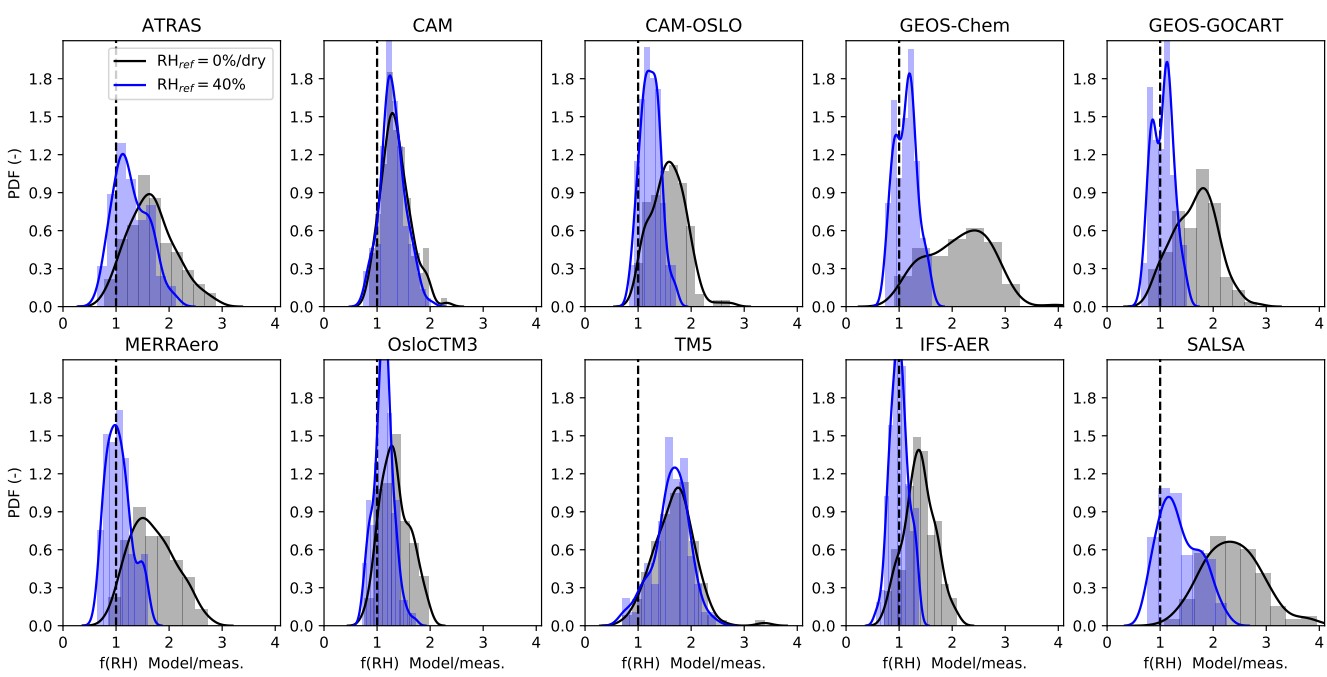

**Figure 6.** Probability density functions of the ratio $f(\mathrm{RH})_{\mathrm{model}}/f(\mathrm{RH})_{\mathrm{meas.}}$ for all sites for each model. The blue values denote the ratios if RH=40 % is taken as reference RH. The gray areas represent the ratio if $\mathrm{RH}_{\mathrm{ref}}$=0 % (models) or $\mathrm{RH}_{\mathrm{ref}}$=dry (measurements) is taken.

.

# A global model-measurement evaluation of particle light scattering coefficients at elevated relative humidity

María A. Burgos[1,2], Elisabeth Andrews[3], Gloria Titos[4], Angela Benedetti[5], Huisheng Bian[6,7], Virginie Buchard[6,8], Gabriele Curci[9,10], Zak Kipling[5], Alf Kirkevåg[11], Harri Kokkola[12], Anton Laakso[12], Julie Letertre-Danczak[5], Marianne T. Lund[13], Hitoshi Matsui[14], Gunnar Myhre[13], Cynthia Randles[6], Michael Schulz[11], Twan van Noije[15], Kai Zhang[16], Lucas Alados-Arboledas[4], Urs Baltensperger[17], Anne Jefferson[3], James Sherman[18], Junying Sun[19], Ernest Weingartner[17,20], and Paul Zieger[1,2]

[1]Department of Environmental Science and Analytical Chemistry, Stockholm University, Stockholm, Sweden
[2]Bolin Centre for Climate Research, Stockholm, Sweden
[3]Cooperative Institute for Research in Environmental Studies, University of Colorado, Boulder, USA
[4]Andalusian Institute for Earth System Research, University of Granada, Granada, Spain
[5]European Centre for Medium-Range Weather Forecasts, Reading, UK
[6]NASA/Goddard Space Flight Center, USA
[7]University of Maryland Baltimore County, Maryland, USA
[8]GESTAR/Universities Space Research Association, Columbia, USA
[9]Dipartimento di Scienze Fisiche e Chimiche, Universita' degli Studi dell'Aquila, L'Aquila, Italy
[10]Centre of Excellence CETEMPS, Università degli Studi dell'Aquila, L'Aquila, Italy
[11]Norwegian Meteorological Institute, Oslo, Norway
[12]Finnish Meteorological Institute, Kuopio, Finland
[13]Center for International Climate Research, Oslo, Norway
[14]Graduate School of Environmental Studies, Nagoya University, Nagoya, Japan
[15]Royal Netherlands Meteorological Institute, De Bilt, Netherlands
[16]Earth Systems Analysis and Modeling, Pacific Northwest National Laboratory, Richland, WA, USA
[17]Laboratory of Atmospheric Chemistry, Paul Scherrer Institute, Villigen, Switzerland
[18]Department of Physics and Astronomy, Appalachian State University, Boone, USA
[19]Key Laboratory of Atmospheric Chemistry of CMA, Chinese Academy of Meteorological Sciences, Beijing 100081, China
[20]Now at: Institute for Sensing and Electronics, University of Applied Sciences, Windisch, Switzerland

**Correspondence:** M.A. Burgos (maria.burgos@aces.su.se) and P. Zieger (paul.zieger@aces.su.se)

**Abstract.** The uptake of water by atmospheric aerosols has a pronounced effect on particle light scattering properties which in turn are strongly dependent on the ambient relative humidity (RH). Earth system models need to account for the aerosol water uptake and its influence on light scattering in order to properly capture the overall radiative effects of aerosols. Here we present a comprehensive model-measurement evaluation of the particle light scattering enhancement factor $f$(RH), defined as the particle light scattering coefficient at elevated RH (here set to 85 %) divided by its dry value. The comparison uses simulations from 10 Earth system models and a global dataset of surface-based in situ measurements. In general, we find a large diversity in the magnitude of predicted $f$(RH) amongst the different models which can not be explained by the site types. Based on our evaluation of sea salt scattering enhancement and simulated organic mass fraction, there is strong indication that differences in the model parameterizations of hygroscopicity ~~and perhaps mixing state~~ and model chemistry are driving at least some of the observed diversity in simulated $f$(RH). Additionally, a key point is that defining dry conditions is difficult from

an observational point of view and, depending on the aerosol, may influence the measured $f(\mathrm{RH})$. The definition of dry also impacts our model evaluation because several models exhibit significant water uptake between RH=0 % and 40 %. ~~An important finding is that the models show a significantly larger discrepancy with the observations if $\mathrm{RH_{ref}}$=0 % is chosen as the model reference RH compared to when $\mathrm{RH_{ref}}$=40 % is used.~~ The multi-site average ratio between model outputs and measurements is 1.64 ~~in the former case~~ when RH=0 % is assumed as the model dry RH and 1.16 ~~in the latter.~~ when RH=40 % is the model dry RH value. The overestimation by the models is believed to originate from the hygroscopicity parameterizations at the lower RH range which may not implement all phenomena taking place (i.e. not fully dried particles and hysteresis effects). This will be particularly relevant when a location is dominated by a deliquescent aerosol such as sea salt. Our results emphasize the need to consider the measurement conditions in such comparisons and recognize that measurements referred to as 'dry' may not be dry in model terms. Recommendations for future model-measurement evaluation and model improvements are provided.

# 1 Introduction

The effects of aerosol particles on the climate system are well known and appear as a consequence of the aerosol-radiation interaction (i.e., by scattering or absorption of solar radiation), and the aerosol-cloud interaction (when aerosols act as cloud condensation nuclei or ice nuclei and thereby change cloud microphysical and radiative properties; IPCC, 2013). ~~, and the rapid adjustments of aerosol-radiation interaction (when aerosols change the cloud cover by heating the atmosphere where clouds reside; see e.g., Bond et al., 2013).~~ Atmospheric aerosol particles are critical forcing agents in the climate system and, despite the increased number of studies in recent years, aerosol forcing remains (together with clouds) the largest uncertainty in climate change predictions (e.g., Ramanathan et al., 2001; IPCC, 2013; Regayre et al., 2018).

Aerosol optical properties, such as the wavelength-dependent light scattering coefficient, $\sigma_{\mathrm{sp}}(\lambda)$, are often measured under dry conditions (relative humidity (RH) below 40 %), as recommended by international protocols (e.g., WMO/GAW, 2016). However, aerosol particles can undergo hygroscopic growth and their optical properties are different at ambient conditions. The response of an aerosol particle to the surrounding RH is dependent on its size and solubility. Aerosol optical properties are thus dependent on RH: water uptake modifies particle size and chemical composition (and thus the complex refractive index) and this, in turn, affects the aerosol optical properties.

The scattering enhancement factor, $f(\mathrm{RH}, \lambda)$, is a key parameter that describes the change in particle light scattering coefficient $\sigma_{\mathrm{sp}}(\lambda)$ as a function of RH:

$$f(\mathrm{RH}, \lambda) = \frac{\sigma_{\mathrm{sp}}(\mathrm{RH}, \lambda)}{\sigma_{\mathrm{sp}}(\mathrm{RH_{dry}}, \lambda)}. \tag{1}$$

$f(\mathrm{RH}, \lambda)$ typically increases with increasing RH and is larger than 1 if particles do not experience significant restructuring when taking up water (Weingartner et al., 1995). The scattering enhancement factor is one way to represent aerosol hygroscopicity and its direct effect on particle light scattering (Titos et al., 2016).

There have been multiple measurement-based studies focused on investigating the scattering enhancement factor measured at different sites around the globe; Titos et al. (2016) compared $f(\mathrm{RH}, \lambda)$ at many of these as a function of dominant aerosol type. In general, they showed that clean marine aerosols exhibit higher $f(\mathrm{RH}, \lambda)$ than is measured at sites with anthropogenic influence, consistent with other studies (e.g., Wang et al., 2007; Fierz-Schmidhauser et al., 2010a; Zieger et al., 2013). In addition to assessing $f(\mathrm{RH}, \lambda)$ as a function of dominant aerosol type, more detailed investigations have also been done. Quinn et al. (2005) utilized co-located chemistry and $f(\mathrm{RH})$ measurements to develop a parameterization relating organic mass fraction and water uptake based on measurements at sites in Canada, the Maldives and South Korea. Zieger et al. (2010) analyzed aerosol water uptake using nephelometer measurements of wet and dry scattering coefficient, aerosol size distribution, and Mie theory at the Arctic site Ny-Ålesund. Svalbard. At Melpitz (a rural site in Germany), Zieger et al. (2014) found a correlation between the scattering enhancement factor and the aerosol chemical composition, in particular with the inorganic mass fraction. This linear relationship was extended for organic-dominated aerosol with observations from a boreal site in Finland (Zieger et al., 2015). Results from seven years of aerosol scattering hygroscopic growth measurements at the rural Southern Great Plains site in the USA indicated higher growth rates in the winter and spring seasons, which correlated with a high aerosol nitrate mass fraction (Jefferson et al., 2017). Burgos et al. (2019) created an open access database of scattering enhancement factors for 26 sites, covering a wide range of aerosol types whose optical properties were measured both long-term and as part of field campaigns.

An accurate estimation of aerosol effects on climate by Earth system models (ESMs) requires a realistic representation of aerosols (aerosol size distribution, mixing state, and composition).[1] Models must also be able to simulate processes in the aerosol life cycle such as primary emissions, new particle formation, coagulation, condensation, water uptake, and activation to form cloud droplets among others. Water uptake by aerosols affects not only their optical properties but also their life cycle by changing their size which can impact processes such as wet and dry deposition, transport, and ability to act as cloud condensation and ice nuclei (Covert et al., 1972; Pilinis et al., 1989; Ervens et al., 2007). Representing aerosol processes and properties in ESMs poses a great challenge due to the diversity and complexity of atmospheric aerosols. ESMs have implemented special modules and treatments for aerosols and the estimates of aerosol radiative forcing and climate impacts will be influenced by the uncertainties associated with the description of these processes. However, a compromise must be achieved between sufficiently representative aerosol and atmospheric process representations and the resultant computational cost (Ghan et al., 2012).

The effect of harmonized emissions on aerosol properties in global aerosol models was analyzed by Textor et al. (2007), who found that the aerosol representation is controlled, to a large extent, by processes other than the diversity in emissions. This implies that the harmonization of aerosol sources has only a small impact on the simulated inter-model diversity of the global aerosol burden and optical properties. Results are largely controlled by model-specific representation of transport, removal, chemistry and aerosol microphysics.

---

[1]Note that we are here using the more general term of Earth system model, while keeping in mind that other definitions (e.g. global climate models, general circulation models, transport models, etc.) are commonly used as well.

Previous model studies have suggested that water associated with aerosol particles can lead to significant differences amongst model predictions, estimates, and the assumptions about water uptake can have a noticeable effect. For example, Haywood et al. (2008) used tandem-humidifier nephelometer measurements from an aircraft to assess the parameterization of aerosol water uptake by the Met Office Unified Model. They found that ambient aerosols were simulated as being too hygroscopic relative to observations as a result of being modeled as composed solely of ammonium sulfate. Zhang et al. (2012) demonstrated that there are significant differences in simulated aerosol water content due to changes in a model's scheme to predict water uptake. Myhre et al. (2013) explored direct aerosol radiative forcing from a suite of models, showing that the primary source of differences among model predictions estimates of the mass extinction coefficient was aerosol hygroscopic growth of sulfate aerosols. Haywood et al., (2008) used tandem-humidifier nephelometer measurements from an aircraft to assess the parameterization of aerosol water uptake by the Met Office Unified Model. They found that ambient aerosols were simulated as being too hygroscopic relative to observations as a result of being modeled as composed solely of ammonium sulfate. Similarly, Reddington et al. (2019) studied the sensitivity of the aerosol optical depth (AOD) simulated by the GLOMAP model to assumptions about water uptake. They found that the AOD decreased when using the $\kappa$-Köhler (Petters and Kreidenweis, 2007) water uptake scheme relative to the AOD calculated using the Zdanovskii–Stokes–Robinson approach (Stokes and Robinson, 1966a). Moreover, Latimer and Martin (2019) also found that the implementation of the $\kappa$-Köhler hygroscopic growth for secondary inorganic and organic aerosols reduced the bias that appears in the representation of aerosol mass scattering efficiency relative to when water uptake was based on the Global Aerosol Data Set (GADS).

The Aerosol Comparison between Observations and Models (AeroCom) project (Textor et al., 2006; Schulz et al., 2006; Kinne et al., 2006, https://aerocom.met.no) aims to analyze global aerosol simulations to enhance understanding of aerosol particles and their impact on climate. In this project, intercomparisons among global aerosol models and comparisons with observations of aerosol properties have been carried out. These types of model evaluations allow for the identification of sources of model diversity and determination of which modeled aerosol properties need improvement. The objective of tier III of the INSITU measurement comparison experiment within AeroCom phase III (https://wiki.met.no/aerocom/phase3-experiments), is to assess how well model simulations represent observations of aerosol water uptake by comparing a high-quality, long-term, in situ measurements dataset with the output of several global aerosol models and that is what was done here.

In this paper, we present a comparison among scattering enhancement factors modeled by 10 different ESMs and observations. Our objectives are (i) to use measurements as a reality check on model simulations, (ii) to assess differences amongst model estimates of aerosol hygroscopic growth and then (iii) to suggest some potential reasons for any observed discrepancies, both between models and measurements and amongst models. This is the first comparison carried out for a wide suite of site types (covering Arctic, marine, mountain, rural, urban and desert stations) and ESMs, and is possible due to a newly published observational dataset of aerosol hygroscopicity (Burgos et al., 2019). A short description of the measurement dataset is presented in Sect. 2, while Sect. 3 gives a brief description of the models and the main references related to them. Section 4 shows the results of the model-measurement comparison for 22 sites and we evaluate the influence of different model choices about chemical species and mixing states on this comparison. We explore the importance of temporal collocation for three sample sites where temporal collocation is possible and use the unique chemical composition at one of these sites to interpret

## 2 Measurements

In this study, measured particle light scattering enhancement factors, $f(\mathrm{RH}, \lambda)$, from 22 different sites covering a wide range
of site types (Arctic, marine, rural, mountain, urban and desert) are used. Note that all results here will be shown for $\lambda$=550 nm; $\lambda$ will be omitted in the equations and variable names and only mentioned when necessary. Table 1 summarizes the station location and acronyms, while Fig. S1 (in supplementary material) shows a map with the location of these sites, color-coded by site type. The $f(\mathrm{RH})$ measurement data comes from the openly available scattering enhancement dataset described by Burgos et al. (2019). Four sites from Burgos et al. (2019) dataset were excluded in this current analysis, either because they
had a small upper size cut (PM$_1$ or PM$_{2.5}$, i.e., particulate matter with aerodynamic diameters less than 1 or 2.5 $\mu$m) or a very low number of data points ($N$<10). This scattering enhancement dataset was developed from dry and wet particle light scattering measurements made as part of field campaigns and long-term monitoring efforts by the USA Department of Energy Atmospheric Radiation Measurements (DoE/ARM), the USA National Oceanic and Atmospheric Administration Federated Aerosol Network (NOAA-FAN, Andrews et al., 2019), the Swiss Paul Scherrer Institute (PSI), and/or the Chinese Academy
of Meteorological Sciences (CAMS).

The scattering coefficients were measured simultaneously under two different conditions. First, under so-called dry or low-RH conditions (namely RH < 40 %), hereafter referred to as RH$_{\mathrm{ref}}$, and measured with a reference nephelometer or DryNeph. Typically RH$_{\mathrm{ref}}$ in the DryNeph will vary over the interval 0-40 % but this variation will depend on the characteristics of the site, e.g., at some marine sites like at GRW, the measurement system was not able to dry the aerosol below 50 % RH during
some months. Data with RH$_{\mathrm{ref}}$ > 40% were not included in this study. Figure S2 presents the probability density function of the measured RH$_{\mathrm{ref}}$ for all sites. Secondly, the scattering coefficients were measured scanning over a programmable range of RH values, mainly between 40 and 95 %, with a second humidified nephelometer or WetNeph (Sheridan et al., 2001; Fierz-Schmidhauser et al., 2010b). The RH in the WetNeph is termed RH$_{\mathrm{wet}}$. The wide range of scanned RH$_{\mathrm{wet}}$ values were typically achieved by passing the aerosol particles through a humidifier system before they entered the WetNeph. One possible limitation
of this approach is that the sample air may not equilibrate if the residence time in the elevated relative humidity downstream of the humidifier is too short (Sjogren et al., 2007). However, the measurements performed by PSI at the European sites JFJ, MHD, CES and MEL (see summary in Zieger et al., 2013) and HYY (Zieger et al., 2015) were all accompanied by optical closure studies using Mie theory together with measured size distribution and chemical composition and/or hygroscopic growth factors, which revealed no apparent bias due to too short residence times ~~inside~~ downstream of the humidifier.
In order to create a benchmark dataset for aerosol scattering enhancement, an identical process for data treatment was applied to all initial raw scattering coefficients, and data quality was assured by a thorough inspection of the scattering time series for each site (Burgos et al., 2019). The final dataset is composed of yearly files organized in three levels, containing scattering coefficients, hemispheric backscattering coefficients, and scattering enhancement factors for three wavelengths (450, 550, and

700 nm) and two particle size cuts (aerodynamic diameters lower than 10 and 1 $\mu$m). Level 1 contains the raw scattering data, Level 2 the corrected scattering coefficients and calculated scattering enhancement factors, and Level 3 contains the calculated $f(\text{RH=85 \%/ RH}_{\text{ref}})$. A detailed description of the data screening process and the corrections applied, the specific wavelengths and size cuts at each site, as well as the design and characteristics of the different instrument systems are given in Burgos et al.

(2019) and references therein. As part of the observational dataset development, uncertainty in $f(\text{RH})$ was also determined. The uncertainty in $f(\text{RH})$ depends on the aerosol load, RH and hygroscopic growth, and was found to vary between 10 and 30 % for PM$_{10}$. Table 4 in Burgos et al. (2019) presents a detailed description of the uncertainty as a function of these variables. One of the strengths of the dataset is that it was developed using a homogenized data treatment - differences in data processing was one of the issues cited in Titos et al. (2016) hygroscopicity overview paper that limited absolute comparisons of $f(\text{RH})$

values reported in the literature. The homogenized data treatment facilitates the intercomparison of the stations included in the dataset as well as the comparison against global model output. A full description of the homogenization process is given in Burgos et al. (2019), and a summary of the process is presented here. The homogenization starts with the light scattering raw data provided by each site manager. Standard corrections are applied to all raw data in an identical manner, and in-depth data screening is carried out to identify data during invalid periods or system malfunctions. Several corrections are applied

to the valid data periods: angular truncation and illumination non-idealities, adjustment to standard temperature and pressure, particles losses, and a 10-minute moving average is applied to the dry scattering coefficient series (this step is specially relevant for pristine sites). Finally, the scattering enhancement factors are reported at common RH$_{\text{ref}}$ and RH$_{\text{wet}}$ which eliminates potential discrepancies among $f(\text{RH})$ values due to choice of RH (Titos et al., 2016), and allows direct comparison between sites. In this study, we use Level 2 $f(\text{RH=85 \%/RH}_{\text{ref}}\text{=40 \%})$ at $\lambda = 550$ nm data from 22 stations (those with PM$_{10}$ size cut

or whole-air measurements) (see Table 1 for information about the station names, IDs, and aerosol types). The dry value of particle light scattering coefficient used to retrieve the scattering enhancement factor can be a) measured with the DryNeph at any RH$_{\text{ref}}$<40 %, or b) extrapolated to exactly RH$_{\text{ref}}$=40 %. We first present the model-measurement comparison results using DryNeph RH values extrapolated to RH$_{\text{ref}}$=40 %. This is followed by a discussion on the implications of making different assumptions about the DryNeph RH value for both measurements and models.

In this study we utilise the scattering enhancement at RH$_{\text{wet}}$=85% to parameterize aerosol hygroscopicity. Choosing RH$_{\text{wet}}$=85% ensures that the reported $f(\text{RH})$ value represents the aerosol in the fully deliquesced state (upper branch of the hysteresis loop). Scattering enhancement at specified RH is a simple metric. There are other methods, of varying complexity, that may also be used to describe the aerosol scattering enhancement; Titos et al. (2016) presents a review of the various empirical parameterizations found in literature that have been used to describe the relationship of $f(\text{RH}, \lambda)$ and RH. The most common other

algorithm is the two-parameter, power law fit referred to as the $\gamma$-fit (Hänel and Zankl, 1979). While fitting over the whole range of RH observations can provide valuable additional information about hygroscopic growth (e.g., investigating the RH ceilings often assumed in models or as a means to identify deliquescence transitions (Zieger et al., 2010; Titos et al., 2014a)) that level of complexity was not desired in this initial model measurement comparison.

## 3 Models

In this section, we present the ten models used in this study. We first provide a brief description of their main characteristics and relevant references, where detailed information on each model's parameterizations/assumptions can be found. The models used are: Community Atmosphere Model version 5 (CAM5), Aerosol Two-dimensional bin module for foRmation and Aging Simulation (CAM-ATRAS), the CAM5.3-Oslo (CAM-OSLO) model, the Goddard Earth Observing System with the MERRA Aerosol Reanalysis (GEOS-MERRAero), the Georgia Institute of Technology-Goddard Global Ozone Chemistry Aerosol Radiation and Transport model (GEOS-GOCART), the GEOS-Chem (GEOS-Chem) model, the Tracer Model (TM5), the Oslo chemistry-transport model (OsloCTM3), the European Centre for Medium-Range Weather Forecasts - Integrated Forecasting System model (ECMWF-IFS) run in the Copernicus Atmosphere Monitoring Service configuration, and the global general circulation model ECHAM6 with the SALSA module (ECHAM6.3-SALSA2.0). For simplicity, we will refer to these models as: CAM, ATRAS, CAM-OSLO, GEOS-Chem, GEOS-GOCART, MERRAero, TM5, OsloCTM3, IFS-AER, and SALSA, respectively.

~~CAM5.3, CAM-ATRAS, and CAM-OSLO make use of the same general circulation model, the Community Atmosphere Model (CAM5.3). Three more models (GEOS-Chem, GEOS-GOCART and MERRAero) use the Goddard Earth Observing System assimilated meteorological fields.~~ Table 2 summarizes some of the most relevant characteristics of each model, such as ~~parameterizations of hygroscopic growth,~~ meteorology, mixing states, species and size bins. Table 3 summarizes the parameterization of hygroscopic growth for the chemical components in each model and provides the growth values $g$(RH) at 90% so that the model assumptions can be more readily compared. The model data used in this study were provided within the ~~AeroCom phase III experiments~~ tier III of the INSITU measurement comparison experiment of AeroCom phase III (https://wiki.met.no/aerocom/phase3-experiments) and are composed of aerosol absorption and extinction coefficients at RH = 0, 40, and 85 %. Models also provided the mass mixing ratios for the chemical constituents they simulated, which we use to assess the impact of composition on hygroscopicity. Model values of scattering coefficient were obtained by subtracting absorption coefficient from extinction coefficient. The models were run for the year 2010 and data at surface level from 22 locations (closest gridpoint to the observational data) have been extracted. Exact temporal collocation between measurements and models can only be achieved at three of the measurement sites (BRW, GRW, and SGP), which made measurements in 2010. The model output files provide data at either 1h, 3h, or daily resolution, while the measurement data is primarily at hourly resolution with some of the more pristine sites averaged to six-hourly resolution (see Tables 1 and 2 for details).

All models considered in this study take into account topography. However, a model's surface elevation for a given gridbox will represent an average of the topography within the given gridbox. Nonetheless, we have used the surface values provided by the models for all sites in this study. For sites located in complex terrain the model surface values may not be representative of the measurement site and this will be exacerbated by models with coarser resolution. For example, Schacht et al. (2019) noted that complex local terrain near ZEP may have impacted their modeling efforts. In this study there is one mountain site (JFJ) in the Swiss Alps with an altitude of 3580 m a.s.l. and seven more sites with elevations above 200 m a.s.l. (APP, FKB, HLM, NIM, PGH, UGR, and ZEP at 1100, 511, 525, 205, 1951, 680, and 475 m a.s.l., respectively). The remaining 14 stations are at

## 3.1 CAM5

CAM5.3 is one of the versions from the CAM family models used in this study. The run we work with provided data at surface level with a grid resolution of 1.9º latitude x 2.5º longitude, and at hourly frequency. CAM5.3 uses the modal aerosol module which provides a compromise between computational resources and a sufficiently accurate representation of aerosol size distribution and mixing states. However, depending on the selected number of modes and aerosol species in each mode, it can still incur differences among models. This model uses the version with three lognormal modes, MAM3, which is described

in detail in Liu et al. (2012b). As a brief description, MAM3 has Aitken, accumulation and coarse modes and it assumes that: a) primary carbon is internally mixed with secondary aerosol, b) coarse dust and sea salt modes are merged, c) fine dust and sea salt modes are similarly merged with the accumulation mode, and d) sulfate is partially neutralized by ammonium. Hygroscopicity is based on $\kappa$-Köhler theory (Ghan et al., 2001), and the values used for the different aerosol components are listed in Table S3 of Liu et al. (2012b).

To represent the meteorological field, the nudging technique (Newtonian relaxation) has been used, with horizontal winds nudged towards ERA-Interim reanalysis, following Zhang et al. (2014). The present day (year 2000) anthropogenic emissions are prescribed using CMIP5 emission data (IPCC, 2013). Natural wind-driven aerosol (dust and sea salt) emissions are calculated online. CAM5.3 accounts for the following important processes that influence aerosols: nucleation, coagulation, condensational growth, gas- and aqueous-phase chemistry, emissions, dry deposition and gravitational settling, water uptake,

in-cloud and below-cloud scavenging, and production from evaporated cloud and rain droplets. Details on the representation of these processes can be found in the supplemental material of Liu et al. (2012a).

## 3.2 CAM-ATRAS

In this case, the CAM model is used but the aerosol module is changed to the Aerosol Two-dimensional bin module for foRmation and Aging Simulation (ATRAS). The run we work with provided data at surface level with the same grid resolution

(1.9º latitude x 2.5º longitude) as CAM5.3, and at hourly frequency. Meteorological nudging was used for temperature and wind fields in the free troposphere (<800 hPa) by using the MERRA2 (Modern-Era Retrospective Analysis for Research and Applications) data.

This model takes into account the following aerosol processes: primary aerosol emissions, gas- and aqueous-phase chemistry, nucleation, condensation and evaporation, secondary organic aerosols processes, dry and wet deposition, aerosol activation to

cloud droplets and water uptake. In this study, aerosol particles from 1 to 10 $\mu$m in dry diameter are represented with 12 size bins for sulfate, ammonium, nitrate, sea salt, dust, organic aerosol (OA), and black carbon (BC). The aerosol module as well as details and references for the aerosol processes treatment can be found in Matsui et al. (2014); Matsui (2017) and Matsui and

Mahowald (2017). Related to to water uptake, $\kappa$-Köhler theory is used with the hygroscopicity parameter $\kappa$ for each species given in Matsui (2017).

## 3.3 CAM-OSLO

In this case, the aerosol module OsloAero5.3 is applied in the atmosphere model CAM5.3 ~~model~~, which runs with a grid
resolution of 0.9º latitude x 1.25º longitude. A thorough description and general modelling and validation results from this aerosol module used in the atmospheric component CAM5.3-Oslo of the Norwegian Earth System Model (NorESM1.2) have been published by Kirkevåg et al. (2018).

For aerosols, the model represents sulfate, black carbon, primary and secondary organic aerosols, sea salt and mineral dust. The following processes are taken into account: nucleation, coagulation, condensational growth, gas- and aqueous-phase chemistry,
emissions, dry deposition and gravitational settling, water uptake, in-cloud and below-cloud scavenging, and cloud processing. Unlike (e.g.) MAM3, this aerosol module makes use of a "production tagged" method to calculate aerosol size and chemical composition. It describes a number of "background" log-normal modes that can change their size distribution due to condensation, coagulation, and cloud processing. A detailed offline size-resolving model carries out the corresponding aerosol micro-physical calculations, and a selection of results are stored in lookup tables. Hygroscopicity is estimated for each particle
size and type by ~~the volume mixing ratios~~ the use of the volume mixing rule for internal mixtures, adding (by condensation) water as a function of RH according to Köhler theory. In CAM-OSLO, optical parameters are found by interpolation in look-up tables at the actual RH in each grid-box and time. The model data is output at hourly frequency.

## 3.4 GEOS-Chem

GEOS-Chem is a community global three-dimensional Eulerian chemistry-model originally described in Bey et al. (2001) with
updates that are described in http://acmg.seas.harvard.edu/geos/geos_chem_narrative.html (last accessed 28 November 2019). Here we use version 10-01 of the model. GEOS-Chem is driven by assimilated meteorological observations from the Goddard Earth Observing System (GEOS) of the NASA Global Modeling and Assimilation Office (GMAO). For this work, we use the GEOS fields version 5.2.0 degraded from the native resolution to the 2° x 2.5° simulation grid and 47 levels, for computational expediency. For anthropogenic emissions we use EDGAR 4.2 complemented with regional inventories where available (US,
Canada, Mexico, Europe and East Asia).

The aerosol module employs a bulk mass approach for sulfate-nitrate-ammonium system and for BC and OA. Soil dust and sea salt are simulated with a sectional approach having four and two size bins, respectively. The aerosol optical properties are calculated from the simulated aerosol mass assuming log-normal size distribution with parameters taken from OPAC (Optical Properties of Aerosols and Clouds, Hess et al., 1998) and updated by Jaeglé et al. (2011) and Heald et al. (2014), adopting an
external mixing representation. The hygroscopic growth factors are taken from Chin et al. (2002).

## 3.5 GEOS-GOCART

The Goddard Chemistry, Aerosol, Radiation, and Transport module (GOCART) (Chin et al., 2002, 2009) was implemented in the NASA GEOS global Earth system model to simulate aerosol processes of sources, sinks, transport, and transformation (Colarco et al., 2010; Bian et al., 2013, 2017). For this study, the aerosol species included are sulfate, dust, organic aerosol (OA), BC, and sea salt. The model is "replayed" from the MERRA meteorological analyses at the same spatial resolution produced by the NASA Global Modeling and Assimilation Office (Rienecker et al., 2011). Every 6 h the model dynamical state (winds, pressure, temperature, and humidity) is set to the balanced state provided by MERRA and then a 6 h forecast is performed until the next analysis is available. The GEOS model is run with a grid resolution of 0.5º latitude x 0.625º longitude and with 72 vertical layers from surface up to 0.01 hPa (about 85 km). Aerosols are considered to have different degrees of hygroscopic growth with ambient RH (with the exception of dust). The hygroscopic growth follows the equilibrium parameterization of Gerber (1985) for sea salt and OPAC (Hess et al., 1998) for other aerosols.

## 3.6 GEOS-MERRAero

The GEOS Earth System Model is a weather- and climate-capable model which includes atmospheric circulation and composition, as well as oceanic and land components. This model includes the same aerosol transport module based on the GOCART (Chin et al., 2002; Colarco et al., 2010) that is used in the previously described GEOS-GOCART. The specific version of GEOS used in this study also includes assimilation of bias-corrected Aerosol Optical Depth (AOD) from the Moderate Resolution Imaging Spectroradiometer (MODIS) sensors. This is the so-called MERRAero aerosol reanalysis (Buchard et al. (2015)). Driven by the MERRA meteorology, MERRAero was run at a global 0.5 x 0.625 latitude-by-longitude horizontal resolution with 72 vertical layers and 3-hour frequency. The data assimilation step provides a direct observational constraint on the simulated 550 nm AOD, but absorption, speciation and vertical distribution remain largely driven by the background simulation. Optical properties of the aerosols are primarily based on Mie calculations using the particles properties as in Chin et al. (2002) and Colarco et al. (2010) with spectral refractive indices and hygroscopic growth parameterizations primarily from the OPAC database (Hess et al., 1998). The Gerber growth curve (Gerber, 1985) is used for sea salt.

## 3.7 OsloCTM3

OsloCTM3 is a chemistry-transport model, described in detail in Lund et al. (2018). The model includes several updates with regards to its predecessor, OsloCTM2, particularly in the convection, advection, proto-dissociation, and scavenging schemes. OsloCTM3 is a global three-dimensional transport model that is driven by 3h offline meteorological forecast data from IFS ECMWF and CEDS emissions as described in Hoesly et al. (2018). With respect to aerosols, it includes BC, primary and secondary organic aerosols, sulfate, nitrate, dust and sea salt and its aerosol module is inherited from OsloCTM2, with the main updates described in Søvde et al. (2012) and Lund et al. (2018). The hygroscopic growth for sulfate, nitrate and sea salt follows Fitzgerald (1975), and for organic aerosols from fossil fuel emissions and of secondary origin from Peng et al. (2001), and finally Magi and Hobbs (2003) for biomass burning aerosols, see further description in Myhre et al. (2007). The

parameterization from Fitzgerald (1975) on hygroscopic growth for inorganic aerosols has been shown to be very similar to using Köhler theory in OsloCTM3 (Myhre et al., 2004). The run used in this study has a grid resolution of 2.25º latitude x 2.25º longitude and daily frequency output was provided.

## 3.8 TM5

The Tracer Model 5 (TM5) is an atmospheric chemistry and transport model. The version used for this study is an update of the model described by van Noije et al. (2014). Essentially the same version was used to carry out the Tier I experiment of the INSITU project in 2016. For the study presented here, additional diagnostics were included in the model to assess the hygroscopic growth at varying relative humidity.

TM5 uses a regular grid with a horizontal resolution of 3º longitude x 2º latitude and 34 vertical levels. At high latitudes, the
number of grid cells in the zonal direction is gradually reduced towards the poles. Dry deposition velocities and the emissions of DMS, sea salt and mineral dust are calculated on a 1º x 1º surface grid, and subsequently coarsened to the atmospheric grid. The hygroscopic growth of the soluble modes follows the description in Vignati et al. (2004). For pure sulfate-water particles the water uptake is calculated using the parameterization from (Zeleznik, 1991). When sea salt is present in the soluble accumulation or coarse modes, the water uptake is calculated using the ZSR method (Stokes and Robinson, 1966b;
Zdanovskii, 1948). Below relative humidities of 45 %, sea salt is assumed to be dry. Additional water uptake in the presence of ammonium-nitrate in the soluble accumulation mode is calculated using EQSAM (Metzger et al., 2002). BC, OA and dust do not influence the water uptake. For calculating the aerosol optical properties at relative humidities other than ambient conditions, additional diagnostic calls to M7 and EQSAM have been included to calculate the water uptake in the relevant modes at these RH values. Apart from the water content, all other aerosol components are kept at their levels calculated at
ambient conditions.

## 3.9 IFS-AER

The European Centre for Medium-Range Weather Forecasts (ECMWF) Integrated Forecasting System (IFS), also used for numerical weather prediction, includes an optional aerosol module (AER). This is described in Morcrette et al. (2009), and an update regarding its parameterizations for aerosol sources, sinks and chemical production is provided in Rémy et al. (2019).
Successive versions of this model, including the aerosol module, are used operationally to produce global analyses and 5-day forecasts for the Copernicus Atmosphere Monitoring Service. The version used here, however, does not correspond precisely to any operational version, and is based on cycle 43r1 but with a number of experimental additions - most notably an early version of the nitrate and ammonium aerosol scheme that is described in Bozzo et al. (2019). The configuration corresponds closely to the ECMWF-IFS-CY43R1-NITRATE-DEV submission to the AeroCom Phase III 2016 control experiment. In this config-
uration, the model runs with a grid resolution of approximately 40km. The data files provided have 3h frequency.The species taken into account are sea salt, desert dust, hydrophilic and hydrophobic OM, and BC and sulfate, nitrate and ammonium. Hygroscopic growth follows the description of Bozzo et al. (2019) for sulfates, sea salt and organic aerosols. This includes the

parameterization of Tang (1997) for sea salt, and Tang and Munkelwitz (1994) for sulfates. The hygroscopic growth for nitrate and ammonium is described in Rémy et al. (2019) along with the rest of the nitrate/ammonium scheme.

## 3.10 SALSA

SALSA is the sectional aerosol module that has been coupled to the ECHAM-HAMMOZ aerosol-chemistry-climate model framework. The model version used in this study was ECHAM6.3-HAM2.3-MOZ1.0. The detailed description of SALSA along with the details of its implementation and evaluation against several types of observations have been presented by Kokkola et al. (2018). The SALSA module describes aerosol size distribution with 10 size classes in size space which include two parallel externally mixed size classes for insoluble and soluble aerosol, thus tracking 17 size classes covering dry diameters from 3 nm to 10 $\mu$m. It simulates all relevant atmospheric aerosol processes including aerosol-cloud interactions. Simulated compounds are sulfate, organic aerosols, BC, sea salt, dust and water. The hygroscopic growth in SALSA is calculated according to the Zdanovskii-Stokes-Robinson (ZSR) equation described in Stokes and Robinson (1966b) assuming that the soluble fraction of particles is always in liquid phase. Simulations were run with T63 spectral resolution (approx 1.9º latitude x 1.9º longitude), with 47 vertical levels and hourly output frequency.

## 3.11 Model main characteristics: hygroscopic growth, size distribution, chemical composition, and mixing state

In order to have a complete vision of the main traits of the models used in this study, we summarize here some of their characteristics and try to group them when possible to facilitate the analysis of the results in the following section. The aerosol size distribution, chemical species, mixing state and assumed hygroscopicity of each species are essential to predict the enhancement in aerosol light scattering. The mixing state, species and the number of size bins for each of the models are provided in Table 2, while Table 3 presents the details about the hygroscopic parameterization and coefficients used for each chemical constituent.

The models assign the chemical species to one or more size bins as described in Table 2. The size bins are typically assigned modal parameters to account for a range of particle sizes. To properly assess the impacts of the disparate approaches to size distribution for the different species would require synthesizing the size assumptions onto a common diameter grid (e.g., Mann et al., 2014). Such an approach is outside the scope of this paper and, therefore, we will not consider assumptions related to particle size in evaluating water uptake differences amongst models. Such an effort could be of value to explore in future work.

With regards to chemical constituents, all models consider five basic species: sulfate, dust, sea salt, BC, and OA. Five models also include nitrate and ammonium (ATRAS, CHEM, OsloCTM3, TM5, and IFS-AER). In addition, TM5 includes methane sulfonic acid (MSA). Figure S4 in the supplemental material shows that, for each species simulated by the models, there are both similarities and differences at the different sites. For example, for some sites (e.g., GRW, MHD, PGH and NIM) the modeled chemistry is quite consistent across models. In contrast, at coastal sites in North America (PYE, THD, PVC and CBG) the contribution of sea salt can be quite variable, possibly depending on where in each model's gridbox the site is located. The

GEOS family of models tend to simulate a larger contribution from dust at individual sites relative to other models - this is most obvious at the Arctic sites BRW and ZEP, but occurs at other sites as well.

In addition to differences in simulated chemistry, there are some differences in model assumptions about water uptake for the different species (see Table 3). The modeled hygroscopic growth in the ten models considered in this study can be either calculated by means of direct parameterization (e.g., GEOS-family models, OsloCTM3, TM5, and IFS-AER), methods based on different theories (e.g., $\kappa$-Köhler theory (Petters and Kreidenweis, 2007; Ghan et al., 2001) used by the CAM-family models, Zdanovskii-Stokes-Robinson (ZSR, Stokes and Robinson, 1966a) equation implemented in SALSA), or thermodynamic equilibrium models (e.g., EQSAM (Metzger et al., 2002) used by TM5 for nitrate). Some models provided hygroscopicity factors in terms of $g$(RH = 90 %) and others in terms of $\kappa$; the $\kappa$-values were converted to $g$(RH = 90 %) using $g(\mathrm{RH}) = (1 + \kappa * \mathrm{RH}/(1 - \mathrm{RH}))^{1/3}$ (see Petters and Kreidenweis, 2007, here ignoring the Kelvin effect). Note: $g$(RH) is analogous to $f$(RH), but represents the aerosol diameter enhancement due to water uptake instead of the scattering enhancement which is an optical property. A $g$(RH) value of 1.0 indicates no hygroscopicity/water uptake, while increasing values of $g$(RH) correspond to higher growth due to water uptake. The parameter $\kappa$ is an indicator of the water uptake for different chemical species.

All models assume similar hygroscopicity for sea salt, with $g$(RH) values ranging from 2.25-2.4, except MERRAero and GEOS-GOCART which utilize lower values (1.90-2.17 depending on the size bin). Sulfate hygroscopicity among models is quite homogeneous, with values ranging from 1.64-1.9. Black carbon is only considered to grow in the GEOS-family models. Organic aerosols are assumed to be non-hygroscopic or have low hygroscopicity except in the GEOS-family models and IFS-AER. Dust is assumed to be non-hygroscopic by most models, but CAM and CAM-Oslo consider $g$(RH) values of 1.17. The models that include nitrate and ammonium assume similar hygroscopicity for these two components, ranging from 1.64 to 1.87. In summary, then, one common trait of the three GEOS-family models is that they assign high hygroscopic values to all components, while the rest of the models assume black carbon, organics and/or dust will undergo little or no hygroscopic growth.

Previous studies have also evaluated the sensitivity of modeled aerosol optical properties to the mixing state assumptions. Curci et al. (2015) found significant differences in simulated ambient AOD between internally and externally mixed assumptions, while Reddington et al. (2019) found that simulated ambient AOD is relatively insensitive to mixing state assumptions, and suggested the bigger impact found by Curci et al. (2015) was due, mainly, to the different calculations of the aerosol number size distribution. Neither study specifically address the effect of the mixing state assumption on water uptake. The models used in this study utilize a variety of assumptions about mixing state as specified in Table 2.

# 4 Results

In this section we present the results showing the comparison between in situ measurements and the ten models described in the previous sections. We first provide a general comparison of scattering enhancement measured at 22 sites in the Burgos et al. (2019) dataset with model outputs. For this analysis, temporal collocation of model and measurement data is made on

a climatological basis. ~~Hourly model output for the simulation year 2010 will be selected only from those months where hourly measurement data is available (regardless of the year the measurements were made). In a second step,~~ Model output for the simulation year 2010 is selected only from those months where measurement data is available (regardless of the year the measurements were made). We included all model data for each month for a given site regardless of the number of measurement data points in that month and for that site. Analysis (not shown) requiring a constraint on the number of measurements in a month in order to include model simulations for that month suggested that our approach had minimal impact on the results. By selecting the entire month from the model dataset, the impact of interannual variability is minimized. An illustration of the possible impact of the difference between model and observational years can be found in the supplemental materials for the site SGP, which has the longest period of measurements (see Fig. S3). In Sect. 4.2 we perform a more detailed analysis for three sites that measured during 2010, and thus allow an exact temporal collocation with the models, collocating for day and month of the year 2010.

## 4.1 Comparison of modeled vs. measured $f$(RH)

Figure 1 shows the box and whisker plots of the particle light scattering enhancement factor $f$(RH=85 %/RH$_{ref}$=40 %), where the dry reference RH is taken at RH$_{ref}$=40 %, for both the measurements and models. Note that models CAM-OSLO and MERRAero have fewer extracted sites (18 and 21, respectively) than the available measurement stations. These models provided data extracted at site locations, rather than the full global simulation and four station locations (CBG, FKB, HLM, and LAN) were not requested from CAM-OSLO at the time of their model run and one (LAN) was not requested from MERRAero at the time of their run. The box edges represent the 25$^{th}$ and 75$^{th}$ percentiles, with a line for the median (50$^{th}$ percentile). The whiskers shows the range of the data expanding from the percentile 10$^{th}$ to the 90$^{th}$. The gray shaded area indicates the range of the 25$^{th}$ to 75$^{th}$ percentiles of the measurements and is plotted to facilitate comparison with the modeled values. This area represents the temporal variability over the time period of the $f$(RH) measurements for each site and does not include measurement error. The number of measurements for each individual site is provided in the top right corner of each plot. As noted above, the model statistics shown represent the same months as the measurements, but the measurement year may not match the model year. For example, MHD has measurements during January and February of 2009, so model data shown for MHD has been restricted to January and February for model year 2010. The sites are organized by site type: Arctic (BRW, ZEP), marine (CBG, GRW, GSN, MHD, PVC, PYE, THD), mountain (JFJ), rural (APP, CES, FKB, HLM, HYY, LAN, MEL, SGP), urban (HFE, PGH, UGR) and desert (NIM).

In general, the top 10 panels (Fig. 1 a-j), comprising the Arctic, marine and mountain sites, and the desert site (Fig. 1 v) tend to exhibit the best agreement among the models and the measurements (i.e., more models fall within the shaded area). These sites tend to be the furthest away from local sources and may be more representative of a larger area. Two sites (CBG and PVC) both on the north-eastern coast of North America (CBG is in Nova Scotia and PVC in coastal Massachusetts) are less well simulated; in both cases the models tend to simulate larger scattering enhancement than is observed. Titos et al. (2014a) showed that there were significant differences in $f$(RH) at PVC depending on whether the sample air was urban influenced or predominantly marine. The rural and urban sites (Fig. 1 k-u) tend to exhibit lower scattering enhancement than is simulated by

the models. In this second group, the sites CES and MEL are the exception, with most of the models falling in the shaded area, and ~~, for MEL,~~ occasionally below the shaded area.

Overall, high variability among the models is observed. The CAM-family models (ATRAS, CAM, and CAM-OSLO) exhibit differences among themselves and also, in general, large variability of $f$(RH) values within each model. In contrast, the three

GEOS models (GEOS-Chem, GEOS-GOCART and MERRAero), OsloCTM3, and IFS-AER exhibit similar predicted scattering enhancement values and quite narrow variability in $f$(RH) within each model. One possible explanation for the fact that GEOS-family models generally show lower median values of $f$(RH) could be that they simulate a larger relative contribution of dust to the aerosol load (see Fig. S4 in supplementary material) which is considered to be non hygroscopic. This could explain the results found at the Arctic sites as well as GSN, JFJ, APP, MEL, SGP and UGR. However, the GEOS-family models also

simulate lower $f$(RH) values for some other sites (e.g., GRW, MHD, PVC, THD and CES) where they don't simulate a large contribution from dust. Additionally, OsloCTM3 and IFS-AER do not simulate enhanced dust contributions so dust is unlikely to be the sole explanation. TM5 and SALSA exhibit the largest variability within their results, as can be seen at some rural (e.g., APP, CES, HYY, and SGP) and urban sites (HFE, PGH, and UGR). ~~IFS-AER, on the other hand, simulates very little variability in $f$(RH) for urban and rural sites and underestimates the $f$(RH) at the vast majority of sites.~~

In general, most of the models tend to overestimate $f$(RH) at almost all site types ~~,except for the IFS-AER model which shows a general underestimation.~~ There are several sites that most models ~~(except IFS-AER)~~ consistently overestimate, for example: CBG, APP, FKB, HYY, LAN, PGH and UGR. For some sites this may be due to complex topography and emissions sources that are not adequately captured by the models. For example, Granada (UGR) is surrounded by mountains and is impacted by desert dust from the Saharan desert and black carbon originating from local emissions (e.g., traffic and biomass burning,

Titos et al., 2017). Similarly, PGH is in the foothills of the Himalayan range and is influenced by local and transported aerosol plumes (Dumka et al., 2017), and LAN is a polluted background station representative of the Yangtze River Delta conditions, influenced by anthropogenic emissions and dust (Zhang et al., 2015). ~~For other sites, model overestimates may be due to other factors such as modeled chemistry or size distribution. It is beyond the scope of this paper to bring measurements of aerosol microphysical and chemical properties into the analysis, but that is a topic intended for future work.~~ However, there is no clear

pattern in the chemistry simulated at each site (e.g., Fig. S4 in supplemental materials) that would explain this overestimation. The data shown in Fig. 1 can be visualized in a different way in order to more readily see the relation between modeled and measured data for each model rather than for each site. Figure 2 shows the mean and standard deviations of the modeled versus measured $f$(RH=85 %/RH$_{\text{ref}}$=40 %) for each model, color-coded by site type. The one to one relationship is indicated by a solid black line and the gray dashed lines represent 30 % uncertainty bounds which is the maximum uncertainty of the

measurements as described in Burgos et al. (2019). ~~ATRAS,~~ The CAM-family models, TM5 and SALSA exhibit a tendency to overestimate $f$(RH) ~~, while IFS-AER tends to underestimate $f$(RH)..~~ The figure also shows a wide diversity between modeled and measured $f$(RH) for the different models. For example, the CAM-family models and TM5 exhibit a wider range in $f$(RH) relative to the GEOS-family models and IFS-AER, which exhibit very little range in $f$(RH). ~~The narrow range of $f$(RH) is also noticeable for the IFS-AER model but with a shift towards lower values (between 1.2 and 1.5), in accordance with the general~~

~~underestimation of this model as discussed above.~~

The other models mostly fall within the 30 % interval of (upper) measurement uncertainty estimate (Burgos et al., 2019). CAM, CAM-Oslo, and OsloCTM3 are the models that most accurately estimate $f$(RH) at all site types, with the simulated results falling closest to the 1:1 black line and being within the 30 % interval. The Pearson correlation coefficient is also shown in the left top corner of each panel. ~~The best correlations are found for OsloCTM3 and CAM ($r = 0.71$), followed by TM5 ($r = 0.65$).~~

~~The GEOS-family models have correlation coefficients close to 0.5, while IFS-AER and SALSA show negative correlation with the measurements.~~ The best correlations are found for CAM-Oslo, CAM, and OsloCTM3 with $r = 0.78$, 0.71, and 0.72, respectively. The GEOS-family models have correlation coefficients close to 0.5, while SALSA exhibits negative correlation with the measurements.

Previous studies (Burgos et al., 2019; Titos et al., 2016) found the largest values of $f$(RH) for Arctic and marine sites and
lowest for urban, desert and polluted sites. CAM and TM5 (and to a lesser extent CAM-OSLO) appear to replicate the observed pattern of the Arctic and marine sites having higher $f$(RH) than other sites. ATRAS and SALSA are similar in that they tend to simulate higher $f$(RH) values for marine, rural, and urban sites and lower for Arctic locations, with ATRAS predicting the highest hygroscopicity at rural sites. The GEOS-family and IFS-AER do not exhibit a large enough range in simulated $f$(RH) to determine if some site types are more hygroscopic than others.

It is useful to consider what causes the discrepancies between models and observations. Potential explanations for the model overestimates of $f$(RH) may be related to model assumptions about chemistry (e.g., the species included, hygroscopicity parameterizations for those species, assumptions about hysteresis, mixing state, etc.) or size distribution. We have already noted that it is beyond the scope of this paper to consider the impact of aerosol size distribution on scattering enhancement, but below we discuss hygroscopicity in relation to hysteresis, mixing state, hygroscopcity parameterization and chemical
composition. Table 3 summarizes the parameterizations used as well as the hygroscopic growth factors, g(RH), at RH=90% and $\kappa$ parameters so that the model assumptions of hygroscopic growth can be more directly compared.

A deliquescent aerosol can exist in the liquid and solid phases at the same RH, an effect known as hysteresis (Orr et al., 1958). This means that, below its deliquescence RH but above its efflorescence RH, the corresponding scattering will be different depending on whether it is in a liquid or dry state. Deliquescent aerosols are typically inorganic species such as ammonium
sulfate and sodium chloride. Modelling hysteresis is complex as the behavior differs for aerosols of mixed composition, relative to single component particles. The hysteresis effect is unlikely to be the cause of differences amongst the models as it has only been accounted for by two of the models considered in this study (CAM and CAM-Oslo). Moreover, $f$(RH) was calculated at RH=85 % to minimize discrepancies due to hysteresis because at that RH the particles will have undergone deliquescence. However, models may make different assumptions about water uptake at low RH which will affect $f$(RH) by impacting the
denominator of the scattering enhancement equation, which will be of importance of strongly deliquescent aerosol. This is explored in more detail in sections 4.3 and 4.4.

The mixing state is another model assumption that could play a role in the observed differences amongst models. Curci et al. (2015) reported that aerosol optical properties calculated from bulk aerosol models which assume external mixing may be inherently different from the optical properties calculated from more detailed microphysical models which assume internal
mixing. In contrast, Reddington et al. (2019) found modeled aerosol optical properties to be insensitive to mixing state and

suggested the differences described in the Curci et al. (2015) study were more related to assumptions about size distribution than mixing state. In this study, a commonality among the models exhibiting low variability in $f$(RH) (e.g., the GEOS-family models and IFS-AER), is that they assume an external mixing state (Table 2). SALSA, however, also assumes an externally mixed aerosol but does not exhibit the narrow range in $f$(RH) seen for the other models making this assumption. This suggests

that mixing state assumptions may not be the reason behind these differences, although we are unable to evaluate this further. The role played by the different parameterizations of aerosol water uptake has also been studied (Reddington et al., 2019; Latimer and Martin, 2019). Reddington et al. (2019) demonstrates that simulated AOD is sensitive to this assumption. Their results show that using the $\kappa$-Köhler theory to describe hygroscopicity decreases AOD significantly relative to the AOD simulated when the ZSR equation is used calculate aerosol water uptake. A comparison of SALSA (which uses ZSR) with the

CAM-family of models (which use $\kappa$-Köhler) in Fig. 1 does not reveal a consistent pattern; sometimes the $f$(RH) is higher for SALSA and sometimes for one or more of the CAM-family models. Since there are other differences amongst these models as well (e.g., simulated chemistry and size), it is impossible to assess the impact of these two different hygroscopicity parameterizations here. Latimer and Martin (2019) shows significant differences in mass scattering efficiency when $\kappa$-Köhler theory is used rather than GADS (Global Aerosol Dataset) to parameterize water uptake; they find that GADS results in an overestimate

of mass extinction efficiency relative to $\kappa$-Köhler. The GADS parameterization is discussed in more detail below.

As noted in Section 3, the hygroscopicity values are generally quite similar for sea salt, sulfate, and dust for all models. There are, however, large differences for BC, POA and SOA amongst the models. The GEOS-family of models assign significantly higher growth for these three species than assumed by the other models. This may, in fact, be the explanation for the narrow range of $f$(RH) exhibited by the GEOS-family of models - regardless of the simulated composition there will always be a large

amount of water uptake. In contrast, the other models can simulate a wider range of $f$(RH), i.e., from low to high $f$(RH), as the proportions of the chemical constituents shift.

The GEOS-family models all use GADS by Köpke et al. (1997) (or OPAC by Hess et al., 1998, which uses essentially the same values) to parameterize hygroscopicity. This simplified aerosol property model provides size and hygroscopic growth parameters of six components (for various size ranges) at selected RH values, where models often use linear interpolation.

Zieger et al. (2013) and, more recently, Latimer and Martin (2019) have shown that OPAC can be problematic for modeling hygroscopicity as it results in an overestimate of $f$(RH) at low and intermediate RH. ~~such an overestimate would not necessarily explain the small range in modeled $f$(RH) for the models using it. Another commonality among the GEOS-family models, and IFS-AER as well, is that they assume an external mixing state. Aerosol optical properties calculated from bulk aerosol models which assume external mixing may be inherently different from the optical properties calculated from more detailed~~

~~microphysical models which assume internal mixing. SALSA, however, also assumes an externally mixed aerosol but does not exhibit the narrow range in $f$(RH) seen for the other models making this assumption.~~ Our analysis suggests another implication of that overestimate - the inability to simulate the range of scattering enhancement factors observed by measurements.

~~As shown in Fig. 1, the Arctic and desert sites appear to be the most accurately simulated sites, as almost all predicted model values lie within the 30 % interval area. CAM, OsloCTM3 and TM5 (and to a lesser extent CAM-OSLO) appear to replicate~~

~~the observed pattern of the Arctic and marine sites having higher $f$(RH) than other sites, although CAM and TM5 both over-~~

~~estimate the observed Arctic and marine $f$(RH) values. ATRAS and SALSA are similar in that they tend to simulate higher $f$(RH) values for marine and rural sites and lower for Arctic locations, with ATRAS predicting the highest hygroscopicity at rural sites. The GEOS-family and IFS-AER do not exhibit a large enough range in simulated $f$(RH) to determine if some site types are more hygroscopic than others.~~

Our study provides the opportunity to challenge the models with a composition-based parameterization of $f$(RH) using the model-simulated chemistry to constrain model estimates of $f$(RH). Previous experimental field work has shown that aerosol hygroscopicity can be parameterized as a function of aerosol composition (Quinn et al., 2005; Zhang et al., 2015; Zieger et al., 2015) without any a priori assumptions about species-dependent water uptake. The simplest parameterization by Quinn et al. (2005) utilizes measured sulfate and organic aerosol mass concentrations to estimate organic mass fraction

(OMF=OA/(OA+sulfate)) and relates OMF to observations of $f$(RH) at three sites (CBG, KCO, GSN). They find that low OMF tends to result in high $f$(RH) and vice versa. More recent efforts (Zhang et al., 2015; Zieger et al., 2015) also applied the simple Quinn parameterization but determined that, for their sites, a more complete chemical characterization (i.e., considering more species) resulted in better correlation between observed chemical composition and $f$(RH). Here, we only compare with the simple Quinn parameterization because there is a disconnect between the measured species considered in the enhanced

parameterizations and the components simulated by the models. Figure 3 shows $f$(RH=85 %/RH$_{\text{ref}}$=40 %) as a function of the OMF simulated by each model. Each point represents one site, color-coded by site-type. Lines representing the relationship between OMF (as defined above) and $f$(RH) observed at different sites by Quinn et al. (2005), Zieger et al. (2015) and Zhang et al. (2015) are displayed as different lines on the figure. Note that the fit lines from Zieger et al. (2015) and Zhang et al. (2015) only represent their fits based on organics and sulfate rather than the relationships they developed using more detailed

chemistry.

Several things can be observed in Fig. 3. First, the models consistently simulate lower OMF values for marine and Arctic sites relative to those simulated for rural, urban, mountain and desert sites. However, those lower OMF values do not correspond to higher $f$(RH) for all models. The CAM-family models, OsloCTM3, and TM5 exhibit similar behavior to the Quinn et al. (2005) parameterization, with $f$(RH) inversely related the OMF. In contrast, the GEOS-family of models, and IFS-AER exhibit

no relationship between OMF and $f$(RH) and SALSA simulates a positive relationship (opposite to what is observed). The models that best reproduce the observed relationship between $f$(RH) and OMF are those that assume lower hygroscopicity for organics - this allows these models to simulate a wider range of $f$(RH) than if organic is assumed to have similar hygroscopicity characteristics as other considered species.

## 4.2   Investigating the importance of temporal collocation at BRW, GRW, and SGP

Temporal collocation of model data with observational data is an important aspect in model-measurement evaluation exercises (Schutgens et al., 2016). The model runs were conducted to simulate the year 2010 and three sites provide data covering almost that entire year. These sites exhibit distinct differences in their prevalent aerosol type: BRW, an Arctic site, GRW, a marine site, and SGP, a rural site. Temporal collocation has been carried out (Fig. 4) by selecting only those model data sampled at the same hour, day, and month (only day and month for OsloCTM3 and GEOS-GOCART models) with valid measurement data.

Because the focus in this section is to study the importance of temporal collocation, no threshold on number of data points within each month was required; the number of data points in each month are provided in Fig. 4 to give an indication of the representativeness (or lack thereof) of the monthly value.

Figure 4 shows, in the left column, the annual cycle (monthly medians) of the scattering enhancement factor for $f$(RH=85 % / RH$_{ref}$=40 %). The black lines represent the observations (solid line: year 2010 only, dashed line: all available measurements, gray area: interquartile range of all measurements), and the colored lines the ~~predictions~~ estimates by the different models. The observations from 2010 do not show obviously different characteristics compared to the climatology of the entire dataset for each site. The exceptions are for BRW in the latter half of the year where the 'all data' climatology is ∼12 % lower than the 2010 values, and for SGP where August and October exhibit monthly 2010 values lower than the climatological values (28 % and 20 %, respectively). In general, the variability in the measured monthly $f$(RH) is significantly narrower than the range of $f$(RH) simulated by the models, suggesting exact collocation in time will have a limited impact on the overall model-measurement comparison. Using all observational data allows extension of the comparison to additional months which were not covered in 2010. Figure S3 shows the annual cycle in $f$(RH) for each individual year of measurements at SGP, the site with the longest time coverage (1999-2016); just 3 out of 18 years exhibit deviations from the climatological values larger than 50 %, suggesting the climatological values are a reasonable proxy for comparison with model values.

Measurements at GRW and SGP do not exhibit a marked seasonal cycle in $f$(RH) , although the $f$(RH) observed at GRW exhibits slightly lower values during April, May, and June. ~~while~~ The seasonal cycle appears to be much larger for BRW, with larger values occurring in the second half of the year. ~~Most of the models (CAM, CAM-OSLO, GEOS-Chem, GEOS-GOCART, MERRAero, OsloCTM3, and IFS-AER) do not capture the observed monthly variations. SALSA exhibits monthly variations similar to measurements at both BRW and GRW, while TM5 performs best at capturing the monthly variations (but not the magnitude) at the three sites (seeFig.?? in supplemental materials). ATRAS shows pronounced variations in the annual cycle of f(RH), with particularly large values in January-February and November-December which are not observed in the measurements.~~ None of the models reproduce the observed annual cycle at BRW; some models (ATRAS, CAM-Oslo, GEOS-GOCART, GEOS-Chem, MERRAero, OsloCTM3, IFS-AER, and SALSA) are better in the early part of the year and fall within the observed interquartile range, while CAM is closer to the observations in the latter part of the year. TM5 exhibits a clear bias towards larger values at BRW. At GRW, only CAM-Oslo reproduces the slightly lower values observed in late spring and early summer, though it is biased towards larger values. TM5, again, shows the largest bias with respect to the measurements. The rest of the models agree better in terms of magnitude of $f$(RH), but do not track the observed seasonal cycle. At SGP, most models reproduce the lack of seasonal cycle suggested by the observations. Only ATRAS indicates a strong seasonal cycle which is not observed in these co-located measurements, although Jefferson et al. (2017) report a seasonal cycle for observed $f$(RH) at SGP similar to that simulated by ATRAS in shape but with a $f$(RH) narrower range. SALSA and TM5 both overestimate the observed $f$(RH). For SGP, the GEOS-family models, OsloCTM3 and IFS-AER fall within the observed interquartile range throughout the year.

This modeled seasonality (or lack thereof) is easier to quantify using Taylor diagrams as discussed below. To the right of each annual cycle plot in Fig. 4 there is a Taylor diagram (Taylor, 2001) showing the skill of the models for these three sites when

the model results are collocated both in time and space with the measurements. Taylor diagrams are used to provide a concise statistical summary on how well models match measurements in terms of standard deviation (represented by the radial distances from the origin to the points) and correlation coefficient (represented by the angle from the normal). Black symbols represent the in situ measurements and colored symbols represent the different models in our study. Note that standard deviation and correlation coefficient have been calculated from all the collocated instantaneous values. The correlation coefficients are quite low, suggesting that the models do not capture the monthly variability seen in the measurements. ~~The correlation coefficients are lower than 0.25 for GRW and SGP for all models. The highest correlation (r=0.38) is observed for BRW by the GEOS-GOCART model while other models exhibit less correlation with the BRW $f$(RH) observations.~~ The correlation coefficients are only larger than 0.3 for GEOS-GOCART (r=0.38 at BRW ), and OsloCTM3 (r=0.36, 0.3 at GRW and SGP, respectively). Negative correlation coefficients are also found for some models at the three sites. The models exhibit a fairly wide range of standard deviations, SD (between 0.1 and ∼0.7, depending on model and site), with ~~SD~~ values both above and below the SD observed for the measurements. The standard deviation is largest (>0.4) for CAM ~~and TM5~~ at the three sites and for TM5 at BRW and SGP. The Taylor diagrams suggest a lack of skill in the models at simulating the seasonality and variability of observed aerosol hygroscopicity even when the data are exactly temporally collocated.

Changes in both aerosol composition and size can cause changes in scattering enhancement (e.g., Zieger et al., 2010; Titos et al., 2014a). Such changes could be driven by annual circulation changes bringing different air masses to a site (Sherman et al., 2015) and/or by normal variability in sources over the year. Both direct measurements of aerosol size distribution and indirect proxies such as the scattering Ångström exponent suggest there are seasonal shifts in aerosol size at these three sites (e.g., Quinn et al., 2002; Marinescu et al., 2019; Pio et al., 2007). Similarly, aerosol composition shifts as a function of season have also been reported for these sites (e.g., Quinn et al., 2002; Parworth et al., 2015; Logan et al., 2014). An in-depth evaluation of observed and modeled seasonal composition cycles at the 22 sites considered in our study is outside the purview of this paper. However, we can look beyond the annual mass mixing ratio comparisons (Fig. S4, discussed in the previous section) to differences in the modeled monthly composition which may contribute to the variability in the modeled seasonal $f$(RH) shown in Fig. 4. Figures S5, S6 and S7 show the monthly variation in mass mixing ratio for the ten models considered in this study and for the year 2010 for these three sites. There is a fair amount of variability amongst the models in the simulated aerosol components at BRW and SGP. The variability in model chemistry for BRW and SGP suggests that at least some (if not all) of the models are simulating substantially different chemistry than is observed at those two sites.

While it is beyond the scope of this paper to do a detailed comparison of measured and modeled chemistry for all sites, some observations can be made. At SGP, Jefferson et al. (2017) note the importance of nitrate in determining $f$(RH), but many models do not include nitrate (see Table 2). From those models considering nitrate, only ATRAS, GEOS-Chem and TM5 show a marked annual cycle in nitrate, but only ATRAS simulates a $f$(RH) annual cycle at SGP which could just as easily be related to the OMF seasonal cycle as that of nitrate.

The models tend to simulate more consistent chemical composition at GRW. The temporal cycle of chemical constituents at GRW is dominated by sea salt (see Fig. S6) with the aerosol being almost entirely composed of sea salt in the winter months. This is consistent with observations of aerosol chemical composition in the region (Pio et al., 2007) and suggests perhaps

wind-driven sea salt emissions are better parameterized than other aerosol species. Despite the similar estimates of chemical composition among the models at GRW, Fig. 4 shows that some models (TM5, CAM and CAM-Oslo) simulate significantly higher $f$(RH=85 % / RH$_{\text{ref}}$=40 %) at GRW throughout the year. Because the chemistry simulated is generally consistent across the models and because models assume very similar hygroscopic growth for sea salt at high RH (Table 3), some other factor is

causing these three models to be biased high. One possibility, which was alluded to previously, is how water uptake is modeled at low RH. Figure S8 shows that the models that exhibit the least growth between 0 % and 40 % RH are the models that simulate the highest $f$(RH) in Fig. 4. In the next section we explore this for the specific case of sea salt hygroscopicity.

## 4.3    Graciosa (GRW) as a test case for modeled sea salt hygroscopicity

The unique characteristics of individual sites can be helpful to understand some features of the models. In this section we focus

on the marine site GRW because all models simulate that the aerosol consists almost entirely of sea salt during winter months (see Fig. S6 in the supplementary material). Figure 5 presents $f$(RH) with RH$_{\text{ref}}$=0 % as a function of RH for the models for cases when the models simulated a sea salt mass fraction larger than 95 %. Here the model values at additional specified RH values (RH=55, 65, and 75 %) are included when available. The figure also shows the observational data and theoretical curves for inorganic sea salt (Zieger et al., 2010) and NaCl. The theoretical curves were calculated using Mie theory (as described

in Zieger et al., 2013) and the revised hygroscopic growth factors of inorganic sea salt and NaCl determined by Zieger et al. (2017). The particle size distribution needed for the Mie calculations was taken from Salter et al. (2015) for inorganic sea salt with a water temperature of 20°C.

From Fig. 5 (left panel), it can be seen that five models (GEOS-Chem, OsloCTM3, TM5, IFS-AER and SALSA) assume that sea salt has the same hygroscopic growth as NaCl. In particular, at low RH, TM5 reproduces the theoretical NaCl behavior, with

no hygroscopic growth up to RH=45 %. GEOS-Chem, IFS-AER and SALSA simulate some hygroscopic growth at RH=40 %, probably due to extrapolation of the hygroscopic growth below 40 %. Above 40 % RH, GEOS-Chem, TM5, and SALSA exhibit the same curvature as the Mie model for NaCl on the upper part of the hysteresis loop. SALSA predicts slightly larger values for all relative humidities, which could point towards smaller model particle sizes (e.g., Zieger et al., 2013). This figure thus suggests that GEOS-Chem, IFS-AER and SALSA are most likely modeling sea salt as NaCl without assuming the aerosol to

be solid at RH=40 %; this is in contrast to TM5 which assumes sea salt to be dry below 40 %. This explains one of the features seen in our previous results, namely, TM5 mostly overestimating Arctic and marine sites (Fig. 2). This is consistent with TM5 considering sea salt aerosol at 40 % RH to be fully crystallized (solid). A dry sea salt particle will be smaller and scatter less than the same particle with associated water. Thus the dry particle will exhibit a larger $f$(RH) because the denominator in Equation 1 will be smaller.

Zieger et al. (2017) have shown that inorganic sea salt exhibits different characteristics than NaCl. For inorganic sea salt, the expected value of $f$(RH=40 %) is around 1.2 for the lower branch (hydration curve) and around 1.7 for the upper branch (dehydration curve, if efflorescence is not taken into account). With these values in mind, Fig. 5 (right panel) shows that CAM and CAM-Oslo (which are the only models implementing the hysteresis effect) exhibit values closer to the hydration curve, while ATRAS, MERRAero and GEOS-GOCART simulate values closer to the dehydration curve. In this hysteresis RH

range, the model values for ATRAS, CAM, CAM-Oslo, MERRAero and GEOS-GOCART are always somewhere between the hydration and dehydration curves. At higher RH (e.g., RH=85 %) ATRAS exhibits a lower scattering enhancement factor than is observed for inorganic sea salt, while CAM and CAM-Oslo show larger scattering enhancement factors than observed. MERRAero and GEOS-GOCART are the models that best match observed sea salt scattering enhancement. Moreover, CAM-Oslo shows the sharpest increase between RH=75-85 %, due to the fact that the hygroscopicity (and thus also $g(\text{RH})$) has a discontinuous increase, which follows from this model's implementation of the hysteresis effect.

These results can be evaluated in the context of the hygroscopic growth factors the models assume for sea salt given in Table 3. The expected growth factor for NaCl at RH=90 % should range between g(90 %)=2.29-2.4. This is consistent with the factors used in GEOS-Chem, OsloCTM3, IFS-AER, and SALSA. GEOS-GOCART and MERRAero assume the lowest growth factor for sea salt at RH=90 % (1.9-2.17), which is consistent with the curves observed in Fig. 5, which are close to the theoretical curves for inorganic sea salt. Finally, the three CAM-family models assume g(RH=90 %)=2.25-2.28, values between those of inorganic sea salt (2.11) and NaCl (2.29-2.4). In accordance with this, CAM and CAM-Oslo simulate curves between those expected for inorganic sea salt and NaCl, while ATRAS exhibits slightly lower values than the inorganic sea salt curve.

## 4.4 The importance of defining the dry reference RH

The previous section has shown the importance of growth assumptions at low RH specifically for a deliquescent sea salt dominated aerosol. What happens at low RH is also important in considering $f(\text{RH})$ for other aerosol types and for model/measurement comparisons. Here, we consider the importance of defining the dry reference RH in general.

Based on recommendations from WMO/GAW (WMO/GAW, 2016), experimentalists try to maintain sampling conditions for 'dry' aerosol optical properties at RH<40 % and, as a first approximation, consider RH values below 40 % to be 'dry'. Measurements at dry conditions enables a comparison of aerosol properties across locations while minimizing the confounding effects of water. Making measurements at low RH is not without issues. Changing the conditions of the aerosol from ambient to RH<40 % can potentially result in the loss of volatile species such as nitrate and some organics (Bergin et al., 1997). Further, depending on the site environment, it can be difficult to maintain the sample conditions such that $\text{RH}_{\text{ref}}$<40 % (see Fig. S2 in the supplementary material). In fact, seasonal changes in ambient temperature and ambient RH can be reflected in the resulting measurement RH.

Complicating the picture is that some types of aerosol particles (e.g., sea salt, sulfuric acid or organic aerosol) will take up water at RH values below 40 %. Figure S9 provides a selection of the scattering enhancement as a function of RH for five sites covering multiple airmass types in Europe (based on Fig. 5 from Zieger et al., 2013). At all of these sites the $\sigma_{\text{sp}}(\text{RH}_{\text{dry}})$ was maintained at RH<30 % and often less than 20 %. These curves, obtained using tandem nephelometer humidograph measurements demonstrate that as RH increases, $f(\text{RH})$ has a tendency to also increase for almost all airmass types depicted. This is true even below RH=40 %. Further, the plots show that $f(\text{RH})$ depends on aerosol type, with cleaner and/or maritime air masses typically exhibiting higher enhancements than more polluted air masses. The magnitude of the enhancement at relatively low RH can be significant, for example, the humidogram for a non-sea salt event measured in the Arctic (see blue curve in Fig. S9 marked by an arrow) shows that particle light scattering increases by approximately 25 % due to water uptake

at $RH_{ref} = 40\%$ relative to dry scattering. For the sea salt event at the same site (dark blue line with markers), the hygroscopic growth is lower, but still observable. The water uptake at low RH even by pure inorganic sea salt has been confirmed by several independent methods (see Fig. 5).

When modelers are asked to provide simulations of aerosol optical properties at dry conditions, they typically will provide output at RH=0%. Depending on model assumptions about aerosol hygroscopicity and the types of aerosol particles studied, this can create large discrepancies between modeled and measured estimates of aerosol hygroscopicity. While the ~~previous~~ discussion ~~has~~ of Sections 4.1 and 4.2 focused on comparisons with model simulations at RH=40% and measurements with $RH_{ref}$ extrapolated to 40%, Section 4.3 shows that models exhibit differences between 0 and 40 % RH for the specific case of sea salt aerosol. Thus, it is useful/instructive to evaluate the impact of comparing the choice of $RH_{ref}$=0% with that of $RH_{ref}$=40%.

Figure 6 demonstrates the impact of the choice of ~~reference RH (RH_{ref})~~ on the comparison of observations and models. The figure shows the probability distribution function of the ratio between the modeled and measured $f(RH)$, for each model for two $RH_{ref}$ conditions. Each distribution takes into account all sites and the full periods of measurements, calculating the ratios between the model monthly median values of $f(RH)$ and the monthly median $f(RH)$ values for each site. A ratio larger than one appears for those models that tend to overestimate measurements.

The blue distributions in Fig. 6, which are for reference $RH_{ref}$=40%, summarize the data that have been shown in ~~the previous two~~ sections 4.1 and 4.2. For most models, the peak of the blue curve is near, but above 1, indicating relatively good agreement between models and measurements, albeit with a slight bias toward higher hygroscopicity than is observed. ~~The IFS-AER curve maximum is below 1, as expected based on the earlier observations that the IFS-AER model tends to underestimate hygroscopicity.~~ The high variability in simulated $f(RH)$ observed for TM5 and ATRAS is reflected in the width of the histograms for those two models, while the low variability for some other models is indicated by narrow histograms.

The gray distribution in Fig. 6 represents the $f(RH)$ model-measurement ratio where $RH_{ref}$=0% (for the model) and $RH_{ref}$ is at dry conditions (for the measurements), meaning measurement $RH_{ref}$ can be any value below 40% - whatever the actual measurement condition was (see Fig. S2). Model overestimation is found to be larger when $RH_{ref}$ is set to 0% for the GEOS-family models (GEOS-Chem, GEOS-GOCART, MERRAero), IFS-AER and SALSA and, to a lesser extent, for ATRAS and CAM-OSLO. A recent study by Latimer and Martin (2019) show a positive bias in the GEOS-Chem model for the GADS hygroscopicity paramterization which appears to be more significant at low (RH<35%) conditions. This finding is consistent with the results shown in Fig. 6 for GEOS-Chem model.

The ratio of the modeled $f(RH)$ to measured $f(RH)$ when $RH_{ref}$=0% is 1.64, and it decreases to 1.15 when using $RH_{ref}$=40%. The implication is that the models that exhibit such large differences between $RH_{ref}$=0% and $RH_{ref}$=40% conditions are simulating significant hygroscopic growth between 0-40% RH. Such growth would often not be seen by the measurements because the measurements are rarely (if ever!) that dry. In contrast, CAM and TM5 exhibit very little difference in their $f(RH=0\%)$ and $f(RH=40\%)$ histograms. This suggests these two models assume little growth below RH=40% and this is seen in Fig. 5 for the specific case of sea salt. In particular, MAM in CAM model assumes that if RH<35% the aerosol particles have fully crystallized (are in solid state) and have not taken up water. ~~As with the distribution for RH_{ref}=40%, the only model~~

showing underestimation of measurements for RH~ref~=0 % is IFS-AER. This underestimation is larger if RH~ref~ is 40 % (ratio of 0.74 for RH~ref~=40 % and 0.88 for RH~ref~=0 %).

The comparison presented in Fig. 6 highlights the differences in the model hygroscopicity parameterizations at the lower RH range (e.g. not fully dried particles and hysteresis effects). The discrepancy in $f$(RH) for the two RH~ref~ conditions presented in Fig. 6 is consistent with the hygroscopic growth simulated between RH=0 and 40 % (i.e., $f$(RH=40 %/RH=0 %), shown in Fig. S10. This finding is further supported by the minimal shift in the $f$(RH) probability distribution function when the two RH~ref~ values are considered (Fig. S11).

This difference between the comparison at RH~ref~=0 % and RH~ref~=40 % may also explain the results of Gliss et al. (2019). They performed model/measurement comparisons for both in situ scattering and aerosol optical depth (AOD). For their in situ scattering comparison, 'dry' scattering measurements were compared with model simulations reported at RH=0 %; they found that the ensemble model value underestimated the observed scattering by 33 %. In contrast, for the AOD comparisons, which were at ambient conditions for both models and measurements, the ensemble model value underestimated only by approximately 20 % (10-33 % depending on the source of AOD data). Thus, Gliss et al. (2019)'s larger model underestimate for in situ scattering than AOD may be due, at least in part, to the disconnect between the model and measurement definition of 'dry', although obviously other factors may also play a role. The results from this study and Gliss et al. (2019) imply that models would need to simulate higher aerosol loads and surface concentrations (or higher mass extinction coefficients) along with a reduced $f$(RH) to reduce the overall bias between models and measurements. This type of comparison demonstrates the usefulness of evaluating models against a variety of independent atmospheric observations - here it suggests further exploration of the role of hygroscopic growth across a range of RH values is warranted.

## 5   Conclusions

This works presents the first comprehensive model-measurement evaluation exercise for aerosol hygroscopicity and its effect on light scattering (22 sites, 10 Earth system models). Model simulations of the scattering enhancement factor $f$(RH), for the year 2010 were compared to spatially collocated measurements. The models exhibited large variability and diversity in the simulated $f$(RH), but tended to overestimate $f$(RH) relative to the measurements (with the exception of the IFS-AER model) when the reference relative humidity is RH~ref~=40 %. The mean ratio between modeled $f$(RH) and measurements is 1.15. 1.16 (0.74 for IFS-AER). The GEOS-family models and IFS-AER tend to simulate a narrow range of $f$(RH) relative to the other models. possibly related to use of the GADS parameterization and/or mixing state (although other unconsidered model assumptions may also be relevant). Hygroscopic growth factors for the different simulated chemical species vary among the models and we attribute the narrow range in $f$(RH) to the high growth factors the GEOS-family models and IFS-AER assume for all species except dust, which limits the range of $f$(RH) those models can simulate.

The chemical composition simulated by each model was compared and exhibited both similarities and differences across the sites studied. The GEOS-family of models tend to simulate more dust at many sites than the other models. The simulated chemistry was used to compare the modeled relationship between organic mass fraction and $f$(RH) with various results from

observational field campaigns. Models which assumed little to no hygroscopic growth for organic aerosol were better able to reproduce the observed relationship than models which assumed high growth factors. It was possible to explain some of the variability in model $f$(RH) at a marine site by comparing the simulated $f$(RH) when models simulated an aerosol dominated by sea salt. Model assumptions about water uptake at low RH were a significant factor, but different assumptions about the hygroscopicity of sea salt also played a role. Some models assumed the hygroscopicity of sea salt could be represented by NaCl, while others assumed water uptake characteristics more similar to the observed hygroscopicity of inorganic sea salt.

Overall, all models fail to capture the annual cycle of observed $f$(RH) at three sites representing distinct regimes (Arctic, rural, and marine) when it was possible to also temporally collocate the observations. Temporal collocation did not appear to improve the comparison of model simulations and observations relative to the comparison with multi-year climatological values. The diversity of the models tended to be larger than the variability in the observed long-term climatology at these three sites.

Agreement between models and measurements was strongly influenced by the choice of $RH_{ref}$. Better agreement between observations and models is found when $RH_{ref}$=40%. In addition, some models exhibited unexpectedly large differences in $f$(RH) at low RH (i.e., modeled scattering enhancement was significantly different for $RH_{ref}$=0 % and $RH_{ref}$=40 %), pointing to the sensitivity in the model parameterization of hygroscopic growth at low RH (e.g., effects of particle hysteresis). This was explicitly demonstrated for the modeled sea salt component, but may also be relevant for other species which exhibit hysteresis. To address this for future evaluations, models and measurements should be compared at similar RH conditions. For example, models could calculate $f$(RH) at the same variable RH conditions as the measurements. This type of study will make the model-measurement comparison more challenging since the same RH conditions should be matched and measurement conditions can vary widely with site and season. ~~, although that would be computationally more intensive since measurement conditions can vary with site and season.~~ Alternatively, if measurements could better control their reference RH, both keeping it below 40 % and maintaining a narrower distribution of $RH_{ref}$, there would be less uncertainty in the model/measurement comparisons. Caution must always be taken when changing the measurement conditions - semivolatile species may volatilize with decreasing RH, inducing a negative artifact. While such losses are known and characterized for some species such as ammonium nitrate, we are still far away from a quantitative understanding such effects for semi-volatile organic species.

Based on the results presented here there are several topics that should be explored. One is to evaluate whether the gamma fit parameter is a more robust indicator for model/measurement comparisons than $f$(RH). Doing so would require model and measurement scattering data over a range of RH conditions. Another avenue is related to the $f$(RH) dependence on both chemical composition of the particles and particle size. Measured chemistry and size data collocated with scattering enhancement measurements at the sites where that information is available could be used in future work to assess modeled simulations of these factors and their impact on modeled scattering enhancement. The diversity in simulated chemical composition at many of the sites suggests this should be pursued. Comparison of size distributions is more challenging due to the variety of methodologies used by the different models to represent aerosol size. Evaluating model size distributions with measurements is a step beyond that and would require integration of measurements from several instruments to get a complete size distribution covering the full range of aerosol sizes simulated by the models. ~~Finally, a~~ Another challenging task on the measurement side is to measure

the scattering at RH > 85 % (e.g., 90-100 %) where the steepest hygroscopic growth happens and where models introduce large diversity in $f$(RH) due to assumptions on sub-grid scale humidity fluctuations and cloud versus cloud-free conditions.

Finally, we recommend that models update their hygroscopic growth parameterization for sea salt by assigning a lower and more realistic hygroscopic growth factor rather than assuming sodium chloride to be representative of sea salt. Models should also, if possible, explicitly provide $f$(RH) at specified RH values for pure components (i.e. for the sulfate or organic components) separately, which can then be compared to theory and observations. In addition, to further evaluate the influence of mixing state and particle size, a new multi-model experiment with a common hygroscopicity scheme would be desirable (e.g., within AeroCom).

*Code and data availability.* The measurement data behind this study is already publicly available (see Burgos et al., 2019). The entire dataset, incl. the corresponding model data, and analysis code is available at the Bolin Data Centre (https://bolin.su.se/data/burgos-2020-esm).

*Author contributions.* M.B. performed the model-measurement evaluation. M.B., E.A., G.T., and P.Z. designed study and wrote the paper. A.B., H.B., V.B., G.C., A.K., H.K., A.L., H.M., G.M., C.R., N.S., T.N. and K.Z. designed and performed model calculations. L.A., U.B., A.J., J.S., J.S., E.W., G.T., and P.Z. provided measurement data. All authors read and commented on the manuscript.

*Competing interests.* The authors declare no competing interests.

*Acknowledgements.* This work was essentially supported by the Department of Energy (USA) under the project DE-SC0016541.

The JFJ measurements and the work by P.Z., U.B. and E.W. were financially supported by the ESA project Climate Change Initiative Aerosol cci (ESRIN/Contract No. 4000101545/10/I-AM), the Swiss National Science Foundation (Advanced Postdoc.Mobility fellowship; Grant No. P300P2_147776), and by the EC-projects Global Earth Observation and Monitoring (GEOmon, contract 036677) and European Supersites for Atmospheric Atmospheric Aerosol Research (EUSAAR, contract 026140). We thank the China Meteorological Administration for their continued support to Lin'an Atmospheric Background Station; National Scientific Foundation of China (41675129), National Key Project of Ministry of Science and Technology of the People's Republic of China (2016YFC0203305 & 2016YFC0203306), Basic Research Project of Chinese Academy of Meteorological of Sciences (2020KJ001) (2017Z011). It was also supported by the Innovation Team for Haze-fog Observation and Forecasts of MOST and CMA. China Meteorological Administration.

CAM5.3-Oslo model development and simulations for this study were supported by the Research Council of Norway (grant nos. 229771 and 285003), by Notur/NorStore (NN2345K and NS2345K), and by the Nordic Centre of Excellence eSTICC (grant no. 57001). We acknowledge the Academy of Finland Projects 317390 and 308292. The ECHAM-HAMMOZ model is developed by a consortium composed of ETH Zurich, Max Planck Institut für Meteorologie, Forschungszentrum Jülich, University of Oxford, the Finnish Meteorological Institute and the Leibniz Institute for Tropospheric Research, and managed by the Center for Climate Systems Modeling (C2SM) at ETH Zurich.

KZ was supported by the Office of Science of U.S. Department of Energy. KZ thanks Steve Ghan for the help and support on the CAM5 AeroCom submission.

HM acknowledges funding from the Ministry of Education, Culture, Sports, Science, and Technology and the Japan Society for the Promotion of Science (MEXT/JSPS) KAKENHI Grant Numbers JP17H04709, JP16H01770, JP19H04253, JP19H05699, and JP19KK0265, the MEXT Arctic Challenge for Sustainability (ArCS) projects, and the Environment Research and Technology Development Fund (2–1703) of Environmental Restoration and Conservation Agency.

We thank Mian Chin (NASA Goddard) and the AeroCom community for valuable discussions.

# 6 Tables

**Table 1.** General site information. The median $RH_{ref}$ refers to the relative humidity inside the (dry) reference nephelometer, while the temporal resolution refers to measured values of $f$(RH). More details and references on the sites can be found in Burgos et al. (2019).

| Station ID | Station Name, Country | Latitude (°) | Longitude (°) | Site Type | Median $RH_{ref}$ (%) | Temporal Resolution (h) |
|---|---|---|---|---|---|---|
| BRW | North Slope of Alaska, USA | 71.3 | -156.6 | Arctic | 6.8 | 6 |
| ZEP | Zeppelin, Norway | 78.9 | 11.9 | Arctic | 11.6 | 6 |
| JFJ | Jungfraujoch, Switzerland | 46.6 | 8 | Mountain | 5.2 | 3 |
| CBG | Chebogue Point, Canada | 43.8 | -66.1 | Marine | 28.2 | 1 |
| GRW | Graciosa, Portugal | 39.1 | -28 | Marine | 28.5 | 1 |
| GSN | Gosan, S. Korea | 33.28 | 126.2 | Marine | 33.0 | 1 |
| MHD | Mace Head, Ireland | 53.3 | -9.9 | Marine | 26.4 | 3 |
| PVC | Cape Cod, USA | 42.1 | -70.2 | Marine | 24.0 | 1 |
| PYE | Point Reyes, USA | 38.1 | -123 | Marine | 28.9 | 1 |
| THD | Trinidad Head, USA | 41.1 | -124.2 | Marine | 28.8 | 1 |
| APP | Appalachian State, USA | 36.2 | -81.7 | Rural | 13.6 | 1 |
| CES | Cabauw, Netherlands | 52 | 4.9 | Rural | 13.3 | 3 |
| FKB | Black Forest, Germany | 48.5 | 8.4 | Rural | 21.5 | 1 |
| HLM | Holme Moss, UK | 53.5 | -1.9 | Rural | 27.6 | 1 |
| HYY | Hyytiälä, Finland | 61.9 | 24.3 | Rural | 28.2 | 3 |
| LAN | Lin'an, China | 30.3 | 119.7 | Rural | 12.2 | 1 |
| MEL | Melpitz, Germany | 51.4 | 12.9 | Rural | 10.7 | 3 |
| SGP | Southern Great Plains, USA | 36.6 | -97.5 | Rural | 18.3 | 1 |
| HFE | Shouxian, China | 32.6 | 116.8 | Urban | 22.4 | 1 |
| PGH | Nainitial, India | 29.4 | 79.5 | Urban | 30.4 | 1 |
| UGR | Granada, Spain | 37.2 | -3.6 | Urban | 15.9 | 1 |
| NIM | Niamey, Niger | 13.5 | 2.2 | Desert | 9.4 | 1 |

**Table 2.** Summary of main characteristics implemented by each model. Model main reference, meteorology, mean RH value (subgrid variability considered), mixing state (black carbon), and species (number of size bins). In Meteorology column: GMAO = Global Modeling and Assimilation Office. In Mixing State column: E = external, I = internal. In Species and size bins column: BC = black carbon, OA = organic aerosol, MSA = methane sulfonic acid.

| Model (temporal resolution) | Main Reference | Meteorology | Mean RH (subgrid variability) | Mixing State | Species (size bins) |
|---|---|---|---|---|---|
| ATRAS (1h) | Matsui (2017) | Nudged to MERRA | clear-sky (no) | I | sulfate, dust, sea salt, BC, and OA, nitrate, ammonium (12) |
| CAM (1h) | Liu et al. (2012b) | Nudged to ERA interim | clear-sky (no) | I | sulfate, dust, sea salt, BC, and OA. 3 modes: Aitken, accumulation and coarse |
| CAM-OSLO (1h) | Kirkevåg et al. (2018) | Nudged to ERA interim | grid (no) | I, E | sulfate, dust, sea salt, BC, and OA (distributed in 12 modes) |
| GEOS-Chem (1h) | Bey et al. (2001) | GEOS5 version of NASA GMAO | grid (no) | E | sulfate, nitrate, ammonium, BC, and OA (bulk-mass), dust (4), sea salts (2) |
| GEOS-GOCART (24h) | Chin et al. (2002) | MERRA reanalysis | grid (no) | E | sulfate, dust and sea salt (5), BC and OA (2) |
| MERRAero (3h) | Buchard et al. (2015) | MERRA reanalysis | grid (no) | E | sulfate, dust and sea salt (5), BC and OA (2) |
| OsloCTM3 (24h) | Lund et al. (2018) | Offline meteorology from IFS ECMWF | grid (no) | I for hydrophilic BC | sulfate, dust (8), sea salt (8), BC, primary and secondary OA, nitrate (2), and ammonium (2) |
| TM5 (1h) | van Noije et al. (2014) | Offline, ERA-Interim | clear-sky (no) | I,E | sulfate, dust, sea salt, BC, and OA (7), ammonium nitrate, and MSA |
| IFS-AER (3h) | Morcrette et al. (2009) | online, initial conditions NWP analysis | grid (no) | E | sulfate, dust (3), sea salt (3), OA and BC (2), nitrate and ammonium |
| SALSA (1h) | Kokkola et al. (2018) | Nudged to ERA interim | clear-sky (no) | E | sulfate, dust, sea salt, OA, and BC (10) |

**Table 3.** Summary of the hygroscopic growth parameterization used by each model and the hygroscopic growth factor ($g$(RH), defined as the wet divided by the dry particle diameter) values for the main chemical species at RH=90%. For models which use the $\kappa$-Köhler parameterization (given in squared brackets), we state $g$(RH) calculated using $g(RH) = (1 + \kappa * RH/(1 - RH))^{1/3}$ (see Petters and Kreidenweis, 2007, here ignoring the Kelvin effect) for comparison reasons. Note, all models use different size parameterizations which vary in particle size and resolution (see Table 2 and Sect. 3). Boxes with hyphen note that this component is not considered by the model. *Only for hydrophilic fraction. ** Either as nitrate or sulphate. ***Parameterizations by Vignati et al. (2004)

| Model | Hygroscopicity parameterization | SS | SO$_4$ | BC | OA POA | OA SOA | NO$_3$ | NH$_4$ | Dust |
|---|---|---|---|---|---|---|---|---|---|
| ATRAS | Based on $\kappa$-Köhler theory | 2.25 [$\kappa = 1.16$] | 1.87 [$\kappa = 0.61$] | 1.0 [$\kappa = 10^{-10}$] | 1.24 [$\kappa = 0.1$] | 1.24 [$\kappa = 0.1$] | 1.87 [$\kappa = 0.61$] | 1.87 [$\kappa = 0.67$] | 1.0 [$\kappa = 0.001$] |
| CAM | Based on $\kappa$-Köhler theory | 2.25 [$\kappa = 1.16$] | 1.77 [$\kappa = 0.507$] | 1.0 [$\kappa = 10^{-10}$] | 1.24 [$\kappa = 0.1$] | 1.31 [$\kappa = 0.14$] | - | - | 1.17 [$\kappa = 0.068$] |
| CAM-OSLO | Based on $\kappa$-Köhler theory | 2.28 [$\kappa = 1.2$] | 1.77-1.80 [$\kappa = 0.507 - 0.534$] | 1.00 [$\kappa = 5 * 10^{-7}$] | 1.31 [$\kappa = 0.14$] | 1.31 [$\kappa = 0.14$] | - | - | 1.17 [$\kappa = 0.069$] |
| GEOS-Chem | Modified GADS/OPAC | 2.38 | 1.64 | 1.4 | 1.64 | 1.64 | 1.64 | 1.64 | 1.0 |
| GEOS-GOCART | Modified GADS/OPAC | 1.90-2.17 | 1.8 | 1.4* | 1.6 | 1.6 | - | - | 1.0 |
| MERRAaero | Modified GADS/OPAC | 1.90-2.17 | 1.8 | 1.4 | 1.64 | - | - | - | 1.0 |
| OsloCTM3 | Own development (see Sect. 3.7) | 2.31-2.39 | 1.72 | 1.0 | 1.46 | 1.46 | 1.82 | ** | 1.0 |
| TM5 | Own development (see Sect. 3.8) | *** | *** | 1.0 | 1.0 | - | *** | *** | 1.0 |
| IFS-AER | Own development (see Sect. 3.9) | 2.36 | 1.73 | 1.0 | 1.64 | - | 1.7 | 1.73 | 1.0 |
| SALSA | Own development (see Sect. 3.10) | 2.4 [$\kappa = 1.46$] | 1.9 [$\kappa = 0.68$] | 1.0 [$\kappa = 0$] | 1.5 [$\kappa = 0.3$] | - | - | - | 1.0 [$\kappa = 0$] |

Note, that the sea salt (SS) component is often assumed to have the same hygroscopic growth as sodium chloride. However, it has been recently shown that pure inorganic sea salt has a 8-15% lower hygroscopic growth than sodium chloride (Zieger et al., 2017).

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

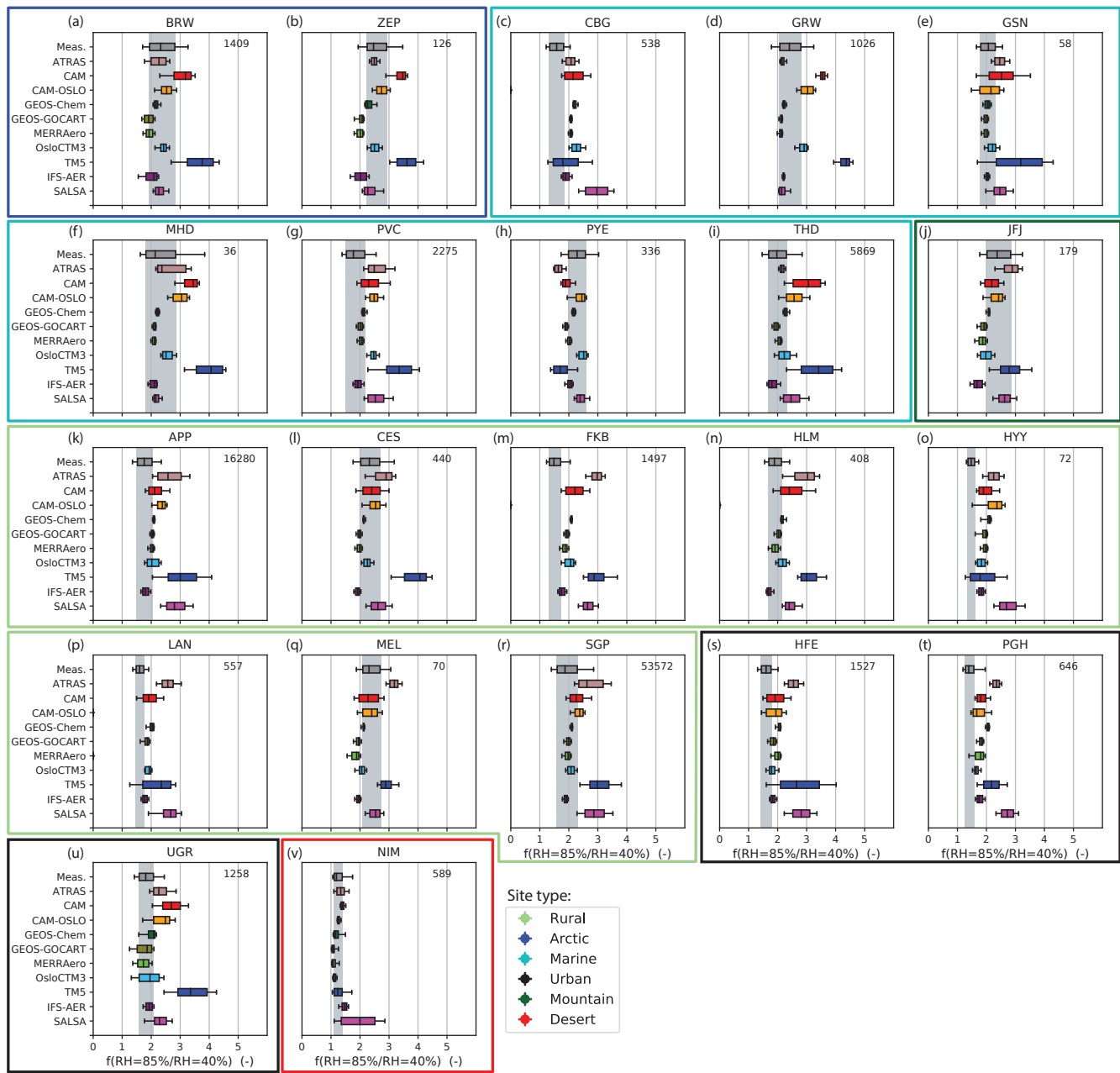

**Figure 1.** The scattering enhancement $f(\mathrm{RH}=85\,\%/\mathrm{RH}_{\mathrm{ref}}=40\,\%)$ at $\lambda = 550\,\mathrm{nm}$ as measured and predicted by the various models for all investigated sites (panel **(a)** - **(v)**). The box edges represent the $25^{\mathrm{th}}$ to the $75^{\mathrm{th}}$ percentile (the gray underlying area represents the quartiles for all measurements), with the center line indicating the median. The whiskers show the range of the data extending from the percentiles 10th to 90th. The number in the top right corner indicates the number of available measurements at each site (temporal resolution shown in Table 1). The colored boxes grouping the different sets of plots indicate the site type. Note: Figure has been updated.

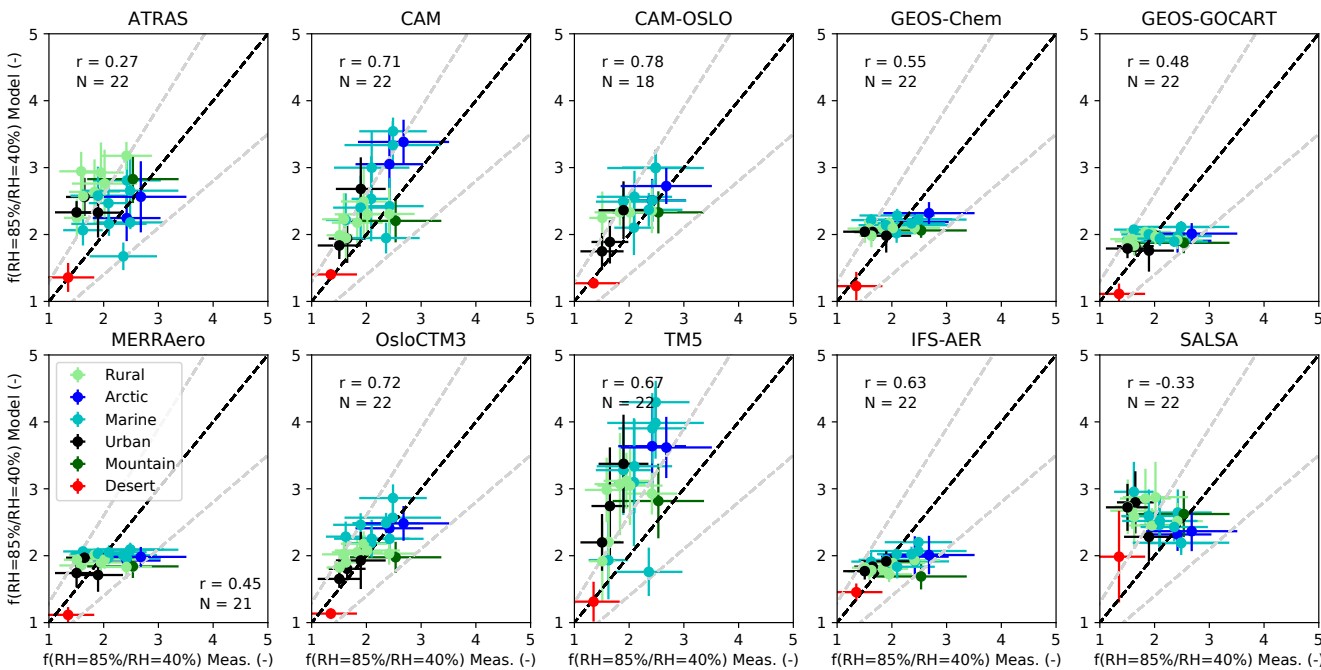

**Figure 2.** Simulated versus measured $f$(RH=85 %/RH$_{\mathrm{ref}}$=40 %) at $\lambda = 550\,$nm for each model color-coded by site type: blue for Arctic, cyan for marine, dark green for mountain, light green for rural, black for urban, and red for desert sites (panel **(a)** - **(j)**). The Pearson correlation coefficient (r) and the number of sites are indicated for each panel. The dashed black line shows the 1:1-line and gray dashed line shows the upper estimate of measurement uncertainties. Note: Figure has been updated.

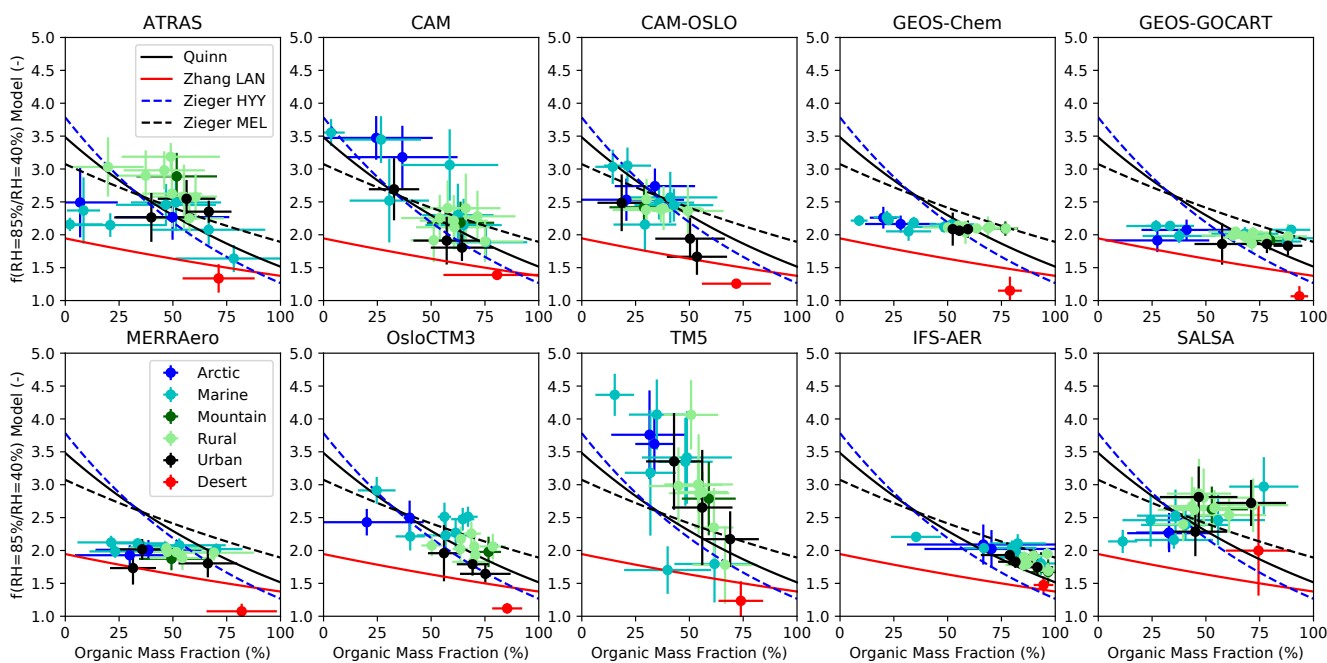

**Figure 3.** $f$(RH=85 %/RH$_{\text{ref}}$=40 %) vs. organic mass fraction for each model considered in this study. Each point represents one site, which are color-coded by site type. Parameterizations by Quinn et al. (2005), Zhang et al. (2015), and Zieger et al. (2015) represented by the solid and dotted lines. Additional new figure.

.

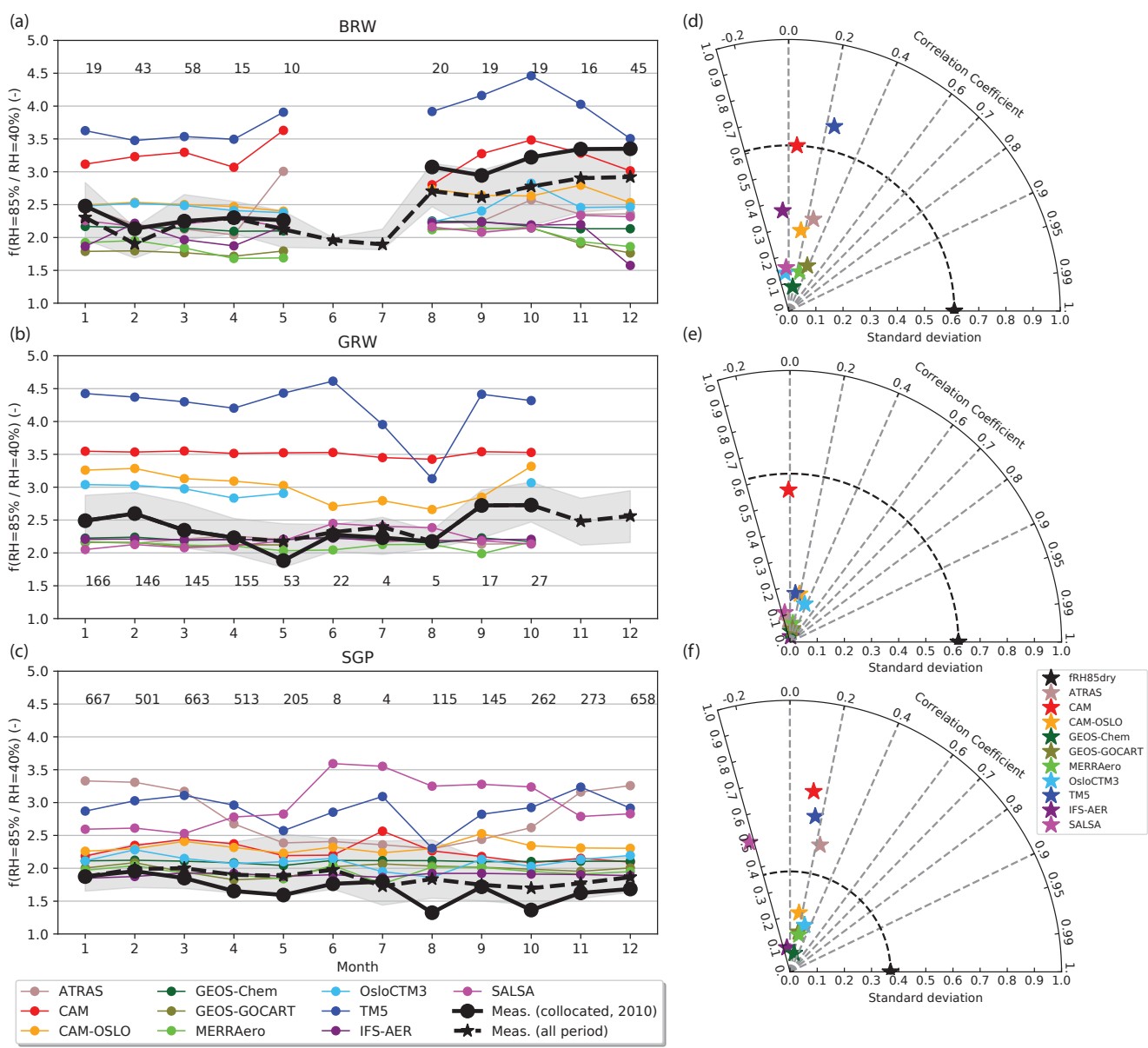

**Figure 4.** Comparison of $f$(RH=85 %/RH$_{ref}$=40 %) at $\lambda = 550$ nm for 2010: Barrow (Arctic site), Graciosa (marine site), and Southern Great Plains (rural site). **(a)-(c)** Annual cycles of the median $f$(RH=85 % /RH$_{ref}$=40 %) as measured (black line) and as predicted by the models (colored lines) collocated for 2010. The black dashed line and gray underlying area represent the median and range for the entire dataset. The numbers of data points in each month are also indicated. **(d)-(f)** Taylor diagrams showing the correlation coefficients and standard deviations of $f$(RH=85 %/RH$_{ref}$=40 %) for measurements (black symbols) and models (colored symbols, see legend). Note: Figure has been updated.

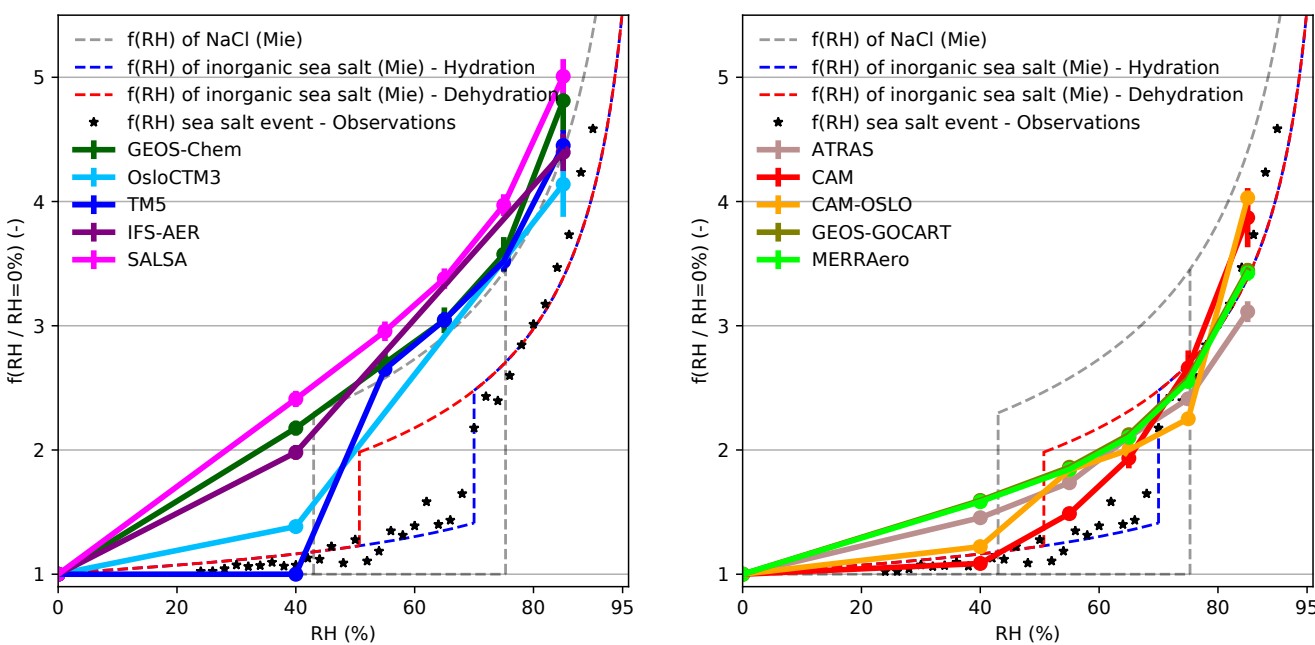

**Figure 5.** The scattering enhancement factor $f$(RH) vs. RH for sea salt dominated aerosol at Graciosa (GRW) as predicted by the different models (left panel: GEOS-Chem, OsloCTM3, TM5, IFS-AER, SALSA. Right panel: ATRAS, CAM, CAM-Oslo, GEOS-GOCART, and MERRAero). The model data is shown for cases when the predicted sea salt mass fractions was larger than 95 %. For comparison, the expected values for $f$(RH) of (i) NaCl determined by Mie modelling, (ii) for inorganic sea salt determined by Mie modeling based on H-TDMA sea salt chamber measurements of Zieger et al. (2017) are shown. The dashed blue and red lines show the corresponding hydration and dehydration line, respectively. Field measurements of $f$(RH) for pristine sea salt aerosol are shown as black stars (taken from Zieger et al., 2010). Additional new figure.

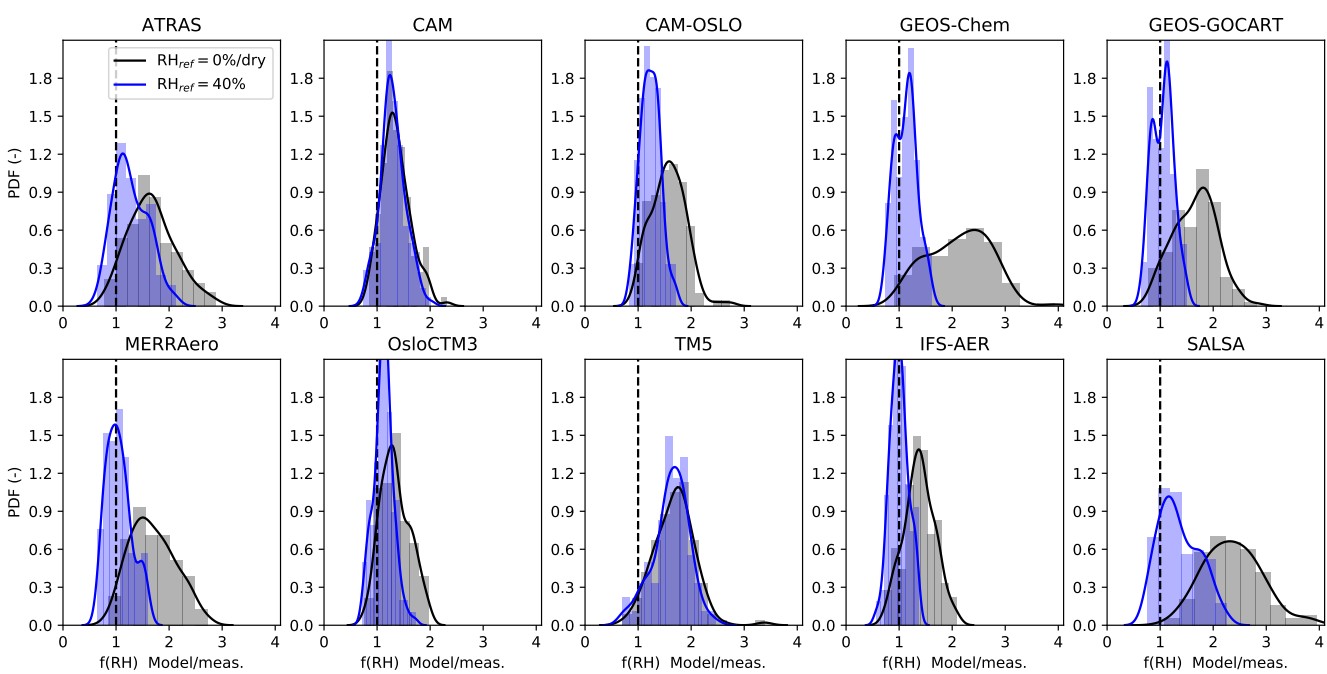

**Figure 6.** Probability density functions of the ratio $f(\mathrm{RH})_{\mathrm{model}}/f(\mathrm{RH})_{\mathrm{meas.}}$ for all sites for each model. The blue values denote the ratios if RH=40 % is taken as reference RH. The gray areas represent the ratio if $\mathrm{RH}_{\mathrm{ref}}$=0 % (models) or $\mathrm{RH}_{\mathrm{ref}}$=dry (measurements) is taken. Note: Figure has been updated.

.