# Peer review of "A global model-measurement evaluation of particle light scattering coefficients at elevated relative humidity"

_Atmospheric Chemistry and Physics, 2019_

## Referee Comment (RC1) · Anonymous Referee #1 · 2 Mar 2020

This study evaluates the model simulations of aerosol scattering enhancement factor using in situ measurements. Model evaluation of the hygroscopic growth of aerosols is a valuable contribution to modeling community. The authors clearly described the methods and results, but my impression is that this manuscript reads like a technical report, rather than a scientific paper. The temporal collocation and the definition of dry RH are the two major discussion points, but I think they are mostly technical issues, rather than scientific findings. The authors show the hygroscopic growth of aerosols in models is highly uncertain and varies across models. I think what is more important is to explain why there is large spread across models, and how the results of this study could help future improvement of modeling aerosol optical properties. Without

sufficient explanations and scientific insights, I don't think this manuscript is suitable to be published at ACP that emphasizes scientific significance. I have the following comments that should be addressed.

1. The most important parameter that determines the f(RH) is the hygroscopic growth factor used in models, but the authors did not explain the differences across models. Table 2 only lists the factor for sea salt, but not other species. The hygroscopic factors of organic carbon, for example, have large uncertainties (Duplissy et al., 2011). What is the hygroscopic factor of OC used in these models? Same for other species. I'd suggest the authors explicitly compare hygroscopic growth factors for each species in each model, which may also help explain the differences across models.

2. The light scattering coefficients used in models are not discussed either.

3. As the authors mentioned, f(RH) largely depends on the aerosol size distribution, mixing state and aerosol composition. Model disagreements in f(RH) should also be attributed to these differences, but the authors did not explain how model differs in in size distribution and aerosol composition, and how this leads to differences in f(RH). For aerosol size distribution, the authors should explain how the size distribution is parameterized in model (e.g. log normal distribution? Modal radius?). For aerosol composition, a simple test would be to compare the aerosol chemical composition across models. I think this manuscript will be valuable contribution to if the authors could explain why the models differ, beyond the two technical issues.

4. The authors mention the mixing state of aerosols may partially explain the narrower range of GEOS-family models, but I don't understand the reasonings for this. The light extinction should decrease when changing from external to internal mixing because the aerosol size becomes larger while the number concentration decreases (Curci et al., 2015). I'd suggest the authors conduct further analysis on this, and provide more convincing explanations.

5. The hysteresis effect of aerosols is an important factor for modeling hygroscopic

growth, but it is not addressed in the manuscript. Few models consider the hysteresis effect, which could partially lead to model-measurement disagreement. The authors briefly mentioned this in the supplement, but I think this should be an important issue to discuss.

6. Line 15 of Page 13: Why is there seasonality of aerosol hygroscopicity in some sites? Is it due to seasonal variation of aerosol composition?

7. Figure 2: I'd suggest include the number of data pairs in the figures.

8. Figure 3: The seasonal variation in hardly seen. I think you could change the y axis limit to a smaller range?

References: Duplissy, J, P F DeCarlo, J Dommen, M R Alfarra, A Metzger, I Barmpadimos, A S H Prevot, et al. 2011. "Relating Hygroscopicity and Composition of Organic Aerosol Particulate Matter." Atmospheric Chemistry and Physics 11 (3), 1155–65. doi:10.5194/acp-11-1155-2011.

Curci, G, C Hogrefe, R Bianconi, U Im, A Balzarini, R Baró, D Brunner, et al. 2015. "Uncertainties of Simulated Aerosol Optical Properties Induced by Assumptions on Aerosol Physical and Chemical Properties: an AQMEII-2 Perspective." Atmospheric Environment 115 (c). 541–52. doi:10.1016/j.atmosenv.2014.09.009.
* * *

---

## Referee Comment (RC2) · Anonymous Referee #2 · 2 Mar 2020

This paper discusses an important aspect of aerosol modelling: the growth of aerosol with increasing humidity. This wet-growth has important implications for aerosol transport and aerosol optical properties (including aerosol-radition-interactions). The authors conduct a process study by comparing the light scattering enhancement factor f(RH) from models with that from several observation sites. The observational data is novel and the evaluation of models highly relevant to numerous modelling groups, including the AEROCOM community.

General comments: The study itself seems properly conducted and the paper is mostly well-written. However, it seems the authors were only interested in a very narrow

objective: numerically comparing model data with observations. There is very little interpretation of the results. While I appreciate that a full interpretation might be a study in itself (and the authors suggest they are working on it), this is rather unsatisfying. It prevents the reader from tapping in to the combined expertise of the authors (observers & modellers) and a better understanding of the limitations and opportunities associated with the current study.

In particular, I missed a discussion of why models might have a different f(RH) from the observations. I suppose there are at least three reasons: 1) incorrect wet-growth of individual species (e.g. incorrect kappa); 2) incorrect internal mixing rule for wet-growth; 3) incorrect internal and external mixing states. The advantage of a process study is of course that the models do not need to accurately simulate mass loads themselves.

If possible, it would be useful to provide more information on per-species wet-growth in individual models, especially because of the finding of substantial wet-growth at low RH.

In addition, I found important information on e.g. observational errors and methodology to be missing. Yes, the authors refer to Burgos et al 2019 but it would be good if brief (!) summaries of relevant sections in Burgos et al 2019 are provided.

Specific comments:

Abstract: the abstract contains several conclusions without any attempt at interpretation.E.g. "An important finding is that the models show a significantly larger discrepancy with the observations if RHref =0 % is chosen as the model reference RH compared to when RHref =40 % is used" might become "The definition of dry conditions is difficult from an observational point-of-view, which affects our model evaluation negatively as several models exhibit significant and unexpected wet-growth between RH=0 and 40%.". One interesting finding (also supported by a recent Gliss study) is not included in the abstract.

Introduction: I miss a discussion of the importance of correctly simulating wet-growth in models. How does wt-growth affect different aspects of simulation? E.g. emissions are unaffected but both wet and dry deposition are. Consequently, so is transport. At the same time, optical properties (important for ARI) are affected. Anything else (e.g. chemistry)? While wet-growth ultimately leads to activation of aerosol as cloud droplets, models usually disconnet these processes.

Two papers that consider impact of f(RH) on modelling of biomass burning: Johnson, B. T., Haywood, J. M., Langridge, J. M., Darbyshire, E., Morgan, W. T., Szpek, K., . . . Bellouin, N. (2016). Evaluation of biomass burning aerosols in the HadGEM3 climate model with observations from the SAMBBA field campaign. Atmospheric Chemistry and Physics, 16, 14657–14685. https://doi.org/10.5194/acp-16-14657-2016 Reddington, C. L., Morgan, W. T., Darbyshire, E., Brito, J., Coe, H., Artaxo, P., Scott, C. E., Marsham, J., and Spracklen, D. V.: Biomass burning aerosol over the Amazon: analysis of aircraft, surface and satellite observations using a global aerosol model, Atmos. Chem. Phys., 19, 9125–9152, https://doi.org/10.5194/acp-19-9125-2019, 2019.

An introduction should end with a: - a brief description of the paper's methods and goals, and why/how they adbvance the field (i.e. add to the existing body of work) - a short description of the content (sections listing). I miss both and the introduction would become substantially stronger if they are added. P 3, l 20-34 does not really provide this.

p 4, sect 2: what I miss here is a brief discussion on expected uncertainties. I have no idea how much can be said about this but admitting to 'uncertainty in the uncertainty'would be acceptable. E.g. at RH=40 or 85%, can we expect measurement uncertainties in f of 5, 10, 15% in individual measurements? Derscribe briefly the main causes of uncertainty. Do you expect errors at a single site to behave like biases or random errors? This discussion would be very useful. If this was discssed in Burgos et al 2019, please provide highlights and the reference.

p 4, l 16: given the uncertainty in the analysis at low RH later identified, it would be good to be a bit more precise about low RH measurements? Do the various instruments measure at different low RH? Does a single instrument measure at a single RH, or does it vary for some reason? If it varies, does it vary in a controlled fashion , or not? I see that Table 1 only provides median values and gives not indication of any variation (large or small). p 5, Section models: it seems that the models broadly fall into 2 (3?) categories based on how f(RH) is calculated: either from direct parametrisation (e.g. OPAC), or from Koehler theory (is it fair to include ZSR theory?), or from equilibrium theory (sulphate-nitarte-amoonia, see e.g. Seinfeld & Pandis). I'm not sure the latter category is present in the current family of models but maybe TM5 uses it? Anyway, would it make sense to stress these three broad categories?

p 5, l 3: can the authors say a little about what aspects of data homogenization are considered important, and were different in Titos et al.? It would help make the paper stand out more.

p 5, l 26: So the GEOS family of models are CTMs? Isn't MERRAero an assimilation product?

p 5, l 27: There is no information in Table 2 on hygrospcopic growth of sulphate (most important anthropogenicv aerosol) and organics (to my knowledge, the most uncertain aerosol wrt hygrospcopic growth). Is it possible to include at least these species?

p 5, l 29: is there a name to identify the relevant AEROCOM experiment? Please also provide link to AEROCOM website so interested readers can follow up.

p 5, l 32: I assume surface values were used? PLease state so.

p 6, l 1: Here or at a more appropriate location, it would be good to have a brief discussion of possible impact of the difference between model and observational years.

p 7, l 16: (Bey at al.) -> Bey et al.

p 10, l 7: all hourly output for that month is used, or only at the hours (and presumably

days) of observations. Given thge possible importance of representation errors, it may be worthwile to have a short section (Methods) that describes actual collocation procedure in a bit more detail, especially as there are basically two cases: sites that provide data for 2010, and sites that don't.

p 10, l 20: is the gray shaded area for measurements mostly due to measurement error or temporal variability? This has consequences, because if it is due to measurement error, a lot of models greatly overestimate f(RH) variability.

p 10, l 23: this information could be part of the short section suggested above?

p 10, l 27: what is the altitude of these mountain sites? Is it ok to use those measurements for model evaluation when model's orography may not ab able to 'simulate' the mountain?

p 11, l 10-20: ignoring the observations, it appears to me that the models exhibit fairly similar behaviour independent of site. E.g. GOES family rather low f(RH) with small variation, CAM, TM5 and SALSA higher f(RH) with more variation. Can anything be said about that? p 11, l 21: if I'm correct, this is the same data as in Fig 1 but represented in a different way? It may be good to specifically state so.

p 11, l 23: This uncertainty should be mentioned earlier, when describing the measurememnts. Is this the uncertainty in a single measurement, or in an average of measurements?

p 11, l 31: Is it fair to conclude that external mixing reduces variation in f(RH)? And if so, why would that be?

p 12, l 2: are you sure? I thought SALSA uses a combination of internal and external mising.

p 12, l 27: for SGP I also see substantial differences but they seem to their own annual cycle (larger differences in summer).

p 12, l 28-30: nice analysis, ultimately uncertainties always have to be considered in the context of other uncertainties.

p 13, l 15-17: Dare the authors guess at why even CAM, OsloCTM and TM5 do poorly? They showed pretty strong correlation across the sites. So it appears that these models have skill in predicting spatial patterns but none in predicting temporal evolution. Yet it would appear that observed variation across sites is similar to variation within each site (guesstimating from Fig 1).

p 14, l 26-27: surely the developers of the GOES and SALSA models (who are co-authors) must have an idea where this is coming from? It would be very useful to add such information to the paper. From a modelling perspective, it would be relatively straight forward to present f(RH) curves for each species (i.c. of internal mixtures: predominant species). Such a figure would be a very useful addition to this paper.

p 15, l 10-15: this is an interesting finding that should be in the abstract (there is only an oblique reference to it at the moment).

p 15, Section Conclusions: the current text focusses entirely on techniqcal issues and ignores interesting findings as mentioned above. It would be good if more emphasis is given to lessons learned with regards to possible model deficiencies. There is no speculation why models over-estimate f(RH). Or do the authors believe this is due to remaining technical issues?

p 15, l 17: Define "f(RH)"

p 15, l 19: Define "RH_ref"

p 15, l 27: "at low RH". Did you mean "for low RH_ref"?

p 15, l 30: models should be able to provide f(RH) at multiple RH without any significant CPU or development overhead.

p 16, l 4-6: this gamma parameter has not been mentioned before so it's odd to dis-
cuss in the conclusions. Obviously, you may want to mention other possible analysis methods.

---

## Author Comment (AC1) · 9 Jun 2020

**Manuscript by Burgos et al. "A global model-measurement evaluation of particle light scattering coefficients at elevated relative humidity" - Reply to reviewers**

We thank the reviewers for their valuable comments, which helped to improve our manuscript. The reviewers comments are in *italicized black font*. Our replies are given below in blue, and when we refer to text that has been changed in the manuscript (main text and supplement materials) we show it in this reply letter in red (these correspond to the changes in the manuscript and supplement which are also shown in red), while original text is shown in black.

5  We would like to start with an overview of the main changes to the manuscript resulting from issues raised by both reviewers. We will then answer all specific comments by each reviewer.

Both reviewers asked for more details about other related variables that could be helpful to explain the results. Among those, one main comment was to study the influence played by chemical composition. To address this, we have further investigated the model chemical composition. We have studied the simulated relationship of $f(\text{RH})$ with the Organic Mass Fraction, which

10  has shown some value in observational studies. Additionally, we reviewed the annual modelled chemical composition for each site in the context of the $f(\text{RH})$. For the three sites considered in section 4.2, the seasonal cycle of chemical composition was also considered and provided insights into model assumptions relating to hysteresis.

Moreover, the reviewers suggested providing more information about the hygroscopic growth factors of all chemical species considered by the models. To address this, we have added this information in the new Table 3. We also analyzed in detail model

15  simulations for the marine site of Graciosa (GRW), enabling a better understanding of how the models treat the hygroscopic growth in the specific case of aerosol dominated by sea salt.

**1 Common issues raised by both reviewers:**

To address the common issues raised by both reviewers we have added to the revised manuscript:

1. Section 3.11 where we describe in more detail the characteristics of the models relevant to hygroscopic growth, size

20    distribution, chemical composition, and mixing state.

2. Table 2 has been split into two tables: Table 2 now describes the general characteristics of the models, and the new Table 3 provides a more thorough description of the models' hygroscopicity, including growth factors for all chemical species considered in each model.

3. An analysis related to simulated chemical composition of the models has been added to the manuscript as follows:

(a) New Figure 3: shows the modelled $f(\mathrm{RH})$ vs. the Organic Mass Fraction compared to parameterizations from the literature based on observations

(b) New Figure S4: shows the annual fractional contribution of chemical species based on simulated mass mixing ratios. for all models and all sites

(c) New Figures S5-S7: monthly cycles of fractional contribution of chemical species based on simulated mass mixing ratios for all models for three sites: Barrow, Graciosa, and Southern Great Plains

4. A new Figure S8 with the monthly values of median $f(\mathrm{RH}{=}85\,\% \,/\, \mathrm{RH}_{\mathrm{ref}}{=}40\,\%)$, $f(\mathrm{RH}{=}85\,\% \,/\, \mathrm{RH}_{\mathrm{ref}}{=}0\,\%)$, and $f(\mathrm{RH}{=}40\,\% \,/\, \mathrm{RH}_{\mathrm{ref}}{=}0\,\%)$ has been added to discuss the differences in hygroscopic growth between different ranges of RH.

5. Discussion of the different hygroscopicity parameterizations (e.g., kappa-Kohler vs GADS, etc.) and their influence on $f(\mathrm{RH})$ based on literature evaluations

6. A case study of modeled $f(\mathrm{RH})$ for sea salt dominated cases at Graciosa (based on when the sea salt contributed > 95 % to the simulated aerosol composition) has been added to the manuscript. This allowed a detailed discussion on the sea salt component as an almost isolated component, where the model results could be compared to literature findings (e.g. $f(\mathrm{RH})$ for inorganic sea salt). As such we added:

(a) New Figure 5: scattering enhancement factor vs RH for Graciosa when the models simulated an aerosol dominated by sea salt.

(b) New Section 4.3: Graciosa (GRW) as a test case for modelled sea salt hygroscopicity.

In the following we detail the new text corresponding to these **Common comments**:

– We added Section 3.11 to the revised manuscript, which describes in a combined way all main model characteristics in addition to the individual model descriptions:

[revised manuscript text omitted]

– New text regarding the discussion of the different hygroscopicity parameterizations has been added to the Introduction section, to Section 4.1, and Section 4.4:

[revised manuscript text omitted]

In what follows below we explicitly respond to each reviewer suggestions, but will refer back to this **Common comments** section where appropriate.

**2 Reviewer 1**

*This study evaluates the model simulations of aerosol scattering enhancement factor using in situ measurements. Model evaluation of the hygroscopic growth of aerosols is a valuable contribution to modeling community. The authors clearly described the methods and results, but my impression is that this manuscript reads like a technical report, rather than a scientific paper. The temporal collocation and the definition of dry RH are the two major discussion points, but I think they are mostly technical issues, rather than scientific findings. The authors show the hygroscopic growth of aerosols in models is highly uncertain and varies across models. I think what is more important is to explain why there is large spread across models, and how the results of this study could help future improvement of modeling aerosol optical properties. Without sufficient explanations and scientific insights, I don't think this manuscript is suitable to be published at ACP that emphasizes scientific significance. I have the following comments that should be addressed.*

*1. The most important parameter that determines the f(RH) is the hygroscopic growth factor used in models, but the authors did not explain the differences across models. Table 2 only lists the factor for sea salt, but not other species. The hygroscopic factors of organic carbon, for example, have large uncertainties (Duplissy et al., 2011). What is the hygroscopic factor of OC used in these models? Same for other species. I'd suggest the authors explicitly compare hygroscopic growth factors for each species in each model, which may also help explain the differences across models.*

We thank reviewer # 1 for his/her comments. We have addressed this first comment by adding the information on the parameterized hygroscopic growth factors (new Table 3 in revised manuscript) and by now also assessing the role of chemical composition on $f$(RH) predicted by the models. The additional analysis of the modeled chemical composition turned out to be very valuable for our work and allowed further assessment of observed variability both between measured and modeled $f$(RH) and among the models themselves. Specifically, we have studied the relationship between simulated organic mass fraction and $f$(RH) and compared it with observations of this relationship (this new information is presented and discussed in section 4.1). One site (Graciosa) was often solely dominated by sea salt within the models and thus allowed a more detailed assessment of how models simulated the hygroscopicity of sea spray. We added a comparison with recent findings from the literature on sea salt hygroscopic growth (new Section 4.3 in revised manuscript). The main changes in the text of the manuscript related to both of these new additions can be found in the **Common comments** Section above.

*2. The light scattering coefficients used in models are not discussed either.*

We have not directly compared measured and modelled scattering. The ability of models to get the magnitude of scattering correct relative to observations (or not) is irrelevant because the scattering enhancement factor is independent of the absolute value of scattering - it's the ratio of wet to dry scattering. It does however depend on other factors such as aerosol size and chemical composition as well as model assumptions about hygroscopicity. We have now included discussion and analysis on the latter two items but investigating the effect of size is beyond the scope of our effort here.

*3. As the authors mentioned, f(RH) largely depends on the aerosol size distribution, mixing state and aerosol composition.*

*Model disagreements in f(RH) should also be attributed to these differences, but the authors did not explain how model differs in in size distribution and aerosol composition, and how this leads to differences in f(RH). For aerosol size distribution, the authors should explain how the size distribution is parameterized in model (e.g. log normal distribution? Modal radius?). For aerosol composition, a simple test would be to compare the aerosol chemical composition across models. I think this manuscript will be valuable contribution to if the authors could explain why the models differ, beyond the two technical issues.*

We agree and have now evaluated the relationship between simulated chemical composition and $f$(RH) - see changes to section 4.1, 4.2 and the new section 4.3 in the manuscript described in the **Common comments** section.

In addition, in Table 2 we provide basic information on the number of bins of the aerosol size distribution set in each model which provides a hint at the level of detail of each model's size distribution parameterization. However, a detailed evaluation of the size distributions within the models is beyond the scope of this paper. An example of the magnitude of the effort that would be required to assess the effect of size distribution differences is shown in Mann et al. (2014). We have modified the text and discussed these issues within the new Section 3.11 (text provided in **Common comments**).

*4. The authors mention the mixing state of aerosols may partially explain the narrower range of GEOS-family models, but I don't understand the reasonings for this. The light extinction should decrease when changing from external to internal mixing because the aerosol size becomes larger while the number concentration decreases (Curci et al., 2015). I'd suggest the authors conduct further analysis on this, and provide more convincing explanations.*

We have addressed this question in the new section 3.11. Please see previous **Common comments** section.

Moreover, we have added the following text to Section 4.1:

"The mixing state is another model assumption that could play a role in the observed differences amongst models. Curci et al. (2015) reported that aerosol optical properties calculated from bulk aerosol models which assume external mixing may be inherently different from the optical properties calculated from more detailed microphysical models which assume internal mixing. In contrast, Reddington et al. (2019) found modeled aerosol optical properties to be insensitive to mixing state and suggested the differences described in the Curci et al. (2015) study were more related to assumptions about size distribution than mixing state. In this study, a commonality among the models exhibiting low variability in $f$(RH) (e.g., the GEOS-family models and IFS-AER), is that they assume an external mixing state (Table 2). SALSA, however, also assumes an externally mixed aerosol but does not exhibit the narrow range in $f$(RH) seen for the other models making this assumption. This suggests that mixing state assumptions may not be the reason behind these differences, although we are unable to evaluate this further."

*5. The hysteresis effect of aerosols is an important factor for modeling hygroscopic growth, but it is not addressed in the manuscript. Few models consider the hysteresis effect, which could partially lead to model-measurement disagreement. The authors briefly mentioned this in the supplement, but I think this should be an important issue to discuss.*

We agree. As suggested by the reviewer more information about hysteresis effect has been added to the text in Section 4.1:

"It is useful to consider what causes the discrepancies between models and observations. Potential explanations for the model overestimates of $f$(RH) may be related to model assumptions about chemistry (e.g., the species included, hygroscopicity parameterizations for those species, assumptions about hysteresis, mixing state, etc.) or size distribution. We have already noted that it is beyond the scope of this paper to consider the impact of aerosol size distribution on scattering enhancement, but below we discuss hygroscopicity in relation to hysteresis, mixing state, hygroscopcity parameterization and chemical composition. Table 3 summarizes the parameterizations used as well as the hygroscopic growth factors, g(RH), at RH=90% and $\kappa$ parameters so that the model assumptions of hygroscopic growth can be more directly compared.

A deliquescent aerosol can exist in the liquid and solid phases at the same RH, an effect known as hysteresis (Orr et al., 1958). This means that, below its deliquescence RH but above its efflorescence RH, the corresponding scattering will be different depending on whether it is in a liquid or dry state. Deliquescent aerosols are typically inorganic species such as ammonium sulfate and sodium chloride. Modelling hysteresis is complex as the behavior differs for aerosols of mixed composition, relative to single component particles. The hysteresis effect is unlikely to be the cause of differences amongst the models as it has only been accounted for by two of the models considered in this study (CAM and CAM-Oslo). Moreover, $f$(RH) was calculated at RH=85 % to minimize discrepancies due to hysteresis because at that RH the particles will have undergone deliquescence. However, models may make different assumptions about water uptake at low RH which will affect $f$(RH) by impacting the denominator of the scattering enhancement equation, which will be of importance of strongly deliquescent aerosol. This is explored in more detail in sections 4.3 and 4.4."

Hysteresis is now also discussed in the new Section 4.3 for the specific case of sea salt, please see **Common comments** for the text of this new section.

*6. Line 15 of Page 13: Why is there seasonality of aerosol hygroscopicity in some sites? Is it due to seasonal variation of aerosol composition?*

As reviewer suggests aerosol chemical composition may be responsible for the seasonal cycle in f(RH). Changes in size associated with the change in chemical composition may also be a factor. The text has been improved (Section 4.2) with information from the literature mentioning the seasonal variation of aerosol composition and size at some of the sites of our study. The new text is:

"Changes in both aerosol composition and size can cause changes in scattering enhancement (e.g., Zieger et al., 2010; Titos et al., 2014a). Such changes could be driven by annual circulation changes bringing different air masses to a site (Sherman et al., 2015) and/or by normal variability in sources over the year. Both direct measurements of aerosol size distribution and indirect proxies such as the scattering Ångström exponent suggest there are seasonal shifts in aerosol size at these three sites (e.g., Quinn et al., 2002; Marinescu et al., 2019; Pio et al., 2007). Similarly, aerosol composition shifts as a function of season have also been reported for these sites (e.g., Quinn et al., 2002; Parworth et al., 2015; Logan et al., 2014). An in-depth evaluation of observed and modeled seasonal composition cycles at the 22 sites considered in our study is outside the purview of this paper. However, we can look beyond the annual mass mixing ratio comparisons (Fig. S4, discussed in the previous section) to

differences in the modeled monthly composition which may contribute to the variability in the modeled seasonal $f(\text{RH})$ shown in Fig. 4. Figures S5, S6 and S7 show the monthly variation in mass mixing ratio for the ten models considered in this study and for the year 2010 for these three sites. There is a fair amount of variability amongst the models in the simulated aerosol components at BRW and SGP. The variability in model chemistry for BRW and SGP suggests that at least some (if not all) of the models are simulating substantially different chemistry than is observed at those two sites.

While it is beyond the scope of this paper to do a detailed comparison of measured and modeled chemistry for all sites, some observations can be made. At SGP, Jefferson et al. (2017) note the importance of nitrate in determining $f(\text{RH})$, but many models do not include nitrate (see Table 2). From those models considering nitrate, only ATRAS, GEOS-Chem and TM5 show a marked annual cycle in nitrate, but only ATRAS simulates a $f(\text{RH})$ annual cycle at SGP which could just as easily be related to the OMF seasonal cycle as that of nitrate.

The models tend to simulate more consistent chemical composition at GRW. The temporal cycle of chemical constituents at GRW is dominated by sea salt (see Fig. S6) with the aerosol being almost entirely composed of sea salt in the winter months. This is consistent with observations of aerosol chemical composition in the region (Pio et al., 2007) and suggests perhaps wind-driven sea salt emissions are better parameterized than other aerosol species. Despite the similar estimates of chemical composition among the models at GRW, Fig. 4 shows that some models (TM5, CAM and CAM-Oslo) simulate significantly higher $f(\text{RH}=85\,\% \,/\, \text{RH}_{\text{ref}}=40\,\%)$ at GRW throughout the year. Because the chemistry simulated is generally consistent across the models and because models assume very similar hygroscopic growth for sea salt at high RH (Table 3), some other factor is causing these three models to be biased high. One possibility, which was alluded to previously, is how water uptake is modeled at low RH. Figure S8 shows that the models that exhibit the least growth between 0 % and 40 % RH are the models that simulate the highest $f(\text{RH})$ in Fig. 4. In the next section we explore this for the specific case of sea salt hygroscopicity."

*7. Figure 2: I'd suggest include the number of data pairs in the figures.*

As suggested, the number of data pairs has been added to each subplot.

*8. Figure 3: The seasonal variation in hardly seen. I think you could change the y axis limit to a smaller range?*

As suggested, the y-axis limits have been narrowed in Figure 4 (previously Figure 3).

References:

Duplissy, J, P F DeCarlo, J Dommen, M R Alfarra, A Metzger, I Barmpadimos, A S H Prevot, et al. 2011. "Relating Hygroscopicity and Composition of Organic Aerosol Particulate Matter." Atmospheric Chemistry and Physics 11 (3), 1155– 65. doi:10.5194/acp-11-1155-2011.

Curci, G, C Hogrefe, R Bianconi, U Im, A Balzarini, R Baró, D Brunner, et al. 2015. "Uncertainties of Simulated Aerosol Optical Properties Induced by Assumptions on Aerosol Physical and Chemical Properties: an AQMEII-2 Perspective." Atmospheric Environment 115 (c). 541–52. doi:10.1016/j.atmosenv.2014.09.009.

**3   Review 2**

*This paper discusses an important aspect of aerosol modelling: the growth of aerosol with increasing humidity. This wet-growth has important implications for aerosol transport and aerosol optical properties (including aerosol-radition-interactions). The authors conduct a process study by comparing the light scattering enhancement factor f(RH) from models with that from sev-*

5  *eral observation sites. The observational data is novel and the evaluation of models highly relevant to numerous modelling groups, including the AEROCOM community.*

*General comments: The study itself seems properly conducted and the paper is mostly well-written. However, it seems the authors were only interested in a very narrow objective: numerically comparing model data with observations. There is very little*

10  *interpretation of the results. While I appreciate that a full interpretation might be a study in itself (and the authors suggest they are working on it), this is rather unsatisfying. It prevents the reader from tapping in to the combined expertise of the authors (observers and modellers) and a better understanding of the limitations and opportunities associated with the current study.*

*In particular, I missed a discussion of why models might have a different f(RH) from the observations. I suppose there are at*

15  *least three reasons: 1) incorrect wet-growth of individual species (e.g. incorrect kappa); 2) incorrect internal mixing rule for wet growth; 3) incorrect internal and external mixing states. The advantage of a process study is of course that the models do not need to accurately simulate mass loads themselves. If possible, it would be useful to provide more information on per-species wet-growth in individual models, especially because of the finding of substantial wet-growth at low RH.*

We thank the reviewer for his or her helpful comments. The main criticism was also raised by the first reviewer. We have

20  addressed these issues by adding an analysis of the modeled chemical composition and added missing details and discussion to the revised manuscript. To see the new information related to per-species wet-growth, please read the **Common comments** section particularly with respect to the description of the new Table 3.

*In addition, I found important information on e.g. observational errors and methodology to be missing. Yes, the authors refer to Burgos et al 2019 but it would be good if brief (!) summaries of relevant sections in Burgos et al 2019 are provided.*

25  As the reviewer suggests, we have added more information about observational errors and methodology in Section 2, the new text is:

"As part of the observational dataset development, uncertainty in $f(RH)$ was also determined. The uncertainty in $f(RH)$ depends on the aerosol load, RH and hygroscopic growth, and was found to vary between 10 and 30 % for $PM_{10}$. Table 4 in Burgos et al. (2019) presents a detailed description of the uncertainty as a function of these variables.

30  A full description of the homogenization process is given in Burgos et al. (2019), and a summary of the process is presented here. The homogenization starts with the light scattering raw data provided by each site manager. Standard corrections are applied to all raw data in an identical manner, and in-depth data screening is carried out to identify data during invalid periods or system malfunctions. Several corrections are applied to the valid data periods: angular truncation and illumination nonidealities, adjustment to standard temperature and pressure, particles losses, and a 10-minute moving average is applied to the dry scattering coefficient series (this step is specially relevant for pristine sites). Finally, the scattering enhancement factors are reported at common $RH_{ref}$ and $RH_{wet}$ which eliminates potential discrepancies among $f(RH)$ values due to choice of RH (Titos et al., 2016)."

5 *Specific comments:*

*Abstract: the abstract contains several conclusions without any attempt at interpretation. E.g. "An important finding is that the models show a significantly larger discrepancy with the observations if RHref =0 % is chosen as the model reference RH compared to when RHref =40 % is used" might become "The definition of dry conditions is difficult from an observational point-of-view, which affects our model evaluation negatively as several models exhibit significant and unexpected wet-growth*

10 *between RH=0 and 40%.". One interesting finding (also supported by a recent Gliss study) is not included in the abstract.*

The abstract has been revised to reflect the additional analysis and interpretation that was done during the revision of the manuscript. In addition, this specific change has been implemented in the abstract. The revised text is as follows (original text in black):

"The uptake of water by atmospheric aerosols has a pronounced effect on particle light scattering properties which in turn

15 are strongly dependent on the ambient relative humidity (RH). Earth system models need to account for the aerosol water uptake and its influence on light scattering in order to properly capture the overall radiative effects of aerosols. Here we present a comprehensive model-measurement evaluation of the particle light scattering enhancement factor $f(RH)$, defined as the particle light scattering coefficient at elevated RH (here set to 85 %) divided by its dry value. The comparison uses simulations from 10 Earth system models and a global dataset of surface-based in situ measurements. In general, we find a large diversity

20 in the magnitude of predicted $f(RH)$ amongst the different models which can not be explained by the site types. Based on our evaluation of sea salt scattering enhancement and simulated organic mass fraction, there is strong indication that differences in the model parameterizations of hygroscopicity  and model chemistry are driving at least some of the observed diversity in simulated $f(RH)$. Additionally, a key point is that defining dry conditions is difficult from an observational point of view and, depending on the aerosol, may influence the measured $f(RH)$. The definition of dry also impacts our model

25 evaluation because several models exhibit significant water uptake between RH=0 % and 40 %.  $RH_{ref}$ $RH_{ref}$ The multi-site average ratio between model outputs and measurements is 1.64  when RH=0 % is assumed as the model dry RH and 1.16  when RH=40 % is the model dry RH value. The overestimation by the models is believed to originate from the hygroscopicity parameterizations at the lower RH

30 range which may not implement all phenomena taking place (i.e. not fully dried particles and hysteresis effects). This will be particularly relevant when a location is dominated by a deliquescent aerosol such as sea salt. Our results emphasize the need to consider the measurement conditions in such comparisons and recognize that measurements referred to as 'dry' may not be dry in model terms. Recommendations for future model-measurement evaluation and model improvements are provided."

*Introduction: I miss a discussion of the importance of correctly simulating wet-growth in models. How does wet-growth affect different aspects of simulation? E.g. emissions are unaffected but both wet and dry deposition are. Consequently, so is transport. At the same time, optical properties (important for ARI) are affected. Anything else (e.g. chemistry)? While wet-growth ultimately leads to activation of aerosol as cloud droplets, models usually disconnect these processes.*

5   *Two papers that consider impact of f(RH) on modelling of biomass burning:* Johnson, B. T., Haywood, J. M., Langridge, J. M., Darbyshire, E., Morgan, W. T., Szpek, K., ... Bellouin, N. (2016). Evaluation of biomass burning aerosols in the HadGEM3 climate model with observations from the SAMBBA field campaign. Atmospheric Chemistry and Physics, 16, 14657–14685. https://doi.org/10.5194/acp-16-14657-2016. Reddington, C. L., Morgan, W. T., Darbyshire, E., Brito, J., Coe, H., Artaxo, P., Scott, C. E., Marsham, J., and Spracklen, D. V.: Biomass burning aerosol over the Amazon: analysis of aircraft, surface and

10   satellite observations using a global aerosol model, Atmos. Chem. Phys., 19, 9125–9152, https://doi.org/10.5194/acp-19-9125-2019, 2019.

We agree and as suggested, we have included more information about the effects of water uptake by adding the following to the introduction:

""Water uptake by aerosols affects not only their optical properties but also their life cycle by changing their size which can

15   impact processes such as wet and dry deposition, transport, and ability to act as cloud condensation and ice nuclei (Covert et al., 1972; Pilinis et al., 1989; Ervens et al., 2007).

Similarly, Reddington et al. (2019) studied the sensitivity of the aerosol optical depth (AOD) simulated by the GLOMAP model to assumptions about water uptake. They found that the AOD decreased when using the $\kappa$-Köhler (Petters and Kreidenweis, 2007) water uptake scheme relative to the AOD calculated using the Zdanovskii–Stokes–Robinson approach (Stokes and

20   Robinson, 1966). Moreover, Latimer and Martin (2019) also found that the implementation of the $\kappa$-Köhler hygroscopic growth for secondary inorganic and organic aerosols reduced the bias that appears in the representation of aerosol mass scattering efficiency relative to when water uptake was based on the Global Aerosol Data Set (GADS)."

*An introduction should end with a: - a brief description of the paper's methods and goals, and why/how they advance the field (i.e. add to the existing body of work) a short description of the content (sections listing). I miss both and the introduction*

25   *would become substantially stronger if they are added. P 3, l 20-34 does not really provide this.*

We address this comment by adding the following text to the introduction:

"In this paper, we present a comparison among scattering enhancement factors modeled by 10 different ESMs and observations. Our objectives are (i) to use measurements as a reality check on model simulations, (ii) to assess differences amongst model estimates of aerosol hygroscopic growth and then (iii) to suggest some potential reasons for any observed discrepancies,

30   both between models and measurements and amongst models. This is the first comparison carried out for a wide suite of site types (covering Arctic, marine, mountain, rural, urban and desert stations) and ESMs, and is possible due to a newly published observational dataset of aerosol hygroscopicity (Burgos et al., 2019). A short description of the measurement dataset is presented in Sect. 2, while Sect. 3 gives a brief description of the models and the main references related to them. Section 4 shows the results of the model-measurement comparison for 22 sites and we evaluate the influence of different model choices

about chemical species and mixing states on this comparison. We explore the importance of temporal collocation for three sample sites where temporal collocation is possible and use the unique chemical composition at one of these sites to interpret model results in the context of the hysteresis phenomenon. Finally, we demonstrate the importance of the definition of the dry reference relative humidity for hygroscopicity studies."

5    *4, sect 2: what I miss here is a brief discussion on expected uncertainties. I have no idea how much can be said about this but admitting to 'uncertainty in the uncertainty' would be acceptable. E.g. at RH=40 or 85%, can we expect measurement uncertainties in f of 5, 10, 15% in individual measurements? Describe briefly the main causes of uncertainty. Do you expect errors at a single site to behave like biases or random errors? This discussion would be very useful. If this was discssed in Burgos et al 2019, please provide highlights and the reference.*

10   We agree and have added more information on the measurement uncertainties to Section 2:

"As part of the observational dataset development, uncertainty in $f(\text{RH})$ was also determined. The uncertainty in $f(\text{RH})$ depends on the aerosol load, RH and hygroscopic growth, and was found to vary between 10 and 30 % for $PM_{10}$. Table 4 in Burgos et al. (2019) presents a detailed description of the uncertainty as a function of these variables."

     *p4,l 16: given the uncertainty in the analysis at low RH later identified, it would be good to be a bit more precise about low RH*
15   *measurements? Do the various instruments measure at different low RH? Does a single instrument measure at a single RH, or does it vary for some reason? If it varies, does it vary in a controlled fashion , or not? I see that Table 1 only provides median values and gives not indication of any variation (large or small).*

     We added information about low RH measurements in the revised manuscript by adding the following text to Section 2:

     "The scattering coefficients were measured simultaneously under two different conditions. First, under so-called dry or low-
20   RH conditions (namely RH < 40 %), hereafter referred to as $RH_{ref}$, and measured with a reference nephelometer or DryNeph. Typically $RH_{ref}$ in the DryNeph will vary over the interval 0-40 % but this variation will depend on the characteristics of the site, e.g., at some marine sites like at GRW, the measurement system was not able to dry the aerosol below 50 % RH during some months. Data with $RH_{ref} > 40\%$ were not included in this study. Figure S2 presents the probability density function of the measured $RH_{ref}$ for all sites."

25   *p 5, Section models: it seems that the models broadly fall into 2 (3?) categories based on how f(RH) is calculated: either from direct parametrisation (e.g. OPAC), or from Koehler theory (is it fair to include ZSR theory?), or from equilibrium theory (sulphate-nitarte-amoonia, see e.g. Seinfeld and Pandis). I'm not sure the latter category is present in the current family of models but maybe TM5 uses it? Anyway, would it make sense to stress these three broad categories?*

     As suggested, we have now indicated categories which we can group the models based on how they calculated $f(\text{RH})$. Please
30   see the text for the new Section 3.11 in the **Common comments** section. We also highlighted this fact with a new table (Table 3) with more details about the hygroscopic growth factor and its implementation for each model and species.

*p 5, l 3: can the authors say a little about what aspects of data homogenization are considered important, and were different in Titos et al.? It would help make the paper stand out more.*

The aspect of data homogenization is indeed important and described in detail by Burgos et al. (2019). The homogenized treatment e.g. allowed that all scattering enhancement factors were given at the same RH. Since the dry reference scattering coefficient is also important but often varied significantly, we also provided that scattering enhancement factor with $RH_{ref} =$ 40% as a common dry reference. As an interpolation method, the most common parameterization ($\gamma$-fit) was used for all humidgram measurements. We have added further information on the data homogenization to the manuscript to Section 2:

"A full description of the homogenization process is given in Burgos et al. (2019), and a summary of the process is presented here. The homogenization starts with the light scattering raw data provided by each site manager. Standard corrections are applied to all raw data in an identical manner, and in-depth data screening is carried out to identify data during invalid periods or system malfunctions. Several corrections are applied to the valid data periods: angular truncation and illumination non-idealities, adjustment to standard temperature and pressure, particles losses, and a 10-minute moving average is applied to the dry scattering coefficient series (this step is specially relevant for pristine sites). Finally, the scattering enhancement factors are reported at common $RH_{ref}$ and $RH_{wet}$ which eliminates potential discrepancies among $f(RH)$ values due to choices of RH (Titos et al., 2016) and allows direct comparison between sites."

*p 5, l 26: So the GEOS family of models are CTMs? Isn't MERRAero an assimilation product?*

A more detailed description of these models can be found in sections 3.4, 3.5 and 3.6, but briefly:

Not all GEOS-family models are CTMs. GEOS-Chem is a CTM that uses off-line assimiliated meteorological fields produced by GMAO (NASA) using their GEOS GCM. But GEOS-GOCART results reported to this study are from online simulations, and GEOS-MERRAero includes the same aerosol transport module based on GOCART, and this specific version also includes assimilation of bias-corrected AOD from MODIS sensors.

*p 5, l 27: There is no information in Table 2 on hygrospcopic growth of sulphate (most important anthropogenic aerosol) and organics (to my knowledge, the most uncertain aerosol wrt hygrospcopic growth). Is it possible to include at least these species?*

As suggested by the two reviewers a new table, Table 3, has been added with the per-species hygroscopic growth factors.

*p 5, l 29: is there a name to identify the relevant AEROCOM experiment? Please also provide link to AEROCOM website so interested readers can follow up.*

Yes, the relevant information has been added to the text in Section 3, Models:

"The model data used in this study were provided within the tier III of the INSITU measurement comparison experiment of AeroCom phase III (https://wiki.met.no/aerocom/phase3-experiments) and are composed of aerosol absorption and extinction coefficients at RH = 0, 40, and 85 %. Models also provided the mass mixing ratios for the chemical constituents they simulated, which we use to assess the impact of composition on hygroscopicity."

*p 5, l 32: I assume surface values were used? PLease state so.*

Yes, surface values have been used. We have clarified this by adding text to Section 3:

"The models were run for the year 2010 and data at surface level from 22 locations (closest gridpoint to the observational data) have been extracted."

5   *p 6, l 1: Here or at a more appropriate location, it would be good to have a brief discussion of possible impact of the difference between model and observational years.*

A reference to this point has been added in the main text (Section 4, and Section 4.2), and supplement material:

"Model output for the simulation year 2010 is selected only from those months where measurement data is available (regardless of the year the measurements were made). We included all model data for each month for a given site regardless of the number

10   of measurement data points in that month and for that site. Analysis (not shown) requiring a constraint on the number of measurements in a month in order to include model simulations for that month suggested that our approach had minimal impact on the results. By selecting the entire month from the model dataset, the impact of interannual variability is minimized. An illustration of the possible impact of the difference between model and observational years can be found in the supplemental materials for the site SGP, which has the longest period of measurements (see Fig. S3). In Sect. 4.2 we perform a more detailed

15   analysis for three sites that measured during 2010, and thus allow an exact temporal collocation with the models, collocating for day and month of the year 2010.

Figure S3 shows the annual cycle in $f$(RH) for each individual year of measurements at SGP, the site with the longest time coverage (1999-2016); just 3 out of 18 years exhibit deviations from the climatological values larger than 50 %, suggesting the climatological values are a reasonable proxy for comparison with model values.

20   Figure S3 shows the annual cycle of $f$(RH=85 % / RH=40 %) at Southern Great Plains for the whole period of measurements (1999-2016) with the black line representing the monthly median values and the gray shading representing the interquartile range for all measurements. The monthly median values for each individual year are showed with gray lines. The year 2010 is shown in red (2010 is the year that is compared to model outputs). A threshold of more than 10 data points per month has been required to minimize the influence of possible outliers on the monthly values. This figure gives an idea of the possible

25   impact of considering different years in the model-measurement comparison. The years showing largest differences with the climatological average values are 2004, 2005, and 2006, with the largest differences found in May (63.2 %, 62 %, and 67 %, respectively) and September (52 %, 85.3 %, and 65.5 %, respectively). In contrast, the average difference (over the 18 years considered in SGP) for a given month between a single year and the climatological average, varies between 11 (January) and 23 % (May)."

30   *p 7, l 16: (Bey at al.) -> Bey et al.*

This change has been implemented

*p 10, l 7: all hourly output for that month is used, or only at the hours (and presumably days) of observations. Given the possible importance of representation errors, it may be worthwhile to have a short section (Methods) that describes actual collocation procedure in a bit more detail, especially as there are basically two cases: sites that provide data for 2010, and sites that don't.*

As suggested, we have improved this explanation about the collocation procedure. However, we don't consider that a new section is needed and hope that the reviewer agrees with us that this explanation is enough to clarify further the procedure that we used. Note that the collocation procedure is linked with the fact that we are using different years in models and measurements, so, in addition, we have included in the supplement (Figure S3) the annual cycle of $f$(RH) at SGP for all individual years along with the climatological median to evaluate the validity of using climatological comparisons rather than exact temporal collocation. The new and more complete description for the collocation procedure is described in Section 4, Results:

"Model output for the simulation year 2010 is selected only from those months where measurement data is available (regardless of the year the measurements were made). We included all model data for each month for a given site regardless of the number of measurement data points in that month and for that site. Analysis (not shown) requiring a constraint on the number of measurements in a month in order to include model simulations for that month suggested that our approach had minimal impact on the results. By selecting the entire month from the model dataset, the impact of interannual variability is minimized. An illustration of the possible impact of the difference between model and observational years can be found in the supplemental materials for the site SGP, which has the longest period of measurements (see Fig. S3)."

We have also highlighted this aspect in Section 4.2:

"Because the focus in this section is to study the importance of temporal collocation, no threshold on number of data points within each month was required; the number of data points in each month are provided in Fig. 4 to give an indication of the representativeness (or lack thereof) of the monthly value."

*p10, l20: is the gray shaded area for measurements mostly due to measurement error or temporal variability? This has consequences, because if it is due to measurement error, a lot of models greatly overestimate f(RH) variability.*

We have clarified the definition of the gray shaded area by adding the following text in Section 4.1:

"This area represents the temporal variability over the time period of the $f$(RH) measurements for each site and does not include measurement error."

*p 10, l 23: this information could be part of the short section suggested above?*

Rather than creating a new subsection in the main "Methods" section, we have improved the explanation that we previously provided in Section 4, as explained in more detail in the previous comment refering to pg10, l7. In this case, we consider that this information about the number of points and the example of MHD can stay in this section 4.1, since it is a useful example to illustrate collocation applied for the boxplot (Figure 1) that is discussed.

*p 10, l 27: what is the altitude of these mountain sites? Is it ok to use those measurements for model evaluation when model's orography may not ab able to 'simulate' the mountain?*

All models take into account orography, however given the coarse resolution of some of the models the surface data may not represent the site elevation or local topography. We have clarified this point in Section 3:

"All models considered in this study take into account topography. However, a model's surface elevation for a given gridbox will represent an average of the topography within the given gridbox. Nonetheless, we have used the surface values provided by the models for all sites in this study. For sites located in complex terrain the model surface values may not be representative of the measurement site and this will be exacerbated by models with coarser resolution. For example, Schacht et al. (2019) noted that complex local terrain near ZEP may have impacted their modeling efforts. In this study there is one mountain site (JFJ) in the Swiss Alps with an altitude of 3580 m a.s.l. and seven more sites with elevations above 200 m a.s.l. (APP, FKB, HLM, NIM, PGH, UGR, and ZEP at 1100, 511, 525, 205, 1951, 680, and 475 m a.s.l., respectively). The remaining 14 stations are at elevations lower than 100 m a.s.l. It should be noted that elevation alone does not describe the wider topography; for example, UGR is surrounded by nearby mountains with elevations above 3000 m a.s.l. (Titos et al., 2014b); while PGH is located on the edge of the Indo-Gangetic Plain in the foothills of the Himalayas (Dumka et al., 2017)."

*p 11, l 10-20: ignoring the observations, it appears to me that the models exhibit fairly similar behavior independent of site. E.g. GOES family rather low f(RH) with small variation, CAM, TM5 and SALSA higher f(RH) with more variation. Can anything be said about that?*

We agree with the reviewer. To highlight this point, we have added the following possible explanation related to the hygroscopicity values for different species to the revised manuscript in Section 4.1:

"As noted in Section 3, the hygroscopicity values are generally quite similar for sea salt, sulfate, and dust for all models. There are, however, large differences for BC, POA and SOA amongst the models. The GEOS-family of models assign significantly higher growth for these three species than assumed by the other models. This may, in fact, be the explanation for the narrow range of $f$(RH) exhibited by the GEOS-family of models - regardless of the simulated composition there will always be a large amount of water uptake. In contrast, the other models can simulate a wider range of $f$(RH), i.e., from low to high $f$(RH), as the proportions of the chemical constituents shift."

*p 11, l 21: if I'm correct, this is the same data as in Fig 1 but represented in a different way? It may be good to specifically state so.*

The reviewer is right, is the same data visualized in a different way. To note this, we have updated the text in Section 4.1:

"The data shown in Fig. 1 can be visualized in a different way in order to more readily see the relation between modeled and measured data for each model rather than for each site."

*p 11, l 23: This uncertainty should be mentioned earlier, when describing the measurements. Is this the uncertainty in a single measurement, or in an average of measurements?*

As suggested by the reviewer, a clearer reference to measurement uncertainty has been added earlier in the text. This is explained in the previous comment: p 4, sect 2. We have added more information on the measurement uncertainties to section 2:

"As part of the observational dataset development, uncertainty in $f(\text{RH})$ was also determined. The uncertainty in $f(\text{RH})$ depends on the aerosol load, RH and hygroscopic growth, and was found to vary between 10 and 30 % for $PM_{10}$. Table 4 in Burgos et al. (2019) presents a detailed description of the uncertainty as a function of these variables."

*p 11, l 31: Is it fair to conclude that external mixing reduces variation in f(RH)? And if so, why would that be?*
From our results we cannot conclude that mixing state is the explanation for the reduced variation in $f(\text{RH})$. To address this point, we have added the following text to Section 4.1: (note that this is also explained in comment 4 from reviewer 1.)

"The mixing state is another model assumption that could play a role in the observed differences amongst models. Curci et al. (2015) reported that aerosol optical properties calculated from bulk aerosol models which assume external mixing may be inherently different from the optical properties calculated from more detailed microphysical models which assume internal mixing. In contrast, Reddington et al. (2019) found modeled aerosol optical properties to be insensitive to mixing state and suggested the differences described in the Curci et al. (2015) study were more related to assumptions about size distribution than mixing state. In this study, a commonality among the models exhibiting low variability in $f(\text{RH})$ (e.g., the GEOS-family models and IFS-AER), is that they assume an external mixing state (Table 2). SALSA, however, also assumes an externally mixed aerosol but does not exhibit the narrow range in $f(\text{RH})$ seen for the other models making this assumption. This suggests that mixing state assumptions may not be the reason behind these differences, although we are unable to evaluate this further."

*p 12, l 2: are you sure? I thought SALSA uses a combination of internal and external missing.*
Yes, for particles sizes that affect radiation (> 100 nm in diameter), SALSA has two parallel externally mixed size distributions for insoluble and soluble particles.

*p 12, l 27: for SGP I also see substantial differences but they seem to their own annual cycle (larger differences in summer).*
We agree and have added the missing information about the largest differences also found in SGP annual cycle to the revised manuscript.:

"The exceptions are for BRW in the latter half of the year where the 'all data' climatology is ∼12 % lower than the 2010 values, and for SGP where August and October exhibit monthly 2010 values lower than the climatological values (28 % and 20 %, respectively)."

*p 12, l 28-30: nice analysis, ultimately uncertainties always have to be considered in the context of other uncertainties.*
As suggested by the reviewer also in the previous comment (p4, sect2), we have added further information about uncertainties of $f(\text{RH})$ to the revised manuscript.

*p13,l15-17: Dare the authors guess at why even CAM, OsloCTM and TM5 do poorly? They showed pretty strong correlation across the sites. So it appears that these models have skill in predicting spatial patterns but none in predicting temporal evolution. Yet it would appear that observed variation across sites is similar to variation within each site (guesstimating from Fig 1).*

In the new analysis of the annual chemistry (Figs S5-S7) one can observe the variability of simulated chemistry among the models. A more detailed evaluation of the annual estimation of chemical composition for each site could help to elucidate these differences, but that is beyond the scope of this paper. Our new analysis about sea salt (Section 4.3), suggests that the hygroscopic growth of sea salt may be one factor that plays a role. However, the role of particle size and more detailed chemical composition definitely need further analysis.

*p 14, l 26-27: surely the developers of the GEOS and SALSA models (who are coauthors) must have an idea where this is coming from? It would be very useful to add such information to the paper. From a modelling perspective, it would be relatively straight forward to present f(RH) curves for each species (i.c. of internal mixtures: predominant species). Such a figure would be a very useful addition to this paper.*

We have addressed this problem by showing the $f$(RH) curves for sea salt and its corresponding analysis (Section 4.3). At the site of Graciosa, the models predict for certain months that the aerosol particles consist almost entirely of sea salt. We used this site and compared the $f$(RH) values to recent findings from laboratory studies of inorganic sea salt (Zieger et al., 2017) and to field observations of pristine sea spray from the Arctic (Zieger et al., 2010). This work turned out to be very useful and helped to evaluate the effects of both hysteresis and model assumptions about sea salt hygroscopicity. Please see the **common comments** section above for details.

*p 15, l 10-15: this is an interesting finding that should be in the abstract (there is only an oblique reference to it at the moment).*

We also find interesting that these two papers (the present one and the study by Gliss et al., (2019)) lead to similar conclusions and we mention it in the text. Nevertheless, we consider it to be sufficient that it is noted in the main text as it is. However, we added to the abstract that recommendations for future model-measurement evaluation and model improvements are given in the text.

*p 15, Section Conclusions: the current text focuses entirely on technical issues and ignores interesting findings as mentioned above. It would be good if more emphasis is given to lessons learned with regards to possible model deficiencies. There is no speculation why models over-estimate f(RH). Or do the authors believe this is due to remaining technical issues?*

We agree and as the reviewer suggested, we have updated the conclusion section focusing more on the findings and interpretation, particularly in the light of our new analysis on chemical composition (e.g., the OMF parameterization described in section 4.1 and the sea salt investigation in section 4.3). We have also added some recommendations for future model evaluation based on our results. The new conclusions section reads as follows (original text is shown in black):

[revised manuscript text omitted]

*p 15, l 17: define "f(RH)"* Done, the sentence now reads: "Model simulations of the scattering enhancement factor $f(\text{RH})$..."

*p 15, l 19: define "RH ref"* Done, the sentence now reads: "...but tended to overestimate $f(\text{RH})$ relative to the measurements when the reference relative humidity is $\text{RH}_{\text{ref}}$=40 %."

*p 15, l 27: "at low RH". Did you mean "for low RH ref"?* No, we mean "at low RH". RH ref is used when talking about the DryNeph in the measurements. In this sentence we refer to low RH for $f(\text{RH})$, so, the scattering enhancement between, for example, RH = 0 and 40 %.

*p15,l30: models should be able to provide f(RH) at multiple RH without any significant CPU or development overhead.*

We agree with the reviewer. Once the models have been set up it should not be so difficult to provide $f$(RH) at multiple RHs. We have modified the sentence in the conclusion section indicating that this methodology will make the comparison more challenging in order to match same RH in models and measurements:

"For example, models could calculate $f$(RH) at the same variable RH conditions as the measurements. This type of study will make the model-measurement comparison more challenging since the same RH conditions should be matched and measurement conditions can vary widely with site and season."

*p 16, l 4-6: this gamma parameter has not been mentioned before so it's odd to discuss in the conclusions. Obviously, you may want to mention other possible analysis methods.*

We agree. As suggested by the reviewer, a reference to the gamma fit parameter has been added previously, in Section 2: Measurements, so that it makes more sense to reference it in the conclusions.

"In this study we utilize the scattering enhancement at $RH_{wet}$=85% to parameterize aerosol hygroscopicity. Choosing $RH_{wet}$=85% ensures that the reported $f$(RH) value represents the aerosol in the fully deliquesced state (upper branch of the hysteresis loop). Scattering enhancement at specified RH is a simple metric. There are other methods, of varying complexity, that may also be used to describe the aerosol scattering enhancement; Titos et al. (2016) presents a review of the various empirical parameterizations found in literature that have been used to describe the relationship of $f(\mathrm{RH}, \lambda)$ and RH. The most common other algorithm is the two-parameter, power law fit referred to as the $\gamma$-fit (Hänel and Zankl, 1979). While fitting over the whole range of RH observations can provide valuable additional information about hygroscopic growth (e.g., investigating the RH ceilings often assumed in models or as a means to identify deliquescence transitions (Zieger et al., 2010; Titos et al., 2014a)) that level of complexity was not desired in this initial model measurement comparison."

**4 Further changes:**

In addition to changes raised by the reviewers, we have modified the manuscript after discovering bugs in the model output files from IFS-AER and OsloCTM3, and after the discovery of minor bugs in the plotting analysis scripts. As such, some parts of the text had to be modified accordingly.These changes are summarized below:

Coauthors list: We have added two more coauthors given their implication in the running of the new input files of ECMWF model data: Zak Kipling and Julie Letertre-Danczak.

In the introduction, we added several references to model chemistry and model assumptions of water uptake (which is shown later in the comparison to the organic mass fraction). These new and modified sentences are (original text is given in black):

"Quinn et al. (2005) utilized co-located chemistry and $f$(RH) measurements to develop a parameterization relating organic mass fraction and water uptake based on measurements at sites in Canada, the Maldives and South Korea."

"This linear relationship was extended for organic-dominated aerosol with observations from a boreal site in Finland (Zieger et al., 2015)"

"Previous model studies have suggested that water associated with aerosol particles can lead to significant differences amongst model estimates, and the assumptions about water uptake can have a noticeable effect. For example, Haywood et al. (2008) used tandem-humidifier nephelometer measurements from an aircraft to assess the parameterization of aerosol water uptake by the Met Office Unified Model. They found that ambient aerosols were simulated as being too hygroscopic relative to observations as a result of being modeled as composed solely of ammonium sulfate."

Section 3, Models: In this section we have added information about the new model chemistry data we have used:

"Models also provided the mass mixing ratios for the chemical constituents they simulated, which we use to assess the impact of composition on hygroscopicity."

Section 3.8 OsloCTM3: We have updated information about hygroscopic growth of inorganics by adding the following sentence:

"The parameterization from Fitzgerald (1975) on hygroscopic growth for inorganic aerosols has been shown to be very similar to using Köhler theory in OsloCTM3 (Myhre et al., 2004)."

Section 3.9 ECMWF: This section has been updated to describe the model more accurately.

[revised manuscript text omitted]

Our Figures have been updated due to the revised model output files from OsloCTM3 and ECMWF. In addition, we found a bug in our script for reading the models MERRAero and CAM-Oslo. We have updated the following figures (figure numbering as in revised manuscript): Fig. 1 (comparison of all sites and models using box plots), Fig. 2 ($sf$RH Model vs measurements), Fig. 4 (Annual cycles and Taylor diagrams), Fig. 6 (PDF $f$(RH) ratio), and Fig. S10 (Relative frequency of model $f$(RH) between RH=0-40 %). Moreover, we found an error in the new Fig. S2 related to the measured dry reference RH for the sites CES, HYY, JFJ, MEL, MHD, UGR, SGP and ZEP, which has been corrected.

**References**

[revised manuscript text omitted]